# Spatiotemporal ecological chaos enables gradual evolutionary diversification without niches or tradeoffs

**Aditya Mahadevan[1†], Michael T Pearce[1†‡], Daniel S Fisher[2*]**

[1]Department of Physics, Stanford University, Stanford, United States; [2]Department of Applied Physics, Stanford University, Stanford, United States

**Abstract** Ecological and evolutionary dynamics are intrinsically entwined. On short timescales, ecological interactions determine the fate and impact of new mutants, while on longer timescales evolution shapes the entire community. Here, we study the evolution of large numbers of closely related strains with generalized Lotka Volterra interactions but no niche structure. Host-pathogen-like interactions drive the community into a spatiotemporally chaotic state characterized by continual, spatially-local, blooms and busts. Upon the slow serial introduction of new strains, the community diversifies indefinitely, accommodating an arbitrarily large number of strains in spite of the absence of stabilizing niche interactions. The diversifying phase persists — albeit with gradually slowing diversification — in the presence of general, nonspecific, fitness differences between strains, which break the assumption of tradeoffs inherent in much previous work. Building on a dynamical-mean field-theory analysis of the ecological dynamics, an approximate effective model captures the evolution of the diversity and distributions of key properties. This work establishes a potential scenario for understanding how the interplay between evolution and ecology — in particular coevolution of a bacterial and a generalist phage species — could give rise to the extensive fine-scale diversity that is ubiquitous in the microbial world.

## Editor's evaluation

This important study explores the question of "what gives rise to diversity in ecological settings?". By considering the interplay between ecology and evolution, this study proposes a scenario of spatiotemporal chaos, in which interactions between strains drive large changes in the relative abundances of strains. The presented theoretical approach is compelling and goes beyond the current state of the art. This innovative theoretical work is of broad interest to the field of ecology and evolution.

***For correspondence:** dsfisher@stanford.edu

[†]These authors contributed equally to this work

**Present address:** [‡]Meta Data Science, Menlo Park, United States

**Competing interest:** The authors declare that no competing interests exist.






## Introduction

A remarkable discovery of the DNA sequencing revolution is the vast diversity of microbes (***Biller et al., 2015***; ***Acinas et al., 2004***; ***Kashtan et al., 2017***; ***Rosen et al., 2018***). Increasingly it has become clear that this diversity extends far below the level of conventionally defined species to finer and finer genetic scales (***Bonilla-Rosso et al., 2020***; ***Kashtan et al., 2014***; ***Rosen et al., 2015***; ***Tikhonov et al., 2015***), and in some cases, a great multitude of strains coexist and compete in the same spatial location. Why doesn't "survival of the fittest" drive almost all strains extinct, at least locally? Traditional explanations invoke the existence of a great many spatial or functional niches which limit competition between strains, down to "micro-niches" involving finer differences. However, especially for bacteria in relatively simple environments such as the marine cyanobacterium *Prochlorococcus* (***Kashtan et al.,***

***2014***), does it make sense to postulate nano- or pico-niches, *ad absurdum*? Or is a statistical description of the small subtle differences more appropriate? Community ecology models with many similar strains competing for a mixture of resources have been much studied, but in their simplest manifestations the maximum number of coexisting strains is limited by the number of chemicals via which they interact, which in effect create a series of niches, each of which can be occupied by at most one strain (***Chesson, 1990***). Perfect "tradeoffs" are sometimes invoked to enable higher diversity (***Posfai et al., 2017***; ***Beardmore et al., 2011***; ***Erez et al., 2020***) but even tiny differences will destroy this coexistence (***Caetano et al., 2021***).

An alternative to the multi-niche scenario is the *neutral theory* of ecology which postulates that species are similar enough that they are somehow ecologically equivalent, with their population dynamics dominated by stochastic births, deaths, and migration. The predictions of this theory for abundance and spatial distributions are intriguingly similar to some data (***Volkov et al., 2003***; ***Volkov et al., 2007***). However for microbes with short generation times and huge populations without tight bottlenecks, the neutral scenario is not viable: Even if the differences between strains could be neglected over the long times for which they have coexisted, the dynamics from stochastic fluctuations are far too slow. Instead, rapid population dynamics with large changes of relative abundance are often observed (***Ignacio-Espinoza et al., 2020***; ***Martin-Platero et al., 2018***). "Selection", in the broad sense of differential population growth rates, is clearly involved. Thus if a highly diverse population *appears* "neutral" in some respects (including close-to-perfect tradeoffs) this must *emerge* from the complex ecological and evolutionary dynamics; it should not be assumed.

It is often said that pathogens promote diversity (***Bever et al., 2015***; ***Rodriguez-Valera et al., 2009***; ***Thingstad, 2000***; ***Thingstad et al., 2014***). However, there is thus far little understanding of how or under what circumstances ongoing coevolution of hosts and pathogens could cause and sustain extensive coexisting within-species diversity. Understanding this process theoretically is a long-term goal, towards which the present work is a step. To make progress, we need to distill the general phenomenon of fine-scale diversity to its most basic, and endeavor to develop potential scenarios in which evolution, coupled with ecology, might play out. For closely related strains, there is no compelling reason why interactions with siblings should be much stronger than those between distant cousins. Thus, we ask: Without assuming niche-like interactions, perfect tradeoffs, or spatial gradients, can a highly diverse collection of closely related strains stably and robustly coexist? If so, can such a highly diverse "phase" evolve and continue to evolve and diversify? If the evolution is fast, some amount of diversity will always exist (although the common ancestors of the population at any time may be recent and few). Thus we consider the most difficult regime for diversity: when the evolutionary dynamics are much slower than the ecological and spatial population dynamics.

In recent work (***Pearce et al., 2020***), referred to henceforth as PAF, we developed a new scenario for the coexistence of multiple closely related strains that are *assembled* all together into a community, leaving aside the question of their past or future evolution (or even how the community is assembled). In this scenario, we explored a particular key feature of models of many similar strains: the nature of interactions between pairs of strains. It is known that competition for resources in a well-mixed environment leads to positive correlations: if more $A$ individuals are worse for $B$, then more $B$ are worse for $A$. We consider the opposite case where the interactions are anticorrelated. This can arise if the competition is one-on-one: if $A$ beats $B$, then $B$ loses to $A$. A compelling biological motivation for anticorrelated interactions arises from a different scenario: a spectrum of generalist phage strains that prey, with varying efficacies, on a spectrum of bacterial strains. If a particular phage strain, $a$, does better than average against a particular bacterial strain, $b$, then more $b$ individuals are better for $a$, and more $a$ are worse for $b$, leading to anticorrelated interactions. While we are particularly interested in coevolving bacteria-phage diversity, to build up an understanding of the complex eco-evolutionary dynamics, we focus in this paper on simpler models that — as we have shown in PAF — capture many of the key features.

Host-pathogen, and other anticorrelated interactions, give rise to "kill the winner" ecological dynamics (***Thingstad, 2000***). If a strain rises to high abundance, other strains that do well against it will bloom and drive down the abundance of the first, and the process repeats. With many strains that do not have their own niches, this leads to wilder and wilder chaotic variations of abundances, soon driving most types extinct. In PAF we showed that rudimentary spatial structure — a large set of $I$ islands with a low migration rate between all pairs of islands — can maintain much diversity without

a commonly-invoked mainland (***MacArthur and Wilson, 1967***). In this spatial model, many strains go globally extinct, but a large fraction persists indefinitely in a spatiotemporally chaotic phase (hereafter STC). Crucially, the chaotic dynamics desynchronize across the islands allowing strains that go extinct locally to be repopulated from other islands. This mechanism is a manifestation of the "spatial storage effect" (***Chesson, 2000***). On each island, each persistent strain occasionally blooms up to high abundance and subsequently crashes (***Figure 1***). While it is at low abundance, dispersal from blooms on other islands rescues the strain from local extinction until conditions are favorable and its population blooms again, sends out migrants, and crashes. This STC is very robust: strains either go extinct rapidly, or persist globally for times that are exponentially long in the number of islands.

Complementary work (***Roy et al., 2020***; ***Roy et al., 2019***) suggests the generality of the STC beyond anticorrelated pairwise interactions, although in the Lotka-Volterra models studied in these works of Roy et al., the diversity is limited by the strength of self-interactions, which also limits the diversity of stable communities. Indeed, much previous work has focused on ecological dynamics that reach a stable state, where diversity is limited by strength of niche interactions compared to inter-species interactions (***Bunin, 2017***). Here we approach evolutionary dynamics in similar generalized Lotka-Volterra models, but from the opposite starting point: all interactions are of comparable magnitude which makes the effects of self-interactions negligible compared to the effects of the total interactions from all other strains. Then there is no large stable community, and the diversity is maintained by spatial structure and chaotic dynamics.

With anticorrelated interactions, arbitrarily large numbers of strains can coexist in the STC even when spatial mixing — and hence competition — occur on timescales comparable to those of the local ecological dynamics. However, if the strains differ somewhat in their overall growth rate, or other ways that make some *generally* better, these advantages can limit the diversity of the community. A natural assumption is that, having all survived on evolutionary timescales, the persistent strains will be similar enough that such differences are very small. But this assumption — and even more so assumptions of close-to-rigid tradeoffs, (***Posfai et al., 2017***; ***Amicone and Gordo, 2021***; ***Farahpour et al., 2018***) — should surely be questioned. Such features must emerge from the evolution rather than being assumed.

Many theoretical (and some experimental) analyses, have, like our prior work, focused on ecological communities that are assembled without conditioning on their evolutionary histories: a number of species (or strains) is brought together, and the resulting community consists of the species that

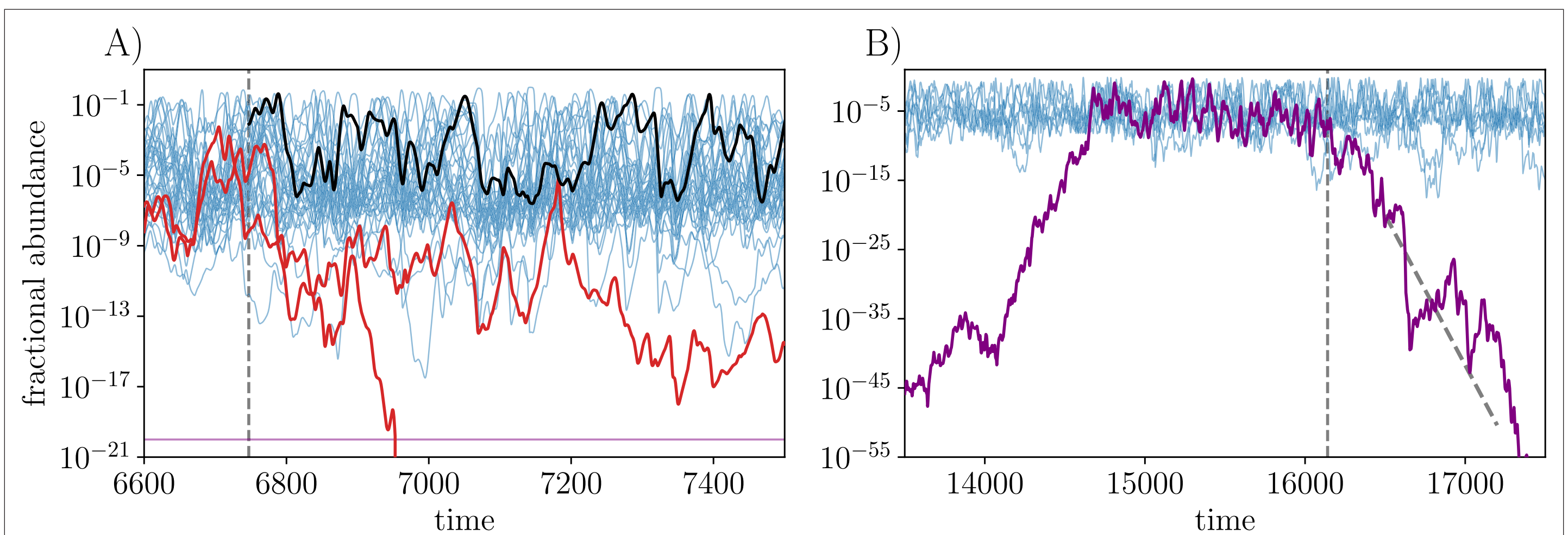


**Figure 1.** Dynamics of strain abundances on a *single island* in the spatiotemporally chaotic state (STC). A subset of strains is plotted. Each persistent strain occasionally blooms up to high abundance and between blooms its abundance is sustained above a migration floor (here $\cong 10^{-7}$) set by migration from other islands, although a few marginal strains fluctuate below this threshold. (**A**) An example of the evolutionary process: at the beginning of an *epoch* (vertical dashed line), a new (here unrelated) strain, (black), is introduced at intermediate abundance. This new strain establishes and persists, causing two (red) strains, which persisted in the previous epoch, to go globally extinct by the end of this epoch. (**B**) Strains that would go extinct on a single island, can persist, and invade from low abundance, due to migration. The purple strain successfully invades. But at the vertical dashed line, migration is turned off for the purple strain only, and it proceeds to go extinct with average exponential decay rate given by its negative *bias*, schematically indicated by the dashed black line (with an extended range of log-abundance shown).

do not go extinct (**Bunin, 2016**; **Bunin, 2017**; **Serván and Allesina, 2021**; **Friedman et al., 2017**; **Goldford et al., 2018**; **Hu et al., 2021**). Although this is an important starting point, it is essential to incorporate evolution to understand how the processes of mutation, inheritance, selection, and extinction could give rise to highly diverse communities.

Previous theoretical work has shown that diverse communities in certain consumer-resource models are destabilized by evolution (**Shoresh et al., 2008**), at odds with the highly diverse continuously evolving microbial populations in nature. Others have focused on eco-evolutionary dynamics when the mutation rate is high enough to sustain diversity: in this case the common ancestor of coexisting strains is recent and extensive diversity over a wide range of genetic divergence does not have time to evolve (**Xue and Goldenfeld, 2017**). Yet others have shown that when niches in phenotype space are assumed, boom-bust dynamics can result in the evolution of higher diversity than occurs in stable equilibrium (**Doebeli et al., 2021**). But overall there is no clear consensus on whether evolution tends to destabilize or to increase diversity in ecologically interacting communities — indeed the answer to this question is likely context-dependent — though observations of the natural world suggest that evolution often results in increased diversity. Here, we investigate evolution starting from a state with spatiotemporally chaotic ecological dynamics as studied in PAF, where niches are absent and the diversity — at least initially — is stabilized by the interplay between endogenous ecological dynamical fluctuations and migration.

We are interested in understanding diversity that has existed for a very broad spectrum of evolutionary timescales, far longer than ecological or spatial mixing timescales. We thus study the extreme limit where the mutation rate is small enough that the ecological and migratory dynamics reach steady state before the introduction of each new strain. This *quasistatic* limit of evolution is the "hardest" for diversification. In addition, and in contrast to some previous work (**Tikhonov and Monasson, 2018**), we assume that global extinctions of a strain are permanent: an extinct organism cannot be resurrected even if conditions later become favorable for it. Focusing on the STC phase, we endeavor to answer: Can a highly diverse STC phase evolve? Under what conditions? Can this phase continue to diversify? Is the diversity stable to *general fitness* mutations that are not artificially constrained by assumptions of tradeoffs? How do statistical properties of the community change during evolution?

## Summary of main results

We first summarize the main results of this work, which concern the behavior of the STC phase under serial invasion of new strains. A parent strain in the community occasionally gives rise to a mutant strain whose properties are correlated with those of its parent with correlations parametrized by $\rho \in [0, 1)$. Many of the behaviors are similar across this range, from independent invaders with $\rho = 0$ to small-effect mutations with $\rho$ close to 1. In all cases, extinctions are irreversible. Key properties of interest are the number of extant strains in the ecosystem, $L$, the number of successful invasions $Z$, and how these evolve with evolutionary time, parametrized by the number of attempted invasions, $T$.

We find that for a wide range of $\rho$ (and expect for any $\rho < 1$) the STC can enter a steadily diversifying state wherein the number of successful invasions, $Z$, and the number of coexisting strains, $L$, both increase linearly with $T$ on average, with only small fluctuations when $L$ is large. Whether diversification occurs, and the rate of the diversification if it does, depends on various parameters, but it is robust over a range of the parameters. If initially the ecosystem has only a modest number of strains, the evolutionary dynamics tend to cause the diversity to crash, after which is it extremely unlikely to transition into the diversifying phase. However, if the initial ecosystem is sufficiently diverse, it is highly likely to diversify further.

We then study the effects of general fitness differences that augment the average growth rate of a strain by an intrinsic amount irrespective of its interactions with the other strains. Focusing on unrelated invaders, we show that a distribution of such general fitness differences (denoted by $s_i$ for strain $i$) can either slow down, prevent, or reverse diversification. For distributions of the $s_i$ whose tails decay faster than exponentially, the diversifying phase still exists, but with the diversification rate gradually slowing down: $L$ increases only as a power of $\log(T)$. If the distribution of the $s_i$ has a broader-than-exponential tail, the diversity decreases and crashes.

The key property of a strain, in terms of which one can understand its behavior, is its *bias*: defined as the rate at which its population would change when at low-abundance and without migration (**Figure 1B**). The crucial effect of migration in the STC is to stabilize many strains with negative bias

which would have gone extinct without migration. Only if its bias is strongly negative will a strain go globally extinct. The bias of strain $i$, denoted by $\xi_i$, has an intrinsic contribution from its general fitness $s_i$, and an extrinsic contribution from the interactions of strain $i$ with all other strains, which includes both an $i$-dependent part determined by the interactions with the other strains in the community, and an $i$-independent part that keeps the total population constant. The extrinsic part of the bias of each strain changes as the community evolves, but its intrinsic part says the same.

Building upon the theoretical understanding of the STC phase, developed in PAF, we first analyze evolution in the simplest case of unrelated invaders ($\rho = 0$) with no general fitnesses ($s_i = 0$). The bias of each strain undergoes a random walk on evolutionary timescales, and we find that for large communities, the number of strains changes at a steady rate. For a range of parameters, this diversification rate is positive, yielding a steadily diversifying phase with the distribution of biases scaling with $1/\sqrt{L}$, as observed in numerics. We then extend our analysis of the changing bias distribution to include the effects of general fitness differences. This yields predictions of how the rate of diversity increase (or decrease) depends on the distribution of the $s_i$, corroborating the behaviors found in simulations.

The distributions of biases and abundances in evolved communities differ subtly from those of the initial communities that were assembled all-at-once from unrelated strains. At early stages of the evolution, most of the close-to-marginal, low-abundance strains are pushed out by the perturbations caused by the invading strains. This extinction process causes the shape of the abundance distributions of assembled and evolved communities to differ at low abundances. Later, in the steadily diversifying state, the numbers of extinctions caused by each invader has a roughly exponential distribution, which is consistent with our theoretical expectations. In contrast to the qualitative (albeit modest) changes in abundance distributions, we find that evolution has only a small effect on the statistics of the interactions between strains.

## Outline

The structure of this paper is as follows: 'Models' introduces the main model and its relation to previous work. 'Results' describes the phenomenology of an evolving community in the STC phase, studying the effects of correlated mutants, interaction statistics, and general fitness differences on the ecological diversification. Then 'Analysis' develops the theory and analysis that are needed to understand these phenomena. Building upon the dynamical mean field theory developed in PAF, we present an approximate framework, and more general scaling arguments, for understanding the evolutionary dynamics, and compare the predictions with simulations. Finally 'Discussion' raises additional questions and discusses possible extensions. Many of the details and further analyses are relegated to appendices.

# Models

We here define the model, discussing the various roles played by local deterministic population dynamics, demographic stochasticity, spatial migration and evolutionary dynamics. Our notation is summarized in ***Table 1***.

## Ecological interactions

We first consider an assembled community of $K$ unrelated strains, labelled by $i = 1, 2, \ldots, K$, with all possible pairwise interactions between them. A paradigmatic model for the ecological dynamics of the strain populations $\{n_i\}$ is the generalized Lotka-Volterra model (***Goel et al., 1971***), with each strain $i$ having an intrinsic growth rate which is modulated by its interactions with all the other strains. These interactions are conveniently represented in a matrix $W$ where $W_{ij}$ describes the effect of strain $j$ on the growth rate of strain $i$. Since we are interested in closely related strains for which all interactions are similar, the total population will be roughly fixed at some $N$ by the balance between the effects of positive intrinsic growth rate and negative competitive interactions. It is convenient to replace these large terms by a Lagrange multiplier $\Upsilon(t)$ that fixes the total population to $\sum_i n_i = N$, and work with fractional abundances, $\nu_i \equiv n_i/N$. This parameterization yields what are known as "replicator equations" (***Chawanya and Tokita, 2002***; ***Yoshino et al., 2008***; ***Tokita, 2004***).

Variations in intrinsic growth rates and net interactions on a strain can be combined to yield *general fitness differences*, $\{s_i\}$, between the strains. We parameterize the residual variations in interactions

**Table 1.** Definitions of commonly used quantities.

| | |
|---|---|
| **Ecology** | |
| STC | Spatiotemporally chaotic state |
| $K$ | Number of strains put into the initial assembled community |
| $V$ | Matrix of pairwise strain interactions |
| $\gamma$ | Symmetry parameter of the interaction matrix; $\mathrm{E}[V_{ij}V_{ji}] = \delta_{ij} + \gamma$ |
| $I$ | Number of islands |
| $N$ | Population size on each island, fixed to be constant |
| $\nu_{i,\alpha}$ | Fractional abundance of strain $i$ on island $\alpha$ |
| $\bar{\nu}_i$ | Time (or space) average of strain $i$ abundance |
| $\nu_{floor}$ | Migration floor $\sim m\bar{\nu}\sqrt{L/\mathcal{M}}$: ~ lower range of local abundances |
| $\Upsilon_\alpha$ | Lagrange multiplier maintaining $\sum_i \nu_{i,\alpha} = 1$; $\Upsilon_\alpha = \sum_i \nu_{i,\alpha}(s_i + \sum_j V_{ij}\nu_{j,\alpha})$ |
| $s_i$ | General fitness of strain $i$ |
| $\mathcal{P}(s)$ | Probability distribution of the $s_i$ |
| $\Sigma$ | Characteristic scale of the $s$ distribution |
| $\psi$ | Exponent characterizing tail of of $\mathcal{P}(s) \sim \exp[-(s/\Sigma)^\psi/\psi]$ |
| $m$ | Migration rate between islands |
| $\mathcal{M}$ | Range of fluctuations in $\log \nu$; $\mathcal{M} = \log(1/m)$ |
| **Evolution** | |
| $\rho$ | Correlation between parent's and mutant's interactions with other strains |
| $T$ | Evolutionary time in epochs, equal to number of attempted invasions |
| $Z$ | Number of successful invasions |
| $L$ | Number of extant strains at any point in the evolution |
| $L_0$ | Number of strains surviving in initial assembled community |
| $U$ | Average diversification rate; $U = \langle dL/dT \rangle$ |
| $\hat{s}$ | Mean general fitness of extant strains; $\hat{s} = \sum_j \nu_j s_j$ |
| $\widehat{\Sigma}$ | The scale of $\mathcal{P}(s)$ of extant strains; $\widehat{\Sigma} = -[\frac{d}{ds}\log \mathcal{P}(s)]^{-1}\vert_{s=\hat{s}}$ |
| **Analysis** | |
| $\xi_i$ | Bias of strain $i$, its growth rate at low abundance without migration; $\xi_i = \zeta_i + s_i - \bar{\Upsilon}$, scales as $1/\sqrt{L}$ |
| $\mathcal{N}(\xi)$ | Mean abundance of a strain as a function of its bias |
| $\xi_c$ | Critical bias (negative in the STC) below which strains go extinct, scales as $1/\sqrt{L}$ |
| $\zeta_i$ | Mean drive on strain $i$ by other strains in its absence; $\zeta_i = \sum_j V_{ij}\bar{\nu}_{j\backslash i}$ |
| $\mathcal{L}$ | Effective number of extant strains; $\mathcal{L} = 1/\sum_j \bar{\nu}_j^2$; scales with $L$ |
| $\chi_i$ | Static response of strain $i$ to perturbations; $\chi_i = \frac{d\bar{\nu}_i}{d\xi_i}$ |
| $X$ | Total static response; $X = \sum_i \chi_i$ |
| $\Xi$ | Fragility of the community to perturbations; $\Xi = \sum_j \chi_j^2/(1 - \sum_j \chi_j^2)$ |

among the strains (after subtracting off $\Upsilon$ and $s_i$) by $V_{ij}$. Since the $V_{ij}$ and $s_i$ are sums and differences of similar magnitude terms, it is natural to approximate them as random variables with the hope that the model will yield behaviors that are robust to specific choices of their statistics: testing this assumption is one of the goals of this paper. For simplicity, we choose $\mathrm{E}[V_{ij}] = 0$, and $\mathrm{E}[V_{ij}^2] = 1$ for $i \neq j$ — setting the overall ecological timescales — and choose the covariances to be zero except for, importantly, correlations between how $i$ and $j$ affect each other, defining $\mathrm{E}[V_{ij}V_{ji}] = \gamma$. For convenience, we choose $\mathrm{E}[V_{ii}^2] = 1 + \gamma$ but this choice has negligible effect in large communities.

The parameter $\gamma$ controls whether the interactions are mainly competitive ($\gamma > 0$) or host-pathogen-like ($\gamma < 0$), the latter being the focus of this work. We have shown in PAF that random interaction matrices with such anticorrelations behave very similarly to host-pathogen models with the appropriate block sub-matrix structure, as discussed further in 'Bacteria-phage interactions and coevolution'.

## Ecological dynamics

We study the simplest model with spatial structure: a large number, $I$, of identical islands (or demes) with interactions only within each island and migration between all pairs of islands. With Greek indices labeling islands, the dynamics of the abundances obey

$$\frac{d\nu_{i,\alpha}}{dt} = \nu_{i,\alpha}\left(s_i + \sum_{j=1}^{K} V_{ij}\nu_{j,\alpha} - \Upsilon_\alpha(t)\right) + \sum_{\beta=1}^{I}(m_{\alpha\beta}\nu_{i,\beta} - m_{\beta\alpha}\nu_{i,\alpha}). \quad (1)$$

with $m_{\alpha\beta}$ the migration rate (per individual) from island $\beta$ to island $\alpha$ and the local Lagrange multiplier, $\Upsilon_\alpha = \sum_i \nu_{i,\alpha}(s_i + \sum_j V_{ij}\nu_{j,\alpha})$, keeping the total population on each island fixed at $N$, (i.e. $\sum_i \nu_{i,\alpha} = 1$ for each island). Here, we focus on the spatial mean field limit in which the migration rate is the same, given by $m/I$, between every pair of islands. The total migration of strain $i$ into and out of island $\alpha$ is then simply $m(\bar{\nu}_i - \nu_{i,\alpha})$ with $\bar{\nu}_i(t)$ the average of $\nu_i(t)$ across islands. As the number of islands becomes large, in steady state, each $\bar{\nu}_i(t)$ becomes constant in time — some being zero corresponding to global extinction. In the STC, the dynamics are asynchronous across islands and ergodicity implies the spatial average, $\bar{\nu}_i$, is equal to the time-averaged abundance of strain $i$ on a single island; this is a crucial self-consistency condition. The magnitude, $m$ of the migration rate, is also of fundamental importance. If $m$ is too small the migration is too rare to repopulate islands after local extinctions. If $m$ is too large and the local dynamics is chaotic, the chaos will synchronize across the islands and the total population of each strain will fluctuate wildly, rapidly driving most strains extinct. We will focus on the wide intermediate $m$ regime, which spans several orders of magnitude when $K$ is large [PAF].

With large populations on each island, demographic fluctuations have little effect on the dynamics. Even when the local population of a strain is small, if it has positive growth rate, fluctuations will not matter much, while if it has negative growth rate it will go deterministically extinct which occurs when the fractional abundance drops below the extinction threshold of $1/N$ indicated in ***Figure 1A*** by the horizontal purple line. The value of the extinction threshold does not much affect the behavior as long as it is much below the lower limit of the abundance caused by migration — which we term the *migration floor*: $\nu_{floor} \sim m\bar{\nu}$. For strains near local extinction (when the fractional abundance is close to $1/N$) demographic fluctuations are potentially important. But with $Nm\bar{\nu}$ very large, local extinctions for viable strains will be rare: thus we model the population dynamics as fully deterministic. If the fractional abundance on an island drops below $1/N$, it is set equal to zero. Global extinction occurs when a strain's bias becomes too negative, which results in it going below the extinction threshold everywhere. The choice of $N$ does not matter much as long as $mN \gg 1$, to which we restrict consideration. Related details of numerical implementation are discussed in 'Appendix 2'. In 'Spatial structure and dynamics', we comment on the effects of local extinctions in the context of real spatial dynamics.

The key properties of the STC phase [PAF] are chaotic coexistence of strains, desynchronized across islands, with the local abundances fluctuating over a range in $\log\nu$ of $\mathcal{M} \equiv \log(1/m)$, which is quite wide for the typical $m = 10^{-5}$ that we use in simulations. Some strains go globally extinct but each persistent strain on each island occasionally has a bloom up to high abundance $\nu \sim \mathcal{M}/L$. These localized blooms are crucial for stabilizing a strain, as they dominate the migration to other islands needed to recover from local extinctions or near-extinctions. On a single island, at any given moment, $\mathcal{O}(L/\mathcal{M})$ strains are at high abundance. A snapshot of the abundances on a single island shows the

strains distributed roughly uniformly in $\log(\nu)$ down to the migration floor, only occasionally fluctuating substantially lower (**Figure 1**).

## Evolutionary dynamics

The evolutionary process we model is much slower than the ecological and migratory dynamics. Simulations are divided into long *epochs*, with new strains added only at the end of an epoch. The epochs are chosen long enough that the ecological and migratory dynamics have reached a steady state, with some fraction of the strains having gone *permanently extinct* globally, leaving $L$ persistent strains. A single new strain is then introduced and the process repeated.

The new strain, generically labeled $A$, is parameterized by its interactions with all other strains in the community, given by $V_{Aj}$ and $V_{jA}$, and its general fitness, $s_A$. In the simplest case, a new strain is unrelated to extant (or extinct) strains. More generally, mutant strains, labelled $M$, are characterized by their degree of correlation, $\rho \in [0,1)$, with a parent strain $P$ chosen from the existing community with probability proportional to its mean abundance $\bar{\nu}_P$. These correlations are realized such that $\mathrm{Corr}[V_{jM}, V_{jP}] = \mathrm{Corr}[V_{Mj}, V_{Pj}] = \rho$ for $j \neq M, P$. The detailed choices for $j = M, P$ are given in 'Appendix 2'. The general fitness, $s_M$, can also be correlated with $s_P$. Unrelated invaders are equivalent to $\rho = 0$ and hence have no parent.

The actual process of invasion from low abundance on one island is complicated, and often leads to failure. To avoid a proliferation of such failed invasions, we instead assess whether the invader *could* successfully invade and persist if it were lucky initially. To do this, we set the mutant's abundance to $1/L$ on all the islands at the same time (and proportionately decrease the abundances of the other strains to maintain $\sum_i \nu_{i,\alpha} = 1$).

## Timescales

There are multiple timescales involved in the dynamics: these are discussed more fully in 'Appendix 1'. The basic timescale for *differential growth or decay* of strains is set by the magnitude of the interactions and the number, $L$, of extant strains. The extant strains have average abundances of order $1/L$, so the average total interaction on a strain, $i$, is the sum of $L$ random terms, each of order $V/L$ for typical interaction strength $V$. With the $V$ having variance unity, the average net interactions of other strains on strain $i$ is roughly its *mean drive*, $\zeta_i$, which is of order $1/\sqrt{L}$, implying that the timescale for systematic population growth or decay is of order $\sqrt{L}$. The mean drive is defined more precisely in Results. When there are general fitness differences, $s_i$, these also contribute to variations in average growth rates. The variations in the $s_i$ within the community have substantial effects over a time $\sim 1/\sigma_s$ with $\sigma_s$ roughly the width of the $s_i$ distribution of the *extant* strains. Together, the mean drive and general fitness of a strain determine its crucial property: the bias $\xi_i = \zeta_i + s_i - \langle \Upsilon \rangle$, with angular brackets denoting a time average. As introduced earlier, the bias of strain $i$ is its average growth rate at low abundance in the absence of migration (**Figure 1B**). As we shall see, the size of the community is limited by the condition that the inter-strain variation in general fitness is no larger than variation in average drive from interactions. This means that the biases are of order $1/\sqrt{L}$.

The local population of each strain undergoes wild fluctuations over a logarithmic range $\mathcal{M} = \log(1/m)$ which is quite large. During blooms, the instantaneous growth and decay rates of local populations are substantially larger than the systematic biases ('Appendix 1') and change rapidly from growth to decay as seen in **Figure 1**. The time for abundances to fluctuate from large to small — the *duration of blooms* — is of order $\mathcal{M}\sqrt{L}$ with systematic and fluctuation contributions comparable.

An important timescale for studying slow evolution is the time to reach the STC steady state: the *ecological relaxation time*. This is determined by the strains that are just barely going extinct and is of order $\mathcal{M}L$ for an evolved community, as discussed in 'Continual assembly and diversification'. We have chosen the evolutionary timescale to be much longer than all the other important timescales. Thus, each epoch between the addition of invaders is chosen to be several times the ecological relaxation time, typically $3\mathcal{M}L$, and we show in 'Appendix 2' that increasing this epoch length by a factor of 10 makes little difference in the diversification dynamics.

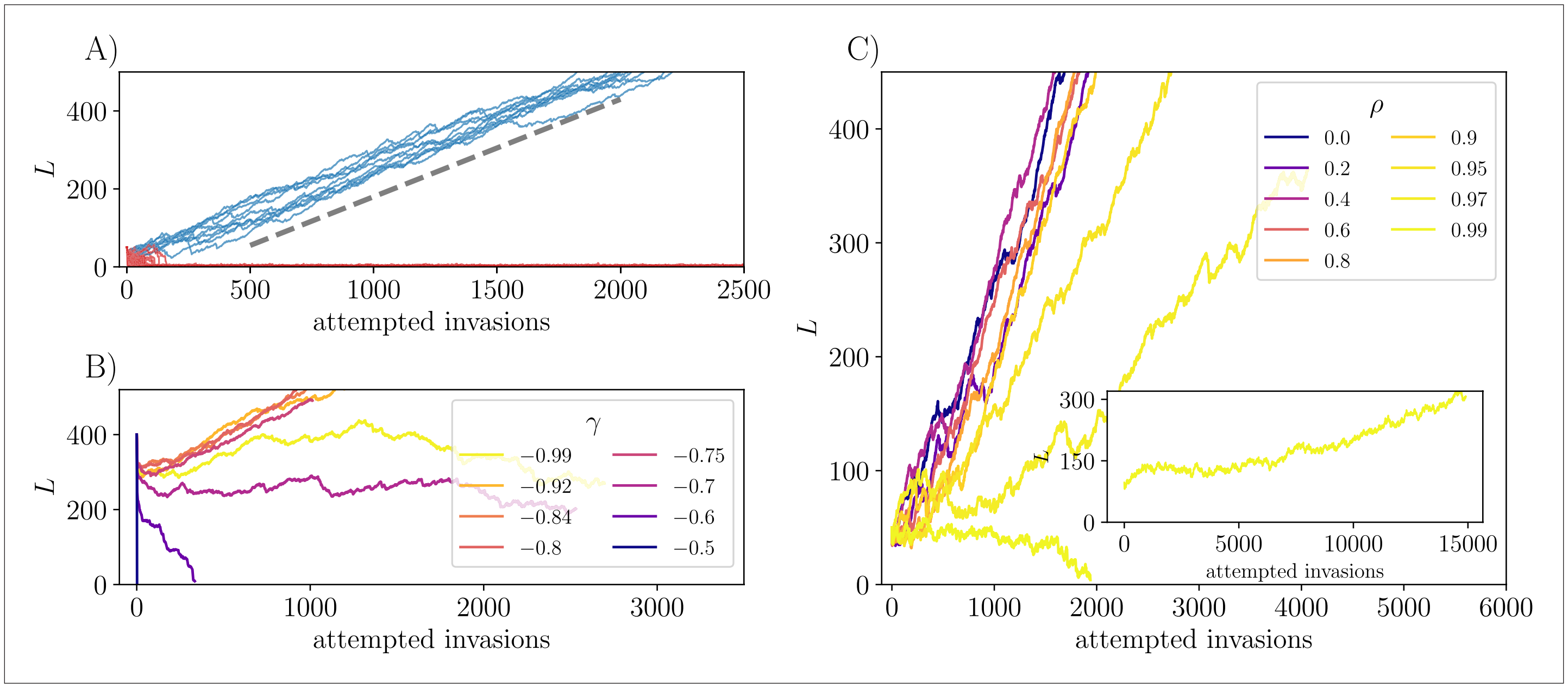


**Figure 2.** Evolution of number of strains without general fitness differences. (**A**) With $\gamma = -0.8$, $m = 10^{-5}$, and initial number of strains $K = 50$, under serial invasion of *unrelated strains* most initial communities (red) crash and fail to recover, while others (about 20%, blue) continually diversify. Once the communities are large, around 80% of further invasions are successful and the mean number of extinctions per successful invasion is $\cong 0.7$ (***Appendix 3—figure 1***) so that on average the number of strains in the community grows linearly with rate $U \cong 0.25$ per invasion attempt (dashed line). (**B**) Whether diversification occurs, and its rate if it does, depends on the symmetry parameter, $\gamma$, as seen here with $K = 400$ and $\rho = 0$. For $\gamma$ close to $-0.7$, evolution reduces the diversity. For less negative $\gamma$, the STC breaks down and the diversity crashes immediately. For more negative $\gamma$, steady diversification occurs, fastest here with $\gamma \approx -0.8$, though again slowing down as $\gamma \to -1$. (**C**) Evolving communities under successive introduction of *mutants*, each with correlation $\rho$ with its parent ($\gamma = -0.8$). The diversification rate varies nonmonotonically with $\rho$, with fastest diversification for $\rho \approx 0.8$. There is a significant slowdown for $\rho$ close to unity. Here $K = 50$ and trajectories are shown conditional on not crashing, except for $\rho = 0.99$, which renders the evolving community very susceptible to crashing from $K = 50$. However the inset shows that even with $\rho = 0.99$, it is possible to reach a diversifying regime starting from $K = 100$.

## Results

The STC is robust, with strains persisting for times that are exponentially long in the number of islands. However, evolutionary perturbations caused by an invading strain can drive strains deterministically extinct. This process can be understood in terms of the biases of the strains.

The bias of a strain, is determined by the community *in its absence*. It can be written precisely as $\xi_i = s_i + \zeta_i - \langle \Upsilon_{\backslash i} \rangle$ with the mean drive $\zeta_i = V_{ij} \langle \nu_{j\backslash i} \rangle$, where the notation $\langle \nu_{j\backslash i} \rangle$ and $\langle \Upsilon_{\backslash i} \rangle$ denote the time-averaged abundance of strain $j$ and average Lagrange multiplier in the absence of strain $i$. The $\Upsilon$ is not much changed by the absence of the one strain, but the abundances of the other strains are affected in small but collectively essential ways by whether or not strain $i$ is present, as discussed in 'Dynamical mean field theory'. This negative feedback — proportional to $\gamma$ — is what stabilizes strains whose abundance would otherwise keep growing.

With many strains participating in the chaos on each island, and desynchronization across islands, we expect the chaos to be ergodic, so that the $i$ time averages and spatial averages (across islands) of all quantities are equal in the STC steady state. Therefore we will use spatial average notation $\bar{\nu}$ instead of time average notation $\langle \nu \rangle$, except when conceptually the latter is clearer. In practice, the $I = 40$ islands used in numerics are enough that the persistence times of almost all surviving strains are very long and averages across islands of the more important quantities do not fluctuate much in steady state.

A crucial feature of the STC phase is that strains with somewhat negative bias can persist due to migration between desynchronized islands (***Figure 1A***). This stabilization is enabled by a nontrivial feature of the STC phase: during a bloom, the systematic changes in $\log \nu_i$ caused by the bias are comparable to the cumulative stochastic growth and decay caused by the endogenous fluctuations

— the zigs and zags in the dynamics of $\log \nu_i$ (***Figure 1***). This is a manifestation of the system "self-tuning" to a special self-consistently chaotic state [PAF].

Despite the possibility of rescue from extinction via rare blooms, there is a critical negative bias, $\xi_c$, (sharp for large $L$ and large $I$) below which strains no longer persist even as $I \to \infty$. For strains with $\xi$ below $\xi_c$ (which depends on the parameters and the number of strains), blooms up to high abundance are not frequent enough to repopulate local extinctions and deterministic global extinction ensues. For large $I$ and large $L$, strains with $\xi < \xi_c$ go extinct, while strains with $\xi > \xi_c$ persist indefinitely. Finite $L$ and finite $I$ effects, together with the finite time for each epoch, will round out the sharpness of the borderline between persistent and extinct. However the marginal strains involved have little effect on others and whether or not they persist does not much matter for the current epoch: we are interested in *deterministic* extinction caused by the introduction of new strains. Therefore, we need to study how the *distribution* of biases in the ecosystem evolves.

## Continual assembly and diversification

The evolutionary process we study starts from an assembled collection of $K$ unrelated strains. After the ecological and migratory dynamics have reached steady state, some of the strains will persist: we call the size of this initial persistent community $L_0$. The $K - L_0$ strains that have gone globally extinct are permanently removed.

When a new strain is introduced into the ecosystem, if it successfully invades it perturbs the biases of the extant strains, and can trigger extinctions of some of them by shifting their bias below $\xi_c$ (***Figure 1A***) by an amount of order $1/L$. We study the slowly evolving regime in which the ecosystem dynamics reach steady state between each introduction of a new strain — this takes time of order $\mathcal{M}L$. The number of persistent strains and number of successful invasions as a function of the number of attempted invasions, $L(T)$ and $Z(T)$ respectively, are of fundamental interest.

We first describe the evolutionary dynamics when the general fitness differences between the strains can be neglected. For $\gamma = -0.8$ and *unrelated invaders* ($\rho = 0$), multiple simulation runs starting with different sets of $K = 50$ initial strains reveal that around one fifth of the replicates enter a steadily diversifying regime in which $L$ increases roughly linearly with the number of attempted invasions, at a rate of around 0.25 per attempt. The remaining replicates crash down to only a few persistent strains. Subsequent invasions can cause $L(T)$ to increase somewhat, but it quickly crashes back down and the community does not steadily diversify (***Figure 2A***). The low diversity regime that occurs after a crash (or with a very small initial community) is discussed further in 'Appendix 3'.

The observations in ***Figure 2*** illustrate one of the crucial findings of this work: spatiotemporally chaotic ecological dynamics can allow — but do not guarantee — gradual strain-level diversification up to arbitrarily high number of strains. The behavior depends on the symmetry parameter $\gamma$, which must be substantially negative for the STC to exist. ***Figure 2B*** shows that the average rate of diversification, $U \equiv \langle dL/dT \rangle$, is nonmonotonic in $\gamma$, with slow diversification close to $\gamma = -1$ (its lower limit). As $\gamma$ becomes less negative the rate of diversification increases at first. However for $\gamma$ even less negative, the STC still supports chaotic coexistence of many strains (since $L_0$ is still large), but the diversity decreases under evolutionary dynamics. The community diversifies most rapidly for $\gamma \approx -0.8$. As we are interested in what can happen with various other additional features, we chose $\gamma = -0.8$ and $m = 10^{-5}$ for all further simulations as shorter runs are needed near these values. We expect that the qualitative conclusions will be similar for a range of $\gamma$ and $m$ around these.

## Evolution with correlated mutants

In addition to studying independent invasions, we study evolution via mutations of existing strains. At the start of each epoch, a parent to mutate is chosen with probability proportional to its mean abundance. The interactions of the mutant with other strains are drawn from the same marginal distribution as the original interactions, but with correlation $\rho$ with the interactions of the parent (Evolutionary dynamics). The direct interactions between the parent and mutant have to be chosen separately as specified in 'Appendix 2' but, as they only account for a small fraction of the total abundance in diverse communities, the specific choice is not important. As a function of $\rho$ (with $\gamma = -0.8$), the rate of diversification is nonmonotonic being fastest for $\rho \approx 0.8$, and only weakly varying for smaller $\rho$ (***Figure 2C***). As $\rho$ nears 1, the mutant and parent are more similar, and it becomes harder for them to coexist, since any difference between them is likely to result in a systematic change in their relative

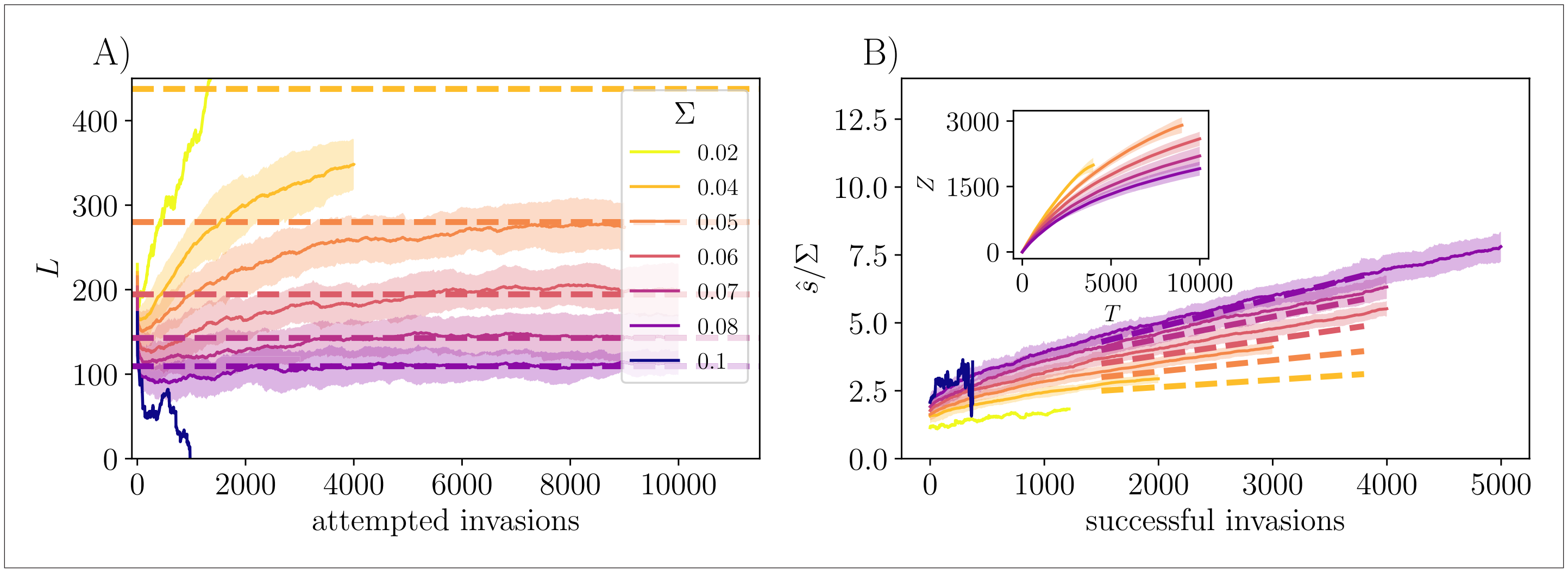


**Figure 3.** Effects of exponentially distributed general fitnesses, $s_i$, on community evolution. Here $K = 250$ initial strains and $\mathcal{P}(s) \sim e^{-s/\Sigma}$, with various $\Sigma$. (**A**) Community size, $L$, as a function of evolutionary time $T$ (the number of attempted invasions) approaching an evolutionary steady state with $L \sim \Sigma^{-2}$ at long times. The dashed lines indicate $0.7 \times \Sigma^{-2}$, which captures the predicted scaling between the steady-state $L$ and $\Sigma$. Data are averaged over 50 runs, conditional on not crashing, with the shaded region showing the standard error. Only single runs are shown for $\Sigma = 0.1$ and 0.02, the former caused crashing and the latter saturation beyond the range of the simulations. For a narrow distribution of the general fitnesses ($L_0 \ll \Sigma^{-2}$), $L$ increases linearly before saturating. For larger $\Sigma$ with many initial strains, immediate extinctions drive $L$ down to $L_0 \sim 1/\Sigma^2$ ('Appendix 6'). (**B**) The average fitness of the community, $\hat{s} = \sum_i \bar{\nu}_i s_i$, grows linearly in the number of *successful* invasions, $Z$, with $d\hat{s}/dZ \sim \Sigma^3$: the dashed lines have slope $\Sigma^2/6$, indicating this expected scaling relationship. The inset shows the rate of successful invasions slowing down with attempted invasions, as it gets harder to draw a general fitness that is sufficiently far into the tail of $\mathcal{P}(s)$.

abundance, eventually driving one of them to extinction (see 'Appendix 9'). Since $L$ can only increase when both the mutant and parent coexist, increasing $\rho$ slows the rate of diversification but $L(T)$ still increases linearly.

This observation implies that, for a large range of $\rho$, despite not enforcing any precise constraints or perfect tradeoffs, strains that would outcompete all extant members of the community are too rare to emerge and reduce diversity. Therefore, we conjecture that a continually diversifying phase exists even for $\rho$ arbitrarily close to 1.

## Evolutionary dynamics with general fitness differences

So far we have observed that when mutants or invaders differ only by their interactions with each other, there is robust and rapid diversification, provided that the initial diversity in the STC phase is high enough. Now we include general fitness differences $s_i$ between strains and show how these affect the evolutionary dynamics.

### Exponential distribution of $s_i$

We first analyze the simplest case: exponentially distributed selective differences with scale $\Sigma$ and probability density $\mathcal{P}(s) = \frac{1}{\Sigma} e^{-s/\Sigma}\Theta(s)$ where $\Theta(s)$ is the Heaviside step function. We consider the evolutionary dynamics in the case of unrelated invaders. As a community evolves, the distribution of the $s_i$ of the community will change, and we are particularly interested in the dynamics of the *population-weighted mean* $\hat{s} = \sum_j \bar{\nu}_j s_j$.

The width of the distribution of $s_i$, here $\Sigma$, plays a controlling role. If one strain has a substantially higher growth rate than all other strains, it will outcompete them, driving many extinct. Thus a broad distribution of $s_i$ is likely inconsistent with a diverse community. We therefore focus on narrow distributions: that is small $\Sigma$. The typical magnitude of the drive of a strain is of order $1/\sqrt{L}$; therefore, when $\Sigma$ is much smaller than this, it will not matter much. On the other hand, if $\Sigma$ were to be much larger than $1/\sqrt{L}$, the differences in the $s_i$ would dominate over the drives and only the strains with the highest and quite similar $s_i$ would survive. Thus $L \gg 1/\Sigma^2$ seems inconsistent. Even for the initial community with $L_0$ strains, we expect that $L_0$ cannot be larger than order $1/\Sigma^2$ (although it can be much smaller if

$K \ll 1/\Sigma^2$). Indeed, in 'Appendix 6' we show that $\Sigma$ sets the initial persistent community size, $L_0$, in a particular limit of the model where $\gamma = -1$.

A natural conjecture is that for small $\Sigma$ with an exponential distribution, steady diversification can occur until the breadth of the $s$ distribution becomes important — when $L \sim 1/\Sigma^2$ — and after that $L$ will saturate, as seen in ***Figure 3***. Thereafter, $\hat{s}$ will grow and invasions of unrelated strains are less and less likely to be successful — in 'Diversification rate with a distribution of general fitnesses' we show that the number of successful invasions $Z$, increases as $\log(T)$. However, successful invasions will on average drive exactly one other strain extinct. ***Figure 3*** illustrates this behavior, including the large initial drop from $K$ to $L_0$ when $K \gg 1/\Sigma^2$, the $\Sigma$-dependence of the steady-state $L$, and the linear increase of $\hat{s}$ with number of successful invasions.

## More general distributions of $s_i$

Building on an understanding of the case of exponentially distributed $s$, we consider a more general family of distributions, motivated by the expectation that the tail of the $s$ distribution is particularly important for evolution:

$$\mathcal{P}(s) \sim \exp\left[-\frac{1}{\psi}\left(\frac{s}{\Sigma}\right)^{\psi}\right]\Theta(s). \tag{2}$$

Anomalously small $s$ strains are very unlikely to successfully invade, so the sharp cutoff at the lower end does not matter. Although we consider only positive $s_i$, all $s_i$ can be shifted by a constant without affecting the dynamics because this constant gets absorbed into $\Upsilon(t)$.

As we will analyze in 'Diversification rate with a distribution of general fitnesses', the evolution of diversity is seen to depend crucially on $\psi$. If the tail of the $s$ distribution falls off faster than a simple exponential, $\psi > 1$, the community continually diversifies, albeit more and more slowly with $L(T)$ increasing only as a power of $\log(T)$. Concomitantly, the mean $s$ of the community, $\hat{s}$, gradually increases. But if the $s$ distribution decays slower than a simple exponential, $\psi < 1$, the diversity decreases (after an initial increase if $\Sigma$ is sufficiently small) and eventually crashes. In the marginal case of a simple exponential tail, $\psi = 1$, as seen above, the diversity saturates and fluctuates around a steady state value while the mean $\hat{s}$ increases linearly with the number of successful invasions. Therefore we conclude that for the evolutionary process in our models to *continually* generate higher diversity, the distribution of general fitnesses must decay sufficiently rapidly. Such rapid decrease of the distribution of available beneficial mutants with ongoing evolution roughly corresponds to "diminishing-returns epistasis".

## Mutants with correlated general fitnesses

What happens if — as one would expect — the invaders are mutants with general fitnesses correlated with their parents? With such mutants, it is possible for the evolution to proceed with less slowing down than for independent invaders. Indeed, with an exponential distribution of the $s_i$ ($\psi = 1$) analysis suggests that evolution proceeds at a constant rate, with both $Z(T)$ and $\hat{s}(T)$ growing linearly as in the absence of general fitness differences, but $L$ still saturating. In 'Appendix 7' simulation results are shown for an exponential $\mathcal{P}(s)$ with correlations in both interactions and general fitnesses. The saturating value of $L$ is quite similar in both the correlated and uncorrelated cases. However, for correlated mutants $\hat{s}(T)$ pushes rapidly into the exponential tail — and surely toward the breakdown of the assumption of the existence of such large $s$ mutations.

For $\mathcal{P}(s)$ that decays faster than exponentially, $\psi > 1$, the behavior is more complicated. However, as discussed in 'Appendix 7', even with correlated mutants, evolution will eventually become very slow, as for uncorrelated invaders. With mutants instead of unrelated invaders, this is a direct example of the effects of diminishing-returns epistasis.

# Analysis

In this section we develop an approximate analytical theory of the evolutionary dynamics and provide heuristic understanding for most of the observed phenomena described above. The underlying basis is the dynamical mean field theory (DMFT) of the STC phase developed in PAF. This takes advantage

of the large number of strains and the large number of islands in order to simplify the descriptions and analyses of the behaviors.

The natural quantities that characterize strains in the DMFT are their biases, $\{\xi_i\}$, and how these set their mean abundances, $\{\bar{\nu}_i\}$. For a large randomly assembled or evolved community the mean abundances will be a function of the biases: $\bar{\nu}_i \approx \mathcal{N}(\xi_i)$, with the function $\mathcal{N}$ depending on the parameters, evolutionary history, and feedback from other strains. As shown in PAF, $\mathcal{N}(\xi)$ is linear for large argument and decays as $\xi$ becomes negative, vanishing at $\xi = \xi_c < 0$.

The relation between $\bar{\nu}_i$, $\xi_i$ and the total average force on strain $i$ — from both direct effects and feedback — enables one to estimate the bias from the simulations ('Appendix 2'). Armed with the DMFT description, we can understand how the biases of extant strains change over the course of invasions. We do this in detail for the simplest case — invasions of unrelated strains without general fitness differences — and show that the evolution causes $L$ to change linearly with the number of invasions — decreasing or increasing depending on the parameters. A simple approximation to the evolution of the biases enables semi-quantitative results. Of particular importance is the result that in evolving communities, the density of biases vanishes linearly as $\xi \to \xi_c$. A corollary of this, as discussed in 'Distribution of biases and number of extinctions', is that the number of extinctions per successful invasion is roughly exponentially distributed. We then analyze the effects of general fitness differences, using our understanding of the exponential $\mathcal{P}(s)$ to generalize to other shapes of the tail of $\mathcal{P}(s)$, parametrized by $\psi$ (**Equation 2**), and showing how the steepness of the tail affects the rate at which $L$ increases or decreases.

## Dynamical mean field theory

The DMFT approximation, which is exact in the limit of a large number of strains with random interactions between them, replaces the full statistical dynamics by the stochastic effects of the others on one chosen strain, with the statistical properties then determined self-consistently from the properties of the distributions over the strains. This approach was first used in the physics of disordered systems such as spin glasses (**Sompolinsky and Zippelius, 1982**), but has been applied to ecological dynamics in a number of subsequent works (**Diederich and Opper, 1989**; **Opper and Diederich, 1992**; **Galla, 2006**; **Roy et al., 2019**; **Rieger, 1989**; **Yoshino et al., 2008**).

When strain $i$ has very low abundance, its effects on the others are very small and the forces of the others on it, $\sum_j V_{ij}\nu_j$, are comprised of roughly independent random variables and thus act like gaussian noise with correlations $C(t,t') = \sum_j \nu_j(t)\nu_j(t')$. However, when it rises to substantial abundance, it will weakly affect the other strains. Because of the correlation $\gamma$ between $V_{ij}$ and $V_{ji}$, these feedback terms add coherently, resulting in a contribution to growth rate of $i$ of form $\gamma \int_0^t R(t,t')\nu_i(t')dt'$, with $R$ a response function determined by feedback from the total impact on the community of the strain's own past history (see 'Appendix 5'). The DMFT allows one to recast the generalized Lotka Volterra equations as an effective single-strain problem, with self consistency conditions on the bias correlations and response function.

With the dynamical mean field understanding of an assembled STC phase in hand, we can proceed to describe the evolutionary process in terms of the distributions of properties of the extant and newly invading strains — in particular their biases and consequent mean abundances. The distribution of biases is perturbed by the introduction of new strains and this can push some of the extant biases below $\xi_c$, which itself depends on the bias distribution as modified by prior evolution, and on the number of extant strains, $L$. It is convenient to define an effective community size $\mathcal{L} = 1/\sum_i \bar{\nu}_i^2$ which controls the variances of the mean-drive part of the bias; $\mathcal{L}$ scales with the actual $L$ but discounts strains that are close to extinction.

## Evolution without general fitness differences

We first analyze invasion of unrelated strains without general fitness differences. The bias of an attempted invader, labelled $A$, is given by $\xi_A = \sum_j V_{Aj}\bar{\nu}_j - \bar{\Upsilon}$, in terms of its interactions, $\{V_{Aj}\}$ with the extant strains: for unrelated invaders, this is gaussian distributed with mean $-\bar{\Upsilon}$ and standard deviation of $1/\sqrt{\mathcal{L}}$ *independent* of correlations among the extant strains, though correlations in the existing community will affect the $\mathcal{N}(\xi)$ and hence $\mathcal{L}$.

Strain $A$ can successfully invade the community and persist if $\xi_A > \xi_c$. The probability of successful invasion is thus $\Phi[(\xi_c + \bar{\Upsilon})\sqrt{\mathcal{L}}]$, with $\Phi$ the standard normal cumulative distribution function. In the

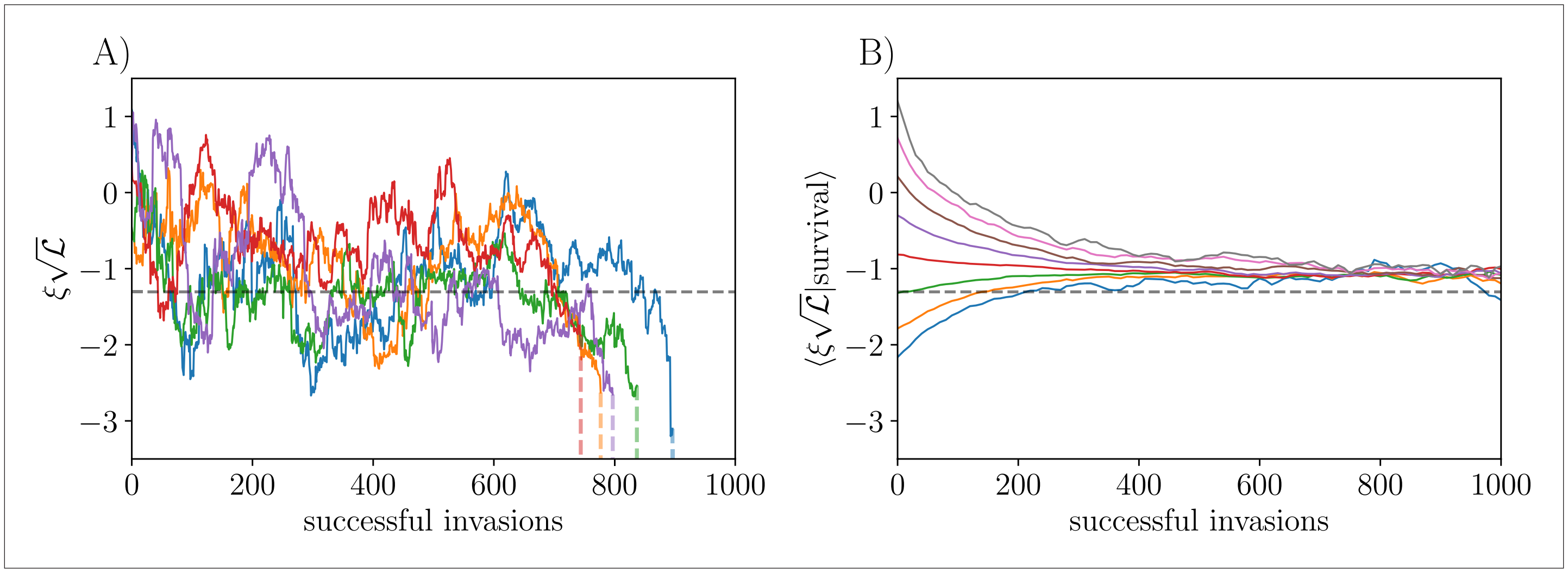


**Figure 4.** Trajectories of biases of persistent strains (normalized by $1/\sqrt{\mathcal{L}}$) under the influence of successive unrelated invaders with $s_i = 0$. (**A**) Bias trajectories of individual strains that invaded and persisted for a number of epochs. Extinctions (shown by a vertical line), occur when the bias goes below the critical bias, seen here to be around $-2.5/\sqrt{\mathcal{L}}$. The horizontal dashed line shows $-\bar{\Upsilon}\sqrt{\mathcal{L}}$. (**B**) Bias trajectories for all strains binned into groups by their starting value and averaged within bins for as long as the strains persist. Without conditioning on success, the biases of new invaders have mean $-\bar{\Upsilon}$ and standard deviation $1/\sqrt{\mathcal{L}}$. However, conditioned on survival, the biases converge to and fluctuate around a larger value. Data in (**A**) are from a single simulation where the community diversifies from 50 to 500 strains, and in (**B**) data are pooled from 10 replicates of the same process.

initial assembled community, $\xi_c\sqrt{\mathcal{L}}$ and $\bar{\Upsilon}\sqrt{\mathcal{L}}$ are independent of $\mathcal{L}$. We make the *Ansatz* that after a long period of evolution the distribution of extant biases, scaled by $\sqrt{\mathcal{L}}$, reaches a steady state — albeit a different state than the initial assembled community. Then the probability of successful invasion will become independent of $L$ for large $L$. If the mean number of extinctions per successful invasion also reaches a steady state value which is less than unity, this explains the steady linear growth of $L(T)$ seen in ***Figure 2***.

With the one-by-one introduction of new strains, the bias of each extant strain undergoes some kind of random walk, and the strain goes extinct if its bias ventures below $\xi_c$. In ***Figure 4A***, we show the evolutionary trajectories of the biases of 5 individual strains that started from similar initial values in a simulation where the community diversified from 50 to 500 strains. Extinctions are caused by $\xi$ being pushed below $\xi_c$ by an invading strain. For finite $L$, the sharpness of $\xi_c$ will be smeared by an amount of order $1/L$ due to variability in the dynamic noise from strain to strain, which we have not explored.

To numerically investigate any systematic components of the random walk of biases, we average over a large number of strains, binning them according to their initial values normalized by $1/\sqrt{\mathcal{L}}$. We observe a strong tendency of anomalously positive and negative biases to regress toward an intermediate value. In this plot, as evolution proceeds, the asymptotic average bias conditioned on survival is larger than $-\bar{\Upsilon}\sqrt{\mathcal{L}}$. (***Figure 4B***). This is likely due to conditioning on survival of the strains: those that persist for many epochs tend to have larger-than-average (but still negative) bias.

In 'Appendix 8', we carry out an analysis of the bias dynamics by approximating these by a Markov process in which the dynamics of the biases depend only on their current values. This analysis shows that when there is a successful invasion, each strain's drive undergoes both a systematic and random change, consistent with our numerical results.

It is convenient to work with the mean drives, $\bar{\nu}_i$ (when $s_i = 0$, this is just $\xi_i + \Upsilon$). Both the systematic and the random changes in the drive are proportional to the average abundance of the invading strain, which is of order $1/L$. The stochastic change is $\delta\zeta_i \sim \pm 1/L$, but the systematic change in the drive is smaller and depends on its current value: $\mathrm{E}[\delta\zeta_i|\zeta_i] \sim \zeta_i/L = \mathcal{O}(1/L^{3/2})$. In the Markovian approximation, there is a simple Langevin equation for the change of the drive of a strain due to invasions:

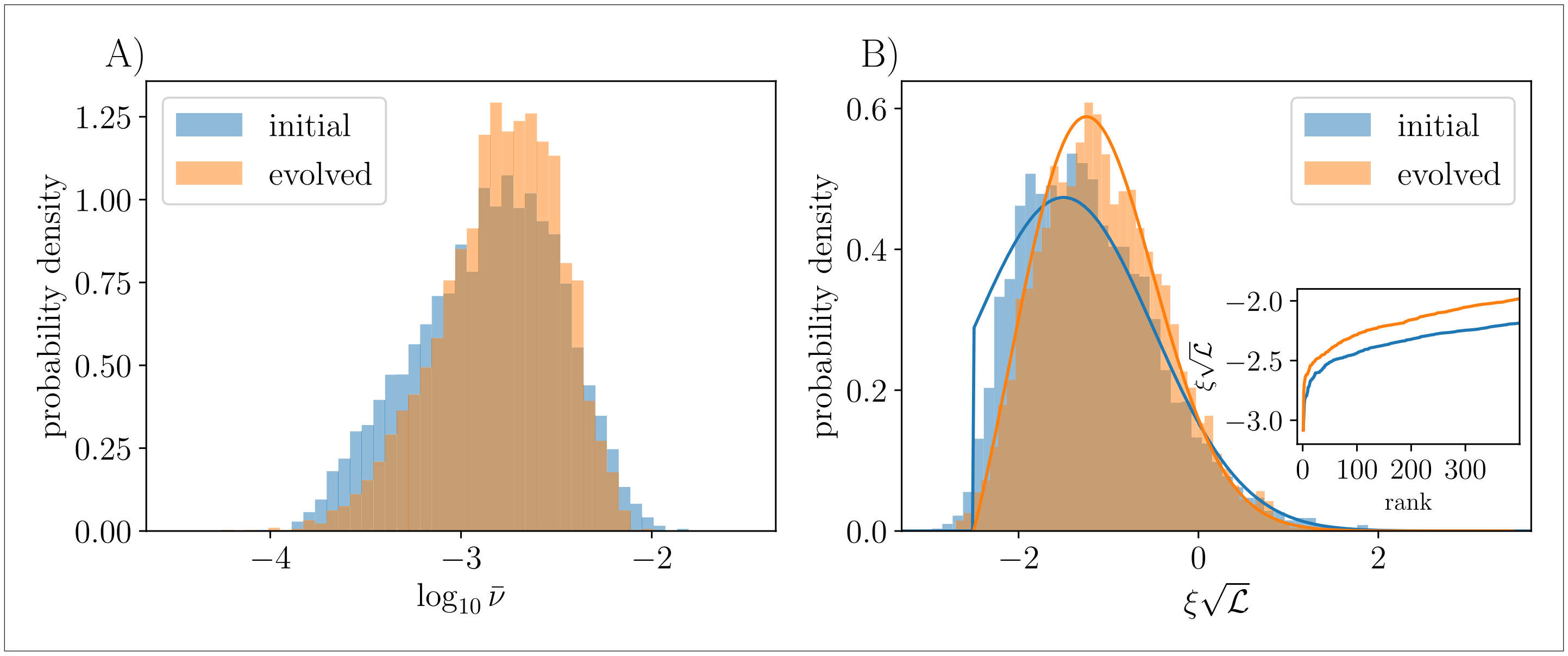


**Figure 5.** Distributions of mean abundances and biases before and after evolution with $\rho = 0$, $s_i = 0$. (**A**) Mean abundances: all these communities have $L \cong 500$, but evolved communities have diversified from $K = 50$ initial strains so that they lose memory of their initial assembly conditions, while assembled communities had $K = 650$ strains with $L_0 \cong 500$ surviving after the initial epoch. Data are pooled across 10 simulation runs of each. Though the distributions are mostly similar, there is a marked depletion in both rare and abundant strains in the evolved community. (**B**) The low end of the bias distribution changes from a truncated gaussian for the initial unevolved community (blue), to a linearly vanishing function (orange) because of evolution-driven extinctions of close-to-marginal strains. Bars show histograms from simulation, and solid lines show theory as detailed in 'Appendix 8'. Inset shows the normalized bias by rank order, illustrating the smoothing of the lower end of the distribution caused by the evolution.

$$\frac{d\zeta_i}{dT} = -B\frac{\zeta_i}{L(T)} + \sqrt{2D}\frac{1}{L(T)}\eta_i(T), \tag{3}$$

with $\eta_i(T)$ approximately gaussian with mean zero and unit variance — an approximation that should be good if one coarse-grains over a substantial range of $T$ (but with range much smaller than $L$). From our analysis in 'Appendix 8', we see that $B$ and $D$ are order-unity coefficients which respectively characterize the average and mean-squared response of the bias to the invasion of a new strain. Both are proportional to $\mathrm{E}[\nu_A^2]$ times the *fragility*, $\Xi$, of the extant community which is given by $\Xi = \sum_j \chi_j^2/(1 - \sum_j \chi_j^2)$ in terms of the individual susceptibilities of strains to changes in their biases, $\chi_i = d\bar{\nu}_i/d\xi_i$. This fragility characterizes the mean-square response of the system to a random perturbation applied simultaneously on all the strains — precisely the effect of a successful invasion. The Langevin equation for the drives must be supplemented by a boundary condition that if $\xi = \zeta - \Upsilon$ goes below $\xi_c$, the strain disappears.

Analysis of the Langevin equation, which can be converted into a Fokker Planck equation for the distribution of the drives ('Appendix 8'), shows that there is an eigenvalue-like condition which determines whether the diversification rate of the community is negative or positive, and that the coefficients $B$ and $D$ play a role in determining whether the community diversifies or not. This is consistent with our numerical results, which show that certain parameter regimes allow diversification and other regimes do not — even in the absence of general fitness differerences.

## Distribution of biases and number of extinctions

The approximate model of the evolution of the biases makes predictions about the shape of the bias distribution as a function of attempted invasions. Before the onset of evolution, the distribution of biases is a truncated gaussian, with a lower cutoff set by $\xi_c$. However as the distribution evolves according to ***Equation A8.3*** with the absorbing boundary condition at the critical bias, it smooths out near this cutoff, going linearly to 0 as $\zeta \to \xi_c + \bar{\Upsilon}$ (or, equivalently, as $\xi \to \xi_c$).

Simulations confirm the expectation that the typical bias scales as $1/\sqrt{L}$ over one order of magnitude in $L$ (***Appendix 3—figure 4***). As predicted, one observes a smoothing out of the bias distribution toward $\xi_c$ when comparing evolved and assembled communities of the same $L$ (***Figure 5B***), and our

analysis allows us to obtain the theory curve for the evolved ecosystem in **Figure 5B** as the solution of the approximate boundary value problem. However, the critical bias is sufficiently negative that the number of strains affected by the differences between the initial and evolved communities is small and the distinctions hard to see numerically.

However, the density of biases near $\xi_c$ determines the response of the community to evolutionary perturbations, since these low-bias strains are the ones most susceptible to extinction. In particular, the predicted linearly vanishing density of biases determines the distribution of the number, $\ell$, of extinctions per successful invasion (**Appendix 3—figure 1B**). To estimate this distribution — particular the probability that $\ell$ is large — we use the fact that an invader will perturb the extant strains' biases by a random amount of order $1/L$ and proportional to the mean abundance of the invader. The positive tail of the invader's mean abundance, $\bar{\nu}_A$, is gaussian, since for positive $\xi_{\rm inv}$, $\bar{\nu}_{\rm inv} \sim \xi_{\rm inv}$ and the invader's bias $\xi_{\rm inv}$ is itself gaussian distributed. The number of strains whose biases are within $\bar{\nu}_A$ of $\xi_c$ is proportional to $L^2\bar{\nu}_A^2$, because the distribution of strains' biases vanishes linearly at $\xi_c$. Thus for fixed $\bar{\nu}_A$, the number of strains that are driven extinct is Poisson distributed with mean proportional to $L^2\bar{\nu}_A^2$: this is of order one for large $L$ as expected. That the tail of the distribution of $\bar{\nu}_A$ is gaussian implies $\bar{\nu}_A^2$ is approximately exponentially distributed in its tail. Integrating the Poisson distribution over this yields, for large $\ell$, $\mathcal{P}(\ell) \sim e^{-\beta\ell}/\sqrt{\ell}$ (with $\beta$ an order-unity coefficient) which is close to exponential as observed in **Appendix 3—figure 1B**.

A similar analysis for the initial randomly-assembled community shows that for fixed $\bar{\nu}_A$, the mean number of extinctions triggered by the first successful invasion is of order $L_0^{3/2}\bar{\nu}_A \sim \sqrt{L_0}$; much larger than after evolution has proceeded for a while. As the Poisson with this mean has a narrow distribution, the probability of an anomalously large number of extinctions will be dominated by the gaussian tail of the distribution of $\bar{\nu}_A$ and hence itself be roughly gaussian, though unless the initial $L_0$ is huge, the tail is unlikely to still be in the asymptotic regime. The transient caused by a set of early invasions will likely cause a total of order $L_0$ strains — with a relatively small coefficient — to go extinct before $L$ starts steadily increasing, and this will occur over of order $L_0$ invasions. For $\gamma = -0.8$ this effect appears to be very small — the critical bias is quite negative — but for smaller or larger $\gamma$ the effects are noticeable (**Figure 2**).

The distribution of mean abundances, $\bar{\nu}_i$, is related to that of the biases via the function $\mathcal{N}(\xi)$: therefore, we expect this also to evolve as the community diversifies. In particular, there should be a reduction in the number of strains at low mean abundance, since these correspond to those with close-to-marginal bias. In **Figure 5A**, we see that the mean abundances in an evolved community are more narrowly distributed than in an assembled community, with both fewer highly abundant and fewer rare strains. This is consistent with our picture of the bias distribution being smoothed out toward $\xi_c$ due to invasion-triggered extinctions, resulting in the depletion of low-abundance strains. The depletion of abundant strains is likely due to the kill-the-winner dynamics which rewards invading strains that push the most abundant extant strains down.

Although the mean abundances are not broadly distributed on a log scale, the *snapshot abundance distributions* are, as seen in **Figure 1**. Note that most widely-used measures of diversity are not really informative for these kinds of logarithmically broad distributions. For example, the Shannon entropy would weight mostly the highly abundant strains, while the "species richness" would be highly sensitive to the lower cutoff in observable abundance.

## Diversification rate with a distribution of general fitnesses

Armed with understanding of the scaling of the bias and mean drives with $L$, we can build upon the analysis of the simple exponential distribution of general fitness ('Evolutionary dynamics with general fitness differences') to analyze the evolution when $\mathcal{P}(s)$ decays faster or slower than exponentially. A heuristic understanding of how the dynamics of $L$ depend on $\mathcal{P}(s)$ follows from the fact that without general fitness differences, the biases are distributed with characteristic scale $1/\sqrt{L}$. As $L$ increases, the distribution of these biases gets narrower, and the system becomes progressively more "neutral" with overall differences in strain biases becoming smaller. The contribution of the general fitnesses is to add a random extra piece to each bias, broadening the distribution of extant $\xi_i$. In the limit of many invasions, the width of the drive distribution becomes comparable to the width of the distribution of the extant $s_i$ and cannot decrease further. Thereafter, the shape of $\mathcal{P}(s)$ determines both the width of the bias distribution, and the number of coexisting strains.

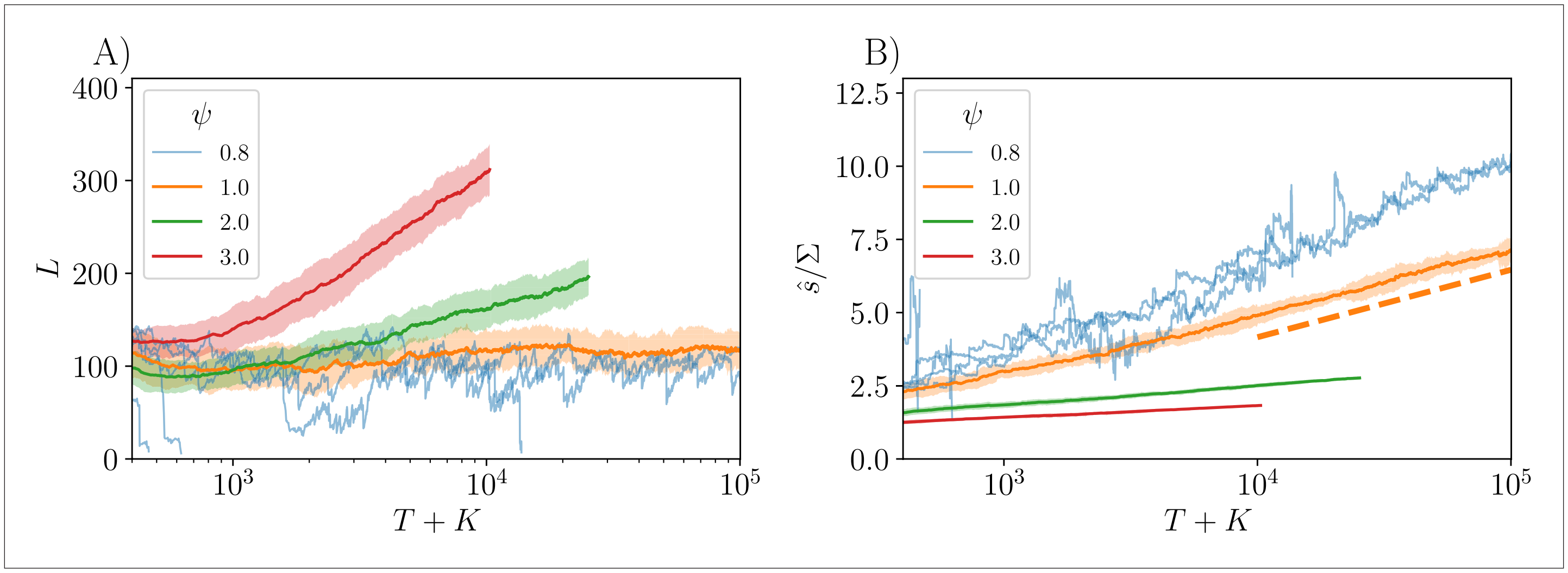


**Figure 6.** Evolutionary dynamics for unrelated invaders ($\rho = 0$), with general fitnesses drawn from distributions parametrized by various values of $\psi$: faster-than-exponential decay for $\psi > 1$ and slower-than-exponential for $\psi < 1$. Data are shown averaged over 50 replicates, conditional on not crashing, starting from $K = 300$ initial strains, with the shaded region showing the standard error. For $\psi = 0.8$, which results in decreasing diversity and crashes, a few individual trajectories are shown instead of an average. (**A**) Size of community as a function of the total number of strains introduced, $T + K$. For very long evolutionary times, we expect $L \sim [\log(T+K)]^{2-2/\psi}$, but transients due to initial conditions are substantial. In order to push up into the tails of the $s$ distributions, the parameters of $\mathcal{P}(s)$ are chosen differently for each $\psi$ for $\psi = 0.8, 1, 2, 3$ respectively. (**B**) Increase of the community-average $\hat{s}$ with $T$, shown pushing into the tail of $\mathcal{P}(s)$. For $\psi = 1$ the dotted line shows the theory prediction $\hat{s}/\Sigma \approx \log[(T+K)\Sigma^2]$, with deviations from this expected to be an $\mathcal{O}(1)$ constant for large $T$.

If $L \ll 1/\Sigma^2$ initially, the $s_i$ play little role and the population-weighted mean fitness, $\hat{s}$, only increases gradually. But once $\hat{s}$ is a few times $\Sigma$, the tail of $\mathcal{P}(s)$ will determine the rate of increase of $\hat{s}$. Henceforth, $\hat{s}$ will grow steadily with subsequent successful invasions, since strains with $s_i \ll \hat{s}$ are very unlikely to persist, and strains with $s_i \gg \hat{s}$ are unlikely to have yet occurred. Thus, the range of *extant* $s_i$ will become much narrower than $\Sigma$. This implies that the distribution over the currently relevant range can be approximated by an exponential distribution $\mathcal{P}(s) \sim e^{-s/\widehat{\Sigma}}$ with the effective width of the extant $s_i$ distribution given by

$$\widehat{\Sigma} = \left. \frac{-1}{d[\log \mathcal{P}(s)]/ds} \right|_{s=\hat{s}} = \frac{\Sigma^\psi}{\hat{s}^{\psi-1}} , \tag{4}$$

where the second equality is for the specific models we study (***Equation 2***). Provided evolution has proceeded long enough that no strains with $s_i$ smaller than the original scale $\Sigma$ survive, $\widehat{\Sigma}$ will vary slowly for a range of evolutionary time and this sets the scale for variations of $s_i$ of *both* extant strains and of potentially-successful invaders. This suggests that understanding the general behavior at long evolutionary times can be built on understanding the case of the simple exponential distribution ($\psi = 1$ for which $\widehat{\Sigma} = \Sigma$). The main difference is that now the community size will change as $L \sim 1/\widehat{\Sigma}^2$, with $\widehat{\Sigma}$ changing as $\hat{s}$ increases: this will govern how $L$ changes as invasions are attempted and occasionally occur.

In the slow evolution regime at long times, successful invaders must have $s_A$ comparable to $\hat{s}$. A simple argument gives an upper bound on how fast $\hat{s}$ can increase with $T$. In order to get a mean of $\hat{s}$ in a community of $L$ strains, at least $L$ attempted invaders must have occurred with $s_A \gtrsim \hat{s}$. This requires a number of invasion attempts, $T$, such that $T\int_0^{\hat{s}(T)} \mathcal{P}(s)ds \geq L(T)$. But if $\hat{s}$ were substantially smaller than this upper bound, many strains would have already occurred with $s_i - \hat{s} \gg \widehat{\Sigma}$ and these would persist for a long time, driving $\hat{s}$ up. We thus make the *Ansatz*, justified by the analysis and simulation data of 'Appendix 6' and ***Figure 6***, that $\hat{s}(T)$ grows with $\log(T)$ at a rate asymptotically given by this upper bound. For the distributions of interest we then have

$$T\mathcal{P}(\hat{s}) \sim L(T) \implies \frac{\hat{s}}{\Sigma} \approx \left[\psi \log(T/L)\right]^{1/\psi} \quad \text{and} \quad \frac{\widehat{\Sigma}}{\Sigma} = \left(\frac{\Sigma}{\hat{s}}\right)^{\psi-1} \implies L \sim \Sigma^{-2}[\log(T\Sigma^2)]^{2-2/\psi}, \quad (5)$$

where the last implication is due to $L \sim \widehat{\Sigma}^{-2}$. These scalings become valid once the distribution of the extant $s_i$ is pushed into the tail of $\mathcal{P}(s)$. To crudely take into account the effect of a large initial number of strains when $K \gtrsim 1/\Sigma^2$, $\log(T)$ can be replaced by $\log(K+T)$, as for the plots of $L(T)$ and $\hat{s}(T)$ in ***Figure 6***. ***Figure 6B*** shows the theoretical prediction for $\hat{s}(T)$ in the simplest case of $\psi = 1$, and we see that $\hat{s}(T)/\Sigma$ is reduced from the $\log[(T+K)\Sigma^2]$ prediction by only an $\mathcal{O}(1)$ constant, as expected.

At long times, the probability of successful invasion decreases very rapidly with $\hat{s}$ and, as we show in 'Appendix 6', the cumulative number of successful invasions for $\psi \geq 1$ grows very slowly, with $Z \sim \Sigma^{-2}(\log T)^{3-2/\psi}$. Nevertheless, for $\psi > 1$, the number of strains grows without bound albeit as a sub-linear power of the cumulative number of successful invasions. The average number of extinctions per successful invasion gradually decreases towards one as $\mathcal{P}(s)$ in the pertinent range, $s \approx \hat{s}$, becomes closer and closer to exponential.

For longer-than-exponential tails, $\psi < 1$, the diversity will decrease (possibly after an initial increase if $K \gg 1/\Sigma^2$) and eventually — in practice rather soon — crash as seen in ***Figure 6A***.

# Discussion

In this paper, we have answered an important issue of principle: Without any assumption of niche-like differences between strains, can diversity continually grow under slow evolution? We have found that this can indeed occur if the community forms a spatiotemporally chaotic phase that we have studied previously in PAF. As new strains are introduced — either separately evolved invaders or mutants of extant strains — some successfully invade, potentially driving extinctions of strains in the community. In a range of parameters, the size of the persistent community continually grows on average, while for other parameters, the diversity decreases and eventually crashes. How fast the diversification proceeds depends on the statistical properties of the strains. If each strain has a different general fitness, $s_i$, then as evolution proceeds the average $s_i$ of the population gradually increases and pushes into the tail of the $s_i$ distribution. If this tail falls off faster than exponentially, the community continues to diversify but more and more slowly, since fewer new strains will have sufficiently large $s_i$ to invade. For broader-than-exponential distributions, the diversity eventually crashes as the general fitness differences dominate over the effects of interactions with other strains.

Building on an analytic and scaling understanding of the STC phase for an assembled community, we have developed a substantial understanding of the dynamics of the diversification or de-diversification. However even for the simple models on which we have focused, there are aspects that we do not understand.

## Unresolved issues with the simple island models

### Development of correlations

Even with invaders uncorrelated with the extant strains, subtle correlations build up in the interaction matrix and — although they appear rather weak ('Appendix 4') — the memory of earlier evolution will affect the way strain abundances change under further evolution, potentially mandating a better treatment of the evolution than the Markovian approximation we have used in 'Evolution without general fitness differences'. With several complicating features — mutants, correlated general fitness differences, and substantial-sized initial communities — included, there are a number of crossovers that we have not attempted to analyze ('Appendix 7'). These, and which aspects promote, slow down, or prevent, continual diversification, are likely to be quantitative and strongly model-dependent.

### Nucleation of diversifying "phase'

An observation from the simulations ('Continual assembly and diversification') gives rise to a broader question: Why is it so hard to nucleate the diversifying STC phase? And, concomitantly, why do initially diverse communities so often crash unless the diversity is rather large? It is likely that the limited number of strains that dominate on each island over any short time interval — of order $L/\mathcal{M}$ — plays a role, but unclear how. Whether the difficulty of nucleating a diverse STC community is special to

the structure of the models and spatial dynamics assumed, or is true more generally, certainly needs further investigation.

### Spectrum of mutants and coexistence of parents and mutants

When invaders are mutants of extant strains that differ from their parent only very slightly, (with correlation coefficient $\rho$ very close to unity), we have found that the parent and mutant coexist surprisingly frequently. Understanding this, even for the first mutant, requires analyzing the dynamics of strains with strongly correlated noise which we have not carried out, although we suspect that the very large local abundance variations that occur with low-migration rate give rise to a small decorrelation scale needed for coexistence. In each simulation, we have considered only mutants with a fixed level of correlation with their parents, leaving a number of natural questions: What are the effects of a distribution of magnitudes of mutational differences? How do these affect the invasion, coexistence, and subsequent properties of the evolving communities?

### Invasion dynamics

Because of the local chaos and low migration, the invasion of a potentially-successful new strain is complex. To avoid this complication, we have introduced new strains at substantial abundance and on all islands simultaneously. In actuality, most initial invasion attempts on an island will fail: only if the strain arrives when the conditions are ripe for it to bloom, can it avoid quick extinction and send out enough offspring to other islands, which — if also sufficiently good timing — allow it to spread. How this process depends on the relatedness of mutant and parent complicates matters greatly because of the boom-bust dynamics. Strains are most likely to beget mutant offspring when their abundances are high, but at that stage of a bloom, a crash in the local population will soon follow. Therefore, although many mutants may arise when a parent strain is doing well, the correlation between their dynamics and those of their parent means that they are likely to quickly go extinct when their parent crashes down from high abundance. In contrast, mutants that emerge right before a parent blooms up to high abundance can ride the bloom and establish more readily, but would have to arise in a small parental population. Understanding the balance between these effects and their consequence for invasion probabilities is a challenge for future work — especially with real spatial structure and dynamics, discussed below.

## Spatial structure and dynamics

While for some microbial populations — for example common human gut commensals — a collection of connected "islands" without much spatial structure may be a rough caricature, for most populations there is spatial structure that makes dispersal from one location to another dependent on the distance in one, two or three dimensions. Thus, instead of having all pairs of islands connected by migration, one could model a $d$-dimensional array of islands with nearest-neighbor migration; a spatial continuum with diffusive dispersal; or a mixture of long and short distance dispersal events as driven by wind, ocean currents, or hitchhiking on migrant animals (***Hallatschek and Fisher, 2014***). With real spatial structure, local sub-populations are much more prone to extinction and cannot be as readily rescued by migration from another location where the strain is blooming. Thus, in contrast to the regime we have worked in for this paper, recovery from local extinctions must play a crucial role. The dynamics of invasions, extinctions, and repopulation is very different than in the spatial mean field model: if the underlying dynamics is diffusive, invasion and repopulation will occur by propagating Fisher-Kolmogorov-Petrovsky-Piscunov (FKPP) fronts (***Fisher, 1937***). The properties of FKPP waves are known to be highly sensitive to dynamics at the wavefront, and the effects of demographic fluctuations have been investigated (***Korolev et al., 2010***). But the approximately multiplicative "noise" from the ecological interactions will surely change this, and even for a single wave understanding the impact of these larger fluctuations is still an open question (***Rocco et al., 2002***; ***Rocco et al., 2000***).

With long-range dispersal over a multitude of length scales, the dynamics of invasion, extinction and repopulation will be very different, as already occurs for a single successful invader without ecological variations (***Hallatschek and Fisher, 2014***). Generally, understanding of the STC phase will have to build on better understanding of repopulation dynamics in the presence of large ecological fluctuations, and then understanding the evolution of communities on top of that. We leave investigations

of this for future work. But we conjecture that a continually diversifying STC phase can still occur with more realistic spatial dynamics.

## Bacteria-phage interactions and coevolution

An obvious weakness of the Lotka-Volterra models studied here is that the strains do not carry their own phenotypes, but are characterized by their interaction with all possible other strains. Furthermore, the antisymmetric correlations in the interaction matrix (especially without substantial general fitness differences) are rather unnatural for multiple strains of a single species. Thus, the most interesting extension of this work is to much more natural models: multiple strains of a phage species that prey on multiple strains of a bacterial species, with varying effectiveness that is a function of phenotypic properties of the particular phage and bacterial strains. Of particular importance is the interaction between a phage tail and bacterial receptor, as modelled in ***Weitz et al., 2005***. We showed previously [PAF] that the block-antisymmetrically-correlated structure of the interaction matrix with the bacteria having no niche-structure (differing only in the way they interact with the phages) can give rise to an STC phase that is very similar to that of the antisymmetrically correlated Lotka-Volterra model studied here: a similar model was further explored in ***Martis, 2022***. Such a bacteria-phage model can naturally accommodate general fitness advantages through phenotypic changes, eliminating the need to introduce them on separate footing. In ongoing work, we show that much of the basic phenomenology we have found here also occurs in evolving bacteria-phage phenotype models — at this stage only roughly and qualitatively.

For bacteria phage models, studying phylogenies and relatedness questions are natural. Whether more specialist phages tend to evolve, making the interaction matrix sparser and perhaps more hierarchical — and if so under what circumstances — is a particularly interesting question.

## Concluding questions

We have studied evolution of communities of many closely related strains in the limit that the evolutionary dynamics is slow compared to ecological and spatial dynamics. For a class of models, and in a particular ecological “phase”, evolution drives continual diversification, provided there is sufficient diversity initially. However mutations that change general fitness of strains tend to strongly slow down or even reverse the diversification. Thus we ask: How ubiquitous is diversification in the absence of any niche-like structure? Are there models in which a diversifying phase is easier to nucleate? Will the diversification always tend to be limited or strongly-slowed by general-fitness mutational effects? Or might “entropic” effects associated with difficulty of finding such general fitness mutations — for example from discrete genomes rather than continuous phenotypic parameters, or from soft tradeoffs — counter this slowdown, or perhaps produce evolutionary dynamics that lead to sparse interaction matrices and broader distributions of biases? Conversely, if strains are initially separated in “niche space” but then start to overlap and interact as the number of strains increases, how does the behavior differ? Is continual diversification easier to nucleate? Are the statistical properties of the phylogenies resulting from this evolutionary process — here driven entirely by “selection” in the broad sense, with ecological interactions creating a balance between the many extant strains — similar to a known class of coalescent trees?

What happens when, as in large microbial populations, evolutionary processes are not slow? Faster evolution is likely to make diversification easier, but understanding this even in simple models will require much better understanding of the invasion probabilities of mutants. Other than our scenario in which spatiotemporal chaos is the key to stabilizing coexisting diversity, what other robust continually diversifying scenarios are there? And of course, most crucially, what observable features of the strain, sub-strain, and sub-sub-strain level diversity in a microbial population (or interacting populations) could provide hints to the underlying causes of extensive diversity?

# Acknowledgements

We thank Pankaj Mehta for useful discussions and comments on the manuscript. This work was supported in part by the National Science Foundation via PHY-1607606,and PHY-2210386, the National Institutes of Health via R01AI13699201, the Simons Foundation via a sabbatical Fellowship to DSF, and the Stanford University Sherlock Cluster for the computations.

# Additional information


## Funding

| Funder | Grant reference number | Author |
|---|---|---|
| National Science Foundation | PHY-160760 | Aditya Mahadevan |
| National Institutes of Health | R01AI13699201 | Aditya Mahadevan |
| Simons Foundation | Sabbatical Fellowship | Daniel S Fisher |
| National Science Foundation | PHY-2210386 | Aditya Mahadevan |

The funders had no role in study design, data collection and interpretation, or the decision to submit the work for publication.


## Author contributions

Aditya Mahadevan, Software, Formal analysis, Investigation, Visualization, Writing - original draft; Michael T Pearce, Software, Formal analysis, Investigation, Visualization, Methodology; Daniel S Fisher, Conceptualization, Formal analysis, Supervision, Writing – review and editing


## Author ORCIDs

Aditya Mahadevan http://orcid.org/0009-0000-5571-9993
Daniel S Fisher http://orcid.org/0000-0002-5559-2491


## Decision letter and Author response

Decision letter https://doi.org/10.7554/eLife.82734.sa1

# Additional files

## Supplementary files

- MDAR checklist

## Data availability

The current manuscript is a computational study, so no data have been generated for this manuscript. Simulations use only standard algorithms: details in paper.

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

# Appendix 1

## Spectrum of timescales

The primary timescale that governs the interplay of evolution and ecology is the *ecological relaxation time* which is $\mathcal{O}(\mathcal{M}L)$, with $\mathcal{M} \equiv \log(1/m)$, as discussed in 'Timescales'. However the STC exhibits a number of other timescales. Although it does not play a role in the current work, there is a short timescale associated with the dynamic fluctuations: roughly the time that a strain spends near the peak of a bloom, which is similar to the inverse of its instantaneous growth or decay rate. This is dominated by the net effect of its interactions with the small subset of $\mathcal{O}(L/\mathcal{M})$ strains that happen to be abundant at that time: the variance of this is $\sum_j \nu_j^2 \sim \mathcal{M}/L$, which makes the *dynamic fluctuation time* $\sim \sqrt{L/\mathcal{M}}$. Between this dynamic fluctuation timescale and the timescale for blooms, correlations in the growth rates decay as a power of the time difference. During a bloom, each strain experiences multiple reversals from growth to decay: this is a special property of the self-organized chaotic state.

The time to go extinct for a strain destined to do so depends logarithmically on the extinction threshold $1/N$, but as long as $Nm$ is very large, whether extinctions occur is not strongly dependent on $N$, as analyzed in PAF. In our simulations we choose for convenience $Nm \gg 1$ so that $\log N$ is a few times $\mathcal{M}$.

The timescale for migration to be effective would, if there were no differences between the strains, be of order $1/m$ which is very long. However, the spatial dynamics are much faster than this because of the exponential growth of local populations when they happen to be in a favorable community. This makes the timescale for *exponential spread across islands* of a successful invader be on the order of the bloom time on a single island, $\mathcal{M}\sqrt{L}$. This is analogous to the rapid spread of a Fisher wave driven by selection, even when the spatial dynamics is diffusive ***Fisher, 1937***; ***Korolev et al., 2010***.

The timescale of demographic fluctuations — even if some strains were phenotypically identical — would be very slow $\sim N\tau_{gen}$ with $\tau_{gen} \ll 1$ (in our units) a generation time. In practice, these fluctuations only matter when the populations happens to be very small and is being driven extinct on the deterministic dynamic timescale $\sim \sqrt{L/\mathcal{M}}$. But as long as $N\bar{\nu}m\tau_{gen} \gg 1$, there are many migrants arriving and the dynamics is essentially deterministic. If the island-average $\bar{\nu}$ drops enough then the migrations become stochastic with time intervals between them of order $1/(N\bar{\nu}m)$. But when this occurs for substantial time, the global population is likely to be on the way to extinction. We focus on $Nm \gg L$ for which the population dynamics of the persistent strains are not strongly influenced by the stochastic or migratory demographic fluctuations.

# Appendix 2

## Numerics

### Parameters and integration

For all numerics, parameter values unless otherwise mentioned are $I = 40$, $m = 10^{-5}$, $\gamma = -0.8$, $N = 10^{20}$. In order to integrate the dynamics, we use an adaptive forward Euler step, with the time step chosen so that the maximum fractional change in the abundance of any strain is no greater than 3/4. This means that the time step scales with the dynamic fluctuation time ('Appendix 1') which is $\mathcal{O}(\sqrt{L/\mathcal{M}})$. Therefore, in a single time step an abundance can change to anywhere from 1/4 to 7/4 of its current value. Because of the wide range over which abundances vary, such large changes do not cause numerical problems. Extinctions and invasions are treated deterministically, with local extinctions occurring for strain $i$ on island $\alpha$ if $\nu_{i,\alpha} < 1/N$. In this case $\nu_{i,\alpha}$ is set to 0. An irreversible global extinction of strain $i$ occurs if $\nu_{i,\alpha} = 0$ for all $\alpha$ simultaneously. Recolonization of a locally extinct strain happens when $Nm\bar{\nu}dt > 1$, where $dt$ is the time step of the integration. Since this time step is chosen to reflect the basic timescale of the abundance dynamics, this choice of the recolonization threshold is consistent.

At the start of a new epoch, one strain is introduced into the community: this is done deterministically, with fractional abundance of the new strain set at $1/L$ on all islands and the other strains" abundances adjusted proportionately to maintain the overall population constraint. To save computational time, some incoming strains are rejected because they are very unlikely to successfully invade. This can be estimated by calculating their biases from the properties of the community before their introduction. By roughly estimating the critical bias from earlier successful invasions, we can conservatively reject some incoming strains without having to run the dynamics. With general fitness differences this can yield substantial speed-up by, for a community with population mean $\hat{s}$, rejecting strains with $s - \hat{s}$ sufficiently negative.

Since the epoch length is $\mathcal{O}(\mathcal{M}L)$ (set by the ecological relaxation time) and the time step scales as $\sqrt{L/\mathcal{M}}$, with each timestep requiring $\mathcal{O}(L^2)$ computations to compute the instantaneous growth rates of the strains, the runtime of a single epoch scales as $\mathcal{O}(\mathcal{M}^{3/2}L^{5/2})$.

### Correlations between mutant and parent interactions

Mutation of a parent strain to create an invader is comprised of two parts: changes in the parent's *interactions* with the other strains, and a change in the parent's general fitness. To generate the interaction part of a mutation, we append a new row and column to $V$ which parameterizes the interaction between the new strain $M$ and the parent strain $P$, each with another strain, $k$, according to $\mathrm{E}[V_{Mk}V_{Pk}] = \mathrm{E}[V_{kP}V_{kM}] = \rho$ for $M, P \neq k$. In order to preserve the correlation $\gamma$ between across-diagonal entries of $V$, we take $V_{Mk} = \rho V_{Pk} + \sqrt{1-\rho^2}Z_1$ and $V_{kM} = \rho V_{kP} + \sqrt{1-\rho^2}(\gamma Z_1 + \sqrt{1-\gamma^2}Z_2)$ where the $Z_i$ are i.i.d. standard normal random variables. This preserves the desired correlations, with $\mathrm{E}[V_{Mk}V_{kM}] = \gamma$.

However, we have to treat the direct interactions between the parent and mutant, $k = M, P$, more carefully, since it is not always possible to preserve $\gamma$ while also having the desired correlations between $V_{PP}$, $V_{PM}$ and $V_{MP}$. Defining $D$, $H$, $J$ and $K$ as

$$\begin{pmatrix} V_{PP} & V_{PM} \\ V_{MP} & V_{MM} \end{pmatrix} = \begin{pmatrix} D & H \\ J & K \end{pmatrix}, \tag{A2.1}$$

the symmetry conditions we enforce are

$$\mathrm{E}[D^2] = \mathrm{E}[K^2] = 1+\gamma, \quad \mathrm{E}[H^2] = \mathrm{E}[J^2], \quad \mathrm{E}[DJ] = \mathrm{E}[DH] = \mathrm{E}[KJ] = \mathrm{E}[KH] = \rho(1+\gamma). \tag{A2.2}$$

In addition, we require that $D = H = J = K$ in the limit $\rho \to 1$, and $\mathrm{E}[HJ] = \gamma$ as $\rho \to 0$. The parameterization that we choose which respects these conditions is

$$H = \rho D + \sqrt{1-\rho^2}Z_1 \tag{A2.3}$$

$$J = \rho D + \sqrt{1-\rho^2}(\gamma Z_1 + \sqrt{1-\gamma^2}Z_2) \tag{A2.4}$$

$$K = -\rho^2 D + \rho(H+J) + (1-\rho^2)\sqrt{1+\gamma}Z_3. \tag{A2.5}$$

where the $Z_i$ are i.i.d. standard normal random variables. Now we have $\mathrm{E}[HJ] = \rho^2 + \gamma(1-\rho^2)$, which does not precisely preserve the desired correlation $\mathrm{E}[HJ] = \gamma$, but it has the correct limit as $\rho \to 0$. Our parameterization gives $\mathrm{E}[DK] = \rho^2(1+\gamma)$, but it is possible to choose an alternative parameterization which has a different value of $\mathrm{E}[DK]$ but still respects the desired symmetries.

The dominant interaction between the parent and the mutant is mediated by the other strains in the ecosystem, with the direct influence of the parent on the growth rate of the mutant and vice versa smaller than the contributions mediated by all the other strains by a factor of $\sqrt{L}$. Therefore the choice of the direct parent-mutant interaction ought not to be of much importance — unless sufficiently strong correlations develop under continuing evolution, for which we do not see evidence.

## Epoch length in the diversifying phase

At the beginning of each simulation run, an ensemble of $K$ randomly drawn strains is assembled and run to reach a chaotic steady state under the dynamics of **Equation 1** for time $\mathcal{O}(\mathcal{M}\sqrt{K})$. Between the introductions of new strains the ecological dynamics are run for epochs of length $c_{\text{epoch}}\mathcal{M}L$ with $c_{\text{epoch}} = 3$. In addition, for most simulations we intersperse regular-length epochs with longer epochs to get rid of any marginal strains which may be barely surviving. These longer epochs (longer than the others by a factor of $\sim 3$) occur every 100 epochs.

The duration of the initial epoch with an assembled community of $K$ strains was usually chosen to be shorter than later epochs. But for the evolutions this does not much matter as the distribution of close-to-marginal strains is different than in later epochs and these are in any case quite likely to go extinct in subsequent epochs. However, for the simulations for which we compared assembled and evolved communities (e.g. **Figure 5**), the dynamics of the initial assembled communities were run for a longer time $12\mathcal{M}L$.

In order to check that the evolutionary dynamics are in the quasistatic limit, where the strains are introduced slowly enough that their introduction rate does not matter, we look at how the diversification rate depends on the length of the epochs between strain introductions. In **Appendix 2—figure 1** we vary $c_{\text{epoch}}$ over one order of magnitude and find similar evolutionary dynamics for all epoch lengths. In the case of $\rho = 0$, there is a longer transient for the long epochs, though once in the diversifying regime, $L$ increases at a similar rate for all epoch lengths. Based on this, all data presented in the paper were generated with $c_{\text{epoch}} = 3$, which makes numerics faster, and should not change qualitative conclusions about the slowly diversifying regime.

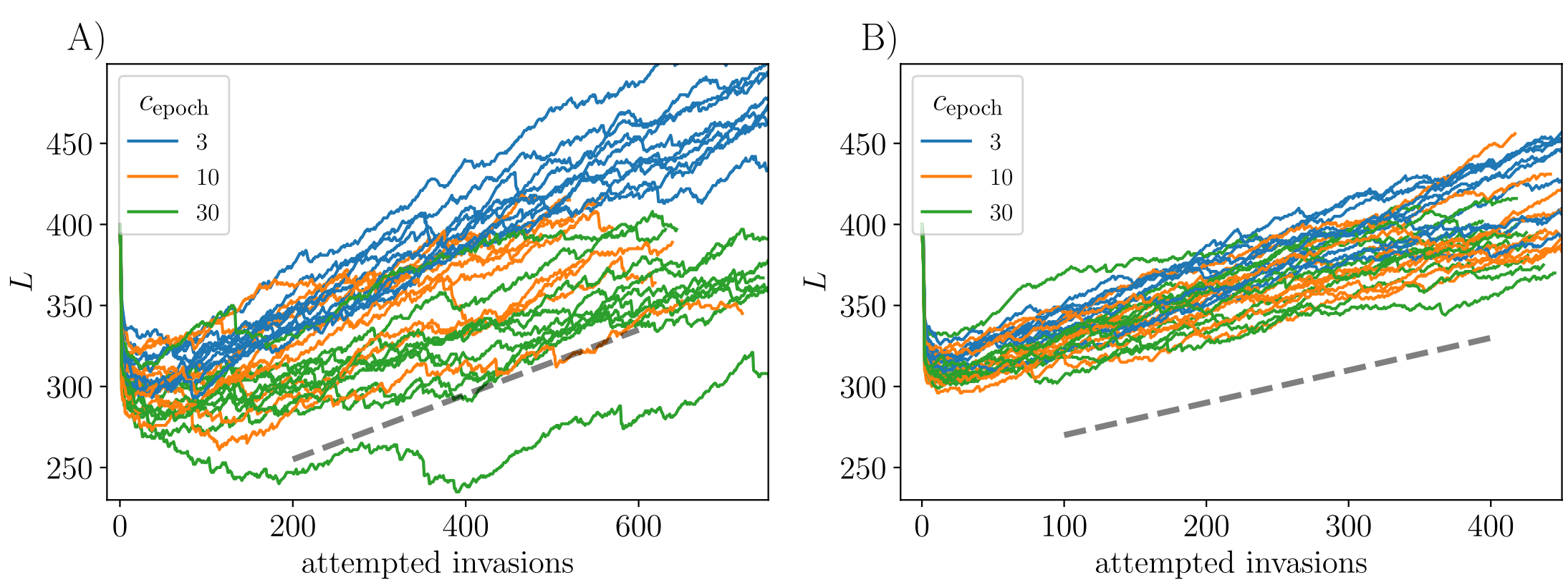


**Appendix 2—figure 1.** $K = 400$. (**A**) $\rho = 0$. (**B**) $\rho = 0.95$. Rates of diversification (measured per attempted invasion) are insensitive to a factor-of-10 change in the intervals between the introduction of the new types: the epoch durations shown are $c_{\text{epoch}}\mathcal{M}L$ with several values of $c_{\text{epoch}}$. Although it can take more invasions to get into the steadily diversifying regime for long epochs, the rates of diversity increase depend little on the epoch lengths. Both dashed lines both have slope 0.2 (diversification rates happen to be similar for $\rho = 0$ and $\rho = 0.95$).

## Estimation of drive and bias

The mean drive and bias are emergent quantities from the mean field analysis ('Appendix 5'), and are therefore not manifest in our direct numerical integration of the dynamics: however we can extract

them from numerics as detailed below. Here, we use the notation $\bar{\nu}_{i\backslash j}$ for the average abundance of strain $i$ in the absence of strain $j$. The averaged growth rate of a strain, excluding migration, can be measured directly. In the mean field approximation, this is decomposed as the sum of its bias and feedback from it perturbing the other strains::

$$\bar{r}_i = s_i + \sum_j V_{ij}\bar{\nu}_j - \bar{\Upsilon} \tag{A2.6}$$

$$= s_i + \zeta_i - \bar{\Upsilon} + \gamma X \bar{\nu}_i \tag{A2.7}$$

$$= \xi_i + \gamma X \bar{\nu}_i, \tag{A2.8}$$

where the drive, $\zeta_i = \sum_j V_{ij}\bar{\nu}_{j\backslash i}$, can be written in terms of mean field quantities as $\zeta_i = \sum_j V_{ij}\bar{\nu}_j - \gamma X \bar{\nu}_i$. The term $\gamma X \bar{\nu}_i$ captures the feedback of strain $i$ back onto itself via the other strains. Therefore, if we know the susceptibility, $X$, we can calculate the drive and bias of a strain: since $\bar{r}_i$ and $\bar{\nu}_i$ are measurable in our simulation, we need only to subtract off the $\gamma X \bar{\nu}_i$ term from $\bar{r}_i$ to get $\xi_i$. We can find $X$ using the self consistency condition that $X = \sum_i \frac{d\bar{\nu}_i}{d\xi_i} = \sum_i \frac{d\bar{\nu}_i}{d\zeta_i}$.

First, we make a guess for the susceptibility $X$ which allows a provisional calculation of the $\zeta_i$. Then, using our numerical simulation data, we fit parameters $a$, $b$ and $c$ to a functional form giving the mean abundance in terms of the drive and $s$:

$$\mathcal{N}(\zeta_i) = b \log\left[1 + \exp\left(\frac{\zeta_i + s_i - c}{a}\right)\right]. \tag{A2.9}$$

The justification for this form is that it gives $\bar{\nu}_i \sim \zeta_i + s_i - c$ for $\zeta_i + s_i - c > 0$. This is the expectation that $\bar{\nu}_i$ becomes proportional to $\xi_i$ for positive $\xi_i$ — though here we include an offset via the parameter $c$ which roughly represents the expected correction to the linear function $\mathcal{N}(\xi)$ at large bias. The quantity $\zeta_i + s_i - c$ is similar to $\xi_i$ but not quite the same due to a systematic difference between $c$ and $\bar{\Upsilon}$. For large negative bias, the fitted form captures the roughly exponential decrease of $\bar{\nu}$ due to the rareness of blooms.

The data are fit by this functional form quite well (**Appendix 2—figure 2**). We observe that $c$ tends to be smaller than the measured $\bar{\Upsilon}$ within a given epoch, which could be due to the effect of migration which elevates $\bar{\nu}_i$ for strains with negative $\xi_i$. The parameter combination $b/a$ is roughly equal to $-1/\gamma X$, as expected from the mean field analysis. From the hypothesized form of $\mathcal{N}(\zeta_i)$ we can recalculate the susceptibility via $X = \sum_i \frac{\Delta\bar{\nu}_i}{\Delta\zeta_i}$ (where the derivative is calculated numerically). We then update our value of $X$ by averaging the old guess with the new estimate.

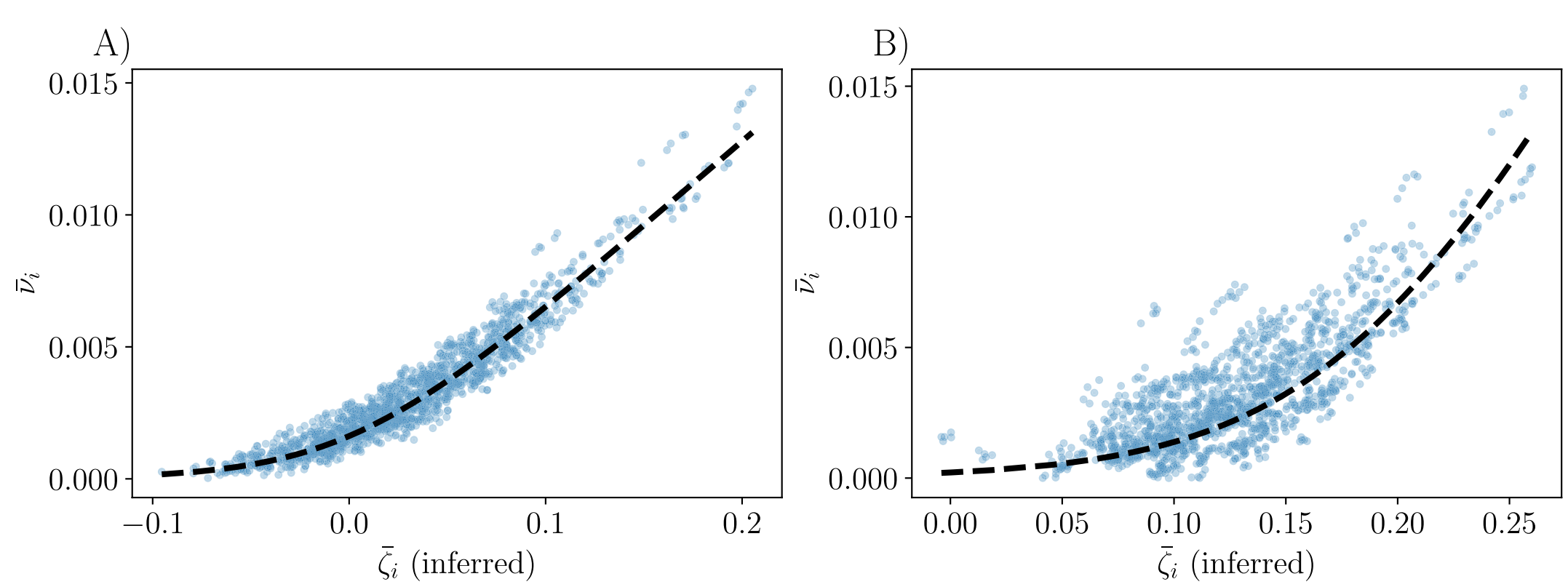


**Appendix 2—figure 2.** Inference of the drives, $\zeta_i$, for (**A**) communities evolving with $\rho = 0$ and (**B**) communities evolving with $\rho = 0.95$. In both cases, all $s_i = 0$. The black lines shows the fit of the functional form for $\mathcal{N}(\zeta)$ after $X$ has converged to a self-consistent value. Data are pooled (both for fitting and for plotting) over 5 consecutive epochs, each with $\sim 300$ extant strains. For $\rho = 0.95$ it seems that correlations in the interactions that have accumulated due to the relatedness are affecting the relationship between $\bar{\nu}$ and $\zeta$. Certain strains are visible outliers from the average dependence of $\bar{\nu}$ on $\zeta$, and appear multiple times on plots since data are pooled across 5 epochs.

By iterating over the susceptibility until it converges to a value where the assumed drives $\zeta_i = \sum_j V_{ij}\bar{\nu}_j - \gamma X \bar{\nu}_i$ reproduce the susceptibility via the self-consistency condition, we can get an estimate for the susceptibility and therefore the drive as well. Then the bias is $\xi_i \approx \zeta_i + s_i - \bar{\Upsilon}$, where we have subtracted the Lagrange multiplier (neglecting effects of order $\bar{\Upsilon} - \bar{\Upsilon}_{\setminus i}$ which are smaller by of order $1/\sqrt{L}$) and added back in the general fitness. We can validate that our bias estimator is reasonably accurate by comparing the inferred biases to the biases of strains in their first epoch of invasion — the latter can be measured directly from simulations since they are simply the invasion eigenvalues for incoming strains. As this works well, we use it to infer the biases for all the other strains in the community. (Note that in contrast to the assembled communities studied in PAF, we cannot use the condition that the drive averaged over all the initial $K$ strains, is zero. PAF used this — along with the expected truncated gaussian shape with variance $\sum_i \bar{\nu}_i^2$ and lower limit $\xi_c + \bar{\Upsilon}$ — as a check in calculating the drives. This no longer works here because conditioning on evolution means that the mean drive is no longer  and the drive distribution is no longer a truncated gaussian.)

To increase the quantity of data on which to perform our fit for $\mathcal{N}(\zeta)$ in a given epoch, we pool data from up to 20 epochs around the focal epoch, provided their $L$ is within 5% of the $L$ in the focal epoch, with the hypothesis that these communities are statistically similar and therefore have a similar $\mathcal{N}(\zeta)$.

The function $\mathcal{N}(\xi)$ for the evolved communities with $\rho = 0$ looks quite similar to the function for assembled communities However for communities evolved with $\rho = 0.95$, $\mathcal{N}(\xi)$ looks substantially different. This indicates that the build-up of correlations in the community requires modifications of the independence assumptions of the mean field theory.

After inferring the function $\mathcal{N}(\zeta)$ and the drive for each strain, we can calculate the fragility, defined as $\Xi = \sum_j \chi_j^2/(1 - \sum_j \chi_j^2)$. The closer $\sum_j \chi_j^2$ is to 1, the more unstable the community is to perturbations. Indeed we find that for simulations diversifying from 50 to 500 strains, $\sum_j \chi_j^2 \cong 0.6$ for the evolved community of $\cong 500$ strains, while early in the simulations, when $L < 100$, we find $\sum_j \chi_j^2 \cong 0.75$. When comparing $\sum_j \chi_j^2$ for assembled and evolved communities, both of 500 strains, there is not a clear difference. Note that the close-to-marginal strains, which are more abundant in the assembled community, only contribute small amounts to $\Xi$, so this is not surprising.

# Appendix 3

## Dynamics of diversification

### Extinctions, diversity crashes, and nucleation from low diversity

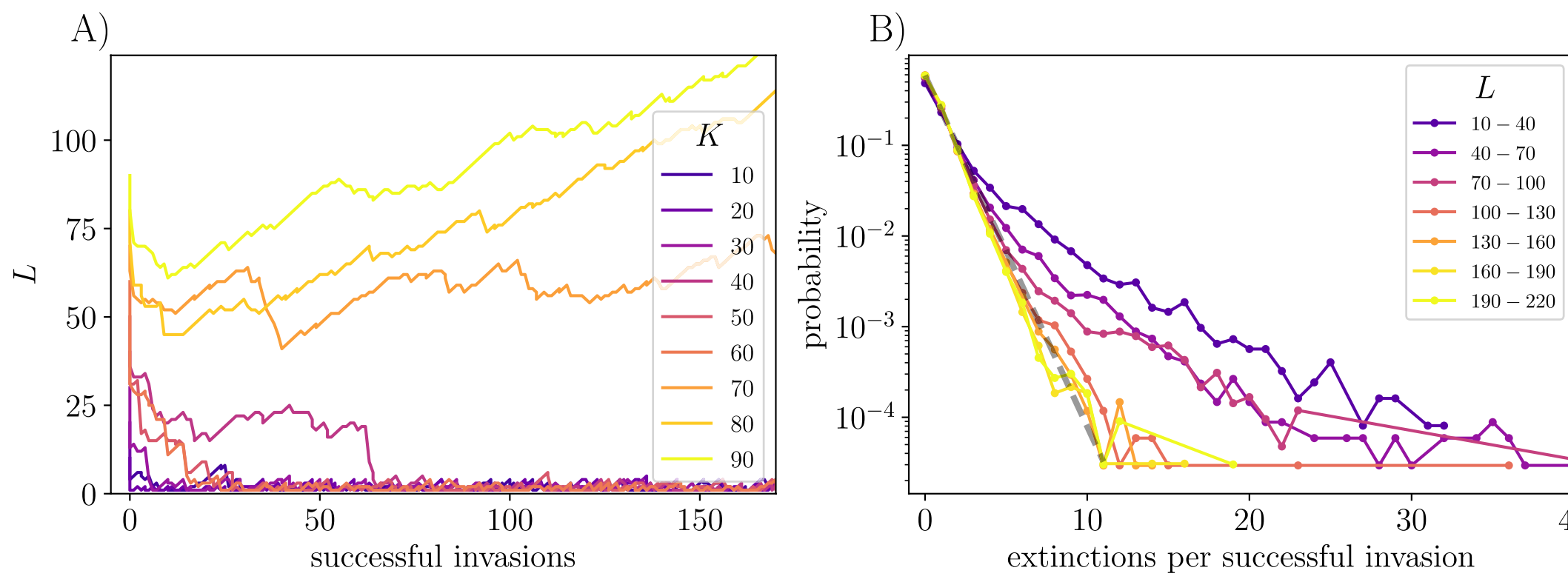


**Appendix 3—figure 1.** Crashes and distribution of number of extinctions per invasion. (**A**) The fate of an evolving community depends on the number of initial strains $K$. The characteristic size below which crashes dominate is $\cong 50$ for the parameters $\gamma = -0.8$ and $\rho = 0$ as shown here. (**B**) The distribution of the number of extinctions per successful invasion depends on the size of the community. Data are aggregated across a range of $K$, each run with 100 replicates. For small community size, $L$, invasions occur in which a substantial fraction of the strains go extinct. But for large $L$, multiple extinction events are very rare and the distribution is close to geometric: the dotted line is $\mathcal{P}(\ell) = (1-\alpha)\alpha^{\ell}$ for the probability of $\ell$ extinctions, with $\alpha \cong 0.41$, corresponding to an average of 0.7 extinctions per successful invasion as shown in ***Figure 2A***.

A crucial question is whether it is possible to build up a highly diverse community from a small initial number of strains and, if so, on what this depends. Starting simulations at values of $K$ between 10 and 90, we observe a crossover size, $K^*$, of the initial number of strains above which diversification is robust, and below which the diversity typically crashes under the evolutionary dynamics and does not again increase substantially for the duration of the simulations, (***Appendix 3—figure 1A***). The presence of a crossover $K^*$ suggests that once the system is sufficiently diverse, it tends toward further diversity. Thus the main obstacle to diversification in these models is going from a single strain to $\cong 50$ strains. We define $K^*$ heuristically as the lowest $K$ that diversifies with probability more than 1/2 (see ***Appendix 3—figure 2***). For each $K$, there is a corresponding $L_0(K)$: the number of strains that persists after the initial drop in diversity. This somewhat-variable $L_0$ appears to be the primary determinant of a community's stability to evolutionary perturbations. However this stability also depends on the community's history. For example, if the community has evolved gradually from a smaller number of strains to some current size $L_E$, its response to continued evolution may be different than if it has been assembled from an initial number $K$ that dropped through its initial evolution to an ecologically stable community of size $L_E$. This motivates definition of a crossover size $L^*$, different from $L_0(K^*)$, given by the minimum size of an *evolved* community for which the probability of entering the diversifying phase is greater than 1/2. We discuss the differences between assembled and gradually evolved communities further in 'Appendix 4', but leave an analysis of how these properties affect the nucleation probability for future work. The size $L^*$ is a natural quantity to work with, as it is the relevant one when considering how a community might nucleate from a small number of strains into a diversifying regime. As we discuss later, we expect $L^* < L_0(K^*)$ since just-assembled communities are more susceptible to perturbations than gradually assembled ones.

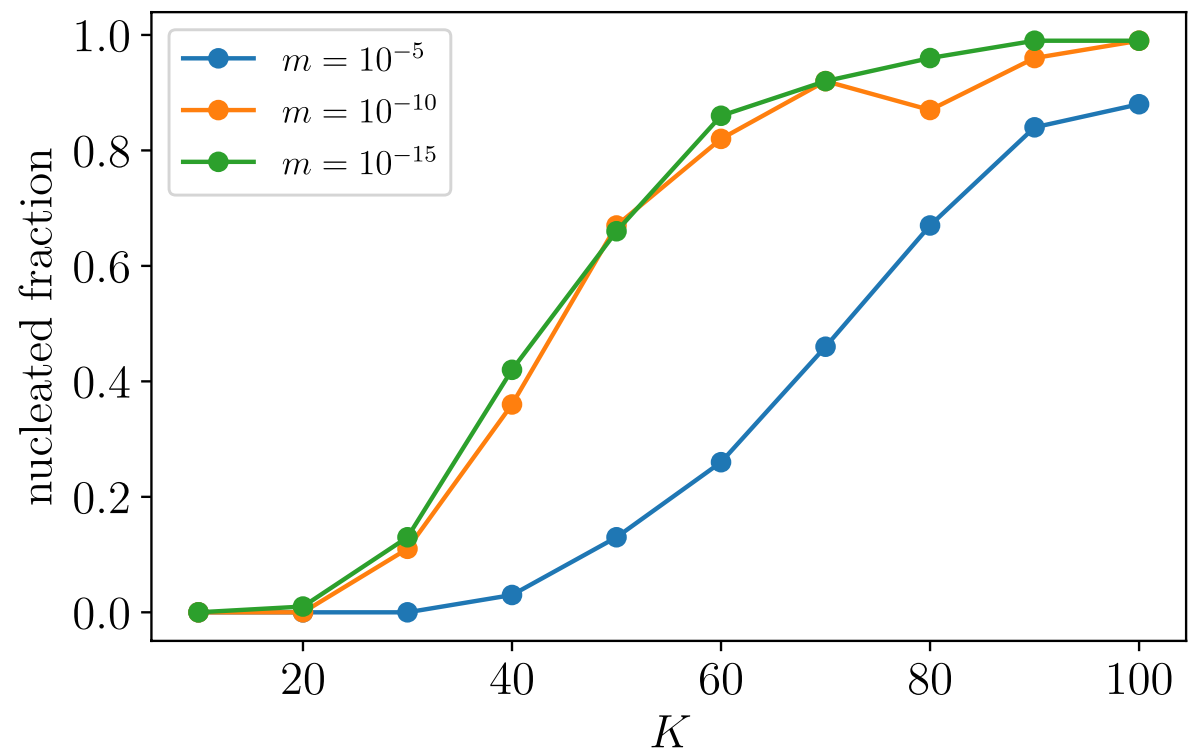


**Appendix 3—figure 2.** The probability of establishing the steadily diversifying regime increases with the initial number of strains and decreases with the migration rate. Although changing the migration rate from $10^{-5}$ to $10^{-10}$ changes the establishment probability significantly, further reducing $m$ has little noticeable effect, indicating that the dependence of establishment probability on $m$ is rather weak over several orders of magnitude. Data are averaged over 100 simulations for each set of parameter values. If $m = 0$, then migration cannot stabilize the spatiotemporal chaos, and so the establishment probability should vanish — but this is a very singular limit.

We can quantify how likely the system is to diversify or crash by inspecting the number of extinctions, $\ell$, that occur for each invader that successfully enters the ecosystem. The distribution of $\ell$ is dependent on $L$ and shows that more diverse ecosystems suffer on average fewer extinctions triggered by invasion of each new strain (***Appendix 3—figure 1B***). For small communities $L$ of order $L^*$ or less, extinctions of a substantial fraction of the strains occur, and cascades of extinctions in response to successive invasions cause $L$ to crash. The fragility of the communities with $L \cong L^*$ to evolutionary perturbations and their strong tendency not to recover after a crash (***Figure 2A***), implies that the process by which a low diversity community *could* diversify is very different than the steady diversification of already large communities. The transition from the low diversity to the diversifying regimes must be mediated by a very rare nucleation event in which $L$ becomes roughly larger than $L^*$. We discuss the low diversity regime and speculate about such nucleations in 'Appendix 3'.

In the steadily diversifying regime the distribution of extinctions per successful invasion is close to exponential — a result that we derive in 'Distribution of biases and number of extinctions'. For large $L$, the chances that a substantial fraction of the strains go extinct is extremely small and decreases exponentially as $L$ increases further: thus for large $L$ the continual diversification is essentially deterministic.

## Probability of diversification as a function of initial community size

The initial community size beyond which diversification becomes likely depends $m$, $\gamma$ and $\rho$, and is defined as $K^*$. For some range of $\gamma$ (***Figure 2B***), $K^*$ is infinite: the system always eventually crashes. The role of $m$ in determining $K^*$ can be partially understood from the mechanism of diversity crashes due to synchronization of dynamics between islands. This occurs more frequently when $m$ is larger, but its dynamics are subtle because of local blooms up to high abundance which can can have outsize effects. Though we have not studied the synchronization and diversity crash process here, it suggests an interpretation of our simulation results (***Appendix 3—figure 2***) which show that decreasing $m$ lowers $K^*$, presumably because it makes the STC more difficult to spatially synchronize and hence less likely to crash.

## Nucleation from low diversity

Here we further investigate the evolutionary dynamics when the diversity is low. We are particularly interested in the nucleation process by which an ecosystem could transition from the $L \ll L^*$ regime to the steadily diversifying regime, where $L^*$ is the crossover beyond which continual diversification becomes likely. For the parameters we have investigated, this turns out to be extremely rare. Even in runs of $10^5$ attempted invasions starting from a single strain, the community always remains small. We observe that the number of extinctions per successful invasion is broadly distributed (as when starting with a larger community $L \sim L^*$ in ***Appendix 3—figure 1***) and the evolutionary dynamics of $L$ typically proceeds by incremental increases punctuated by large decreases. In this regime, the

steady state distribution of $L$ is observed to be very close to Poisson (***Appendix 3—figure 3***) — a result that we do not have a theory for. From this Poisson behavior, one can attempt to extrapolate the probability of reaching $L^*$ and transitioning to steady diversification by a lucky fluctuation in $L$. Taking $L^* \approx L_0(K^* \approx 70) \approx 55$ based on Figures ***Appendix 3—figure 2*** and ***Appendix 3—figure 1***, this estimate suggests $\sim 10^{60}$ attempted invasions would be needed. However, the probability distribution of $L$ will surely deviate from Poisson long before $L^*$, so our prediction for the chance of nucleation is likely too small by many orders of magnitude, depending on the value of $L$ for which the probability distribution starts to be significantly higher than the Poisson extrapolation. At this point we do not have even conjectures for the shape of this distribution, nor for how the value of $L^*$ changes with $\gamma$, $\rho$ or $m$. At the marginal point $\gamma = -1$, we have the special antisymmetric model ('Appendix 6'), and diversification becomes impossible, so near there we expect that $L^*$, along with the fragility, diverges.

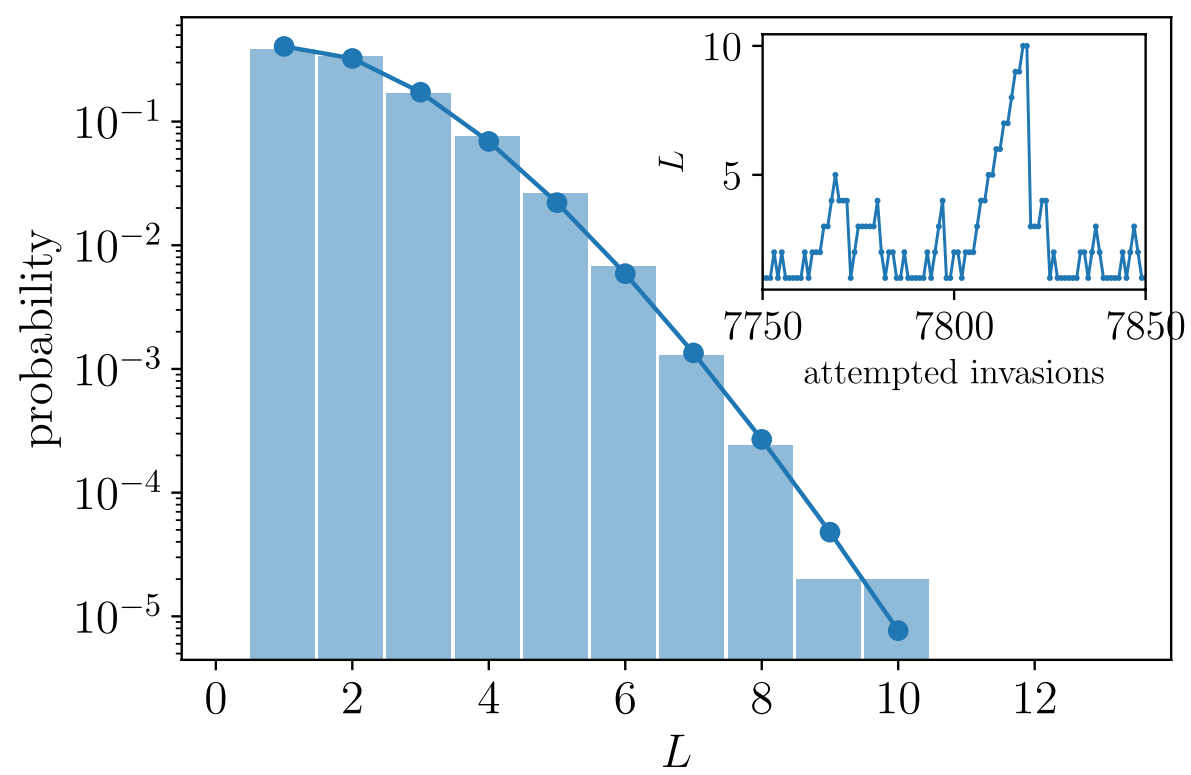


**Appendix 3—figure 3.** Distribution of community size $L$ starting from a single strain, over $10^5$ attempted invasions with $m = 10^{-5}$ and $\gamma = -0.8$. The solid line shows a Poisson fit with mean 1.6, excluding $L = 0$. This fit is remarkably good. The inset shows the trajectory of $L$ as it reached its maximum value which occurred only once, followed by a crash back down.

Results from 'Appendix 3' suggest that the nucleation process may be related to desynchronization which is necessary to enter the steadily diversifying regime. In fact we observed that decreasing the migration rate $m$ increases the mean of the Poisson-like distribution of $L$ in the low-diversity regime, and also reduces $L^*$ (***Appendix 3—figure 2***), thereby perhaps substantially increasing the probability of nucleation — although it is still too improbable to observe.

## Scaling relationship between bias distribution and $L$

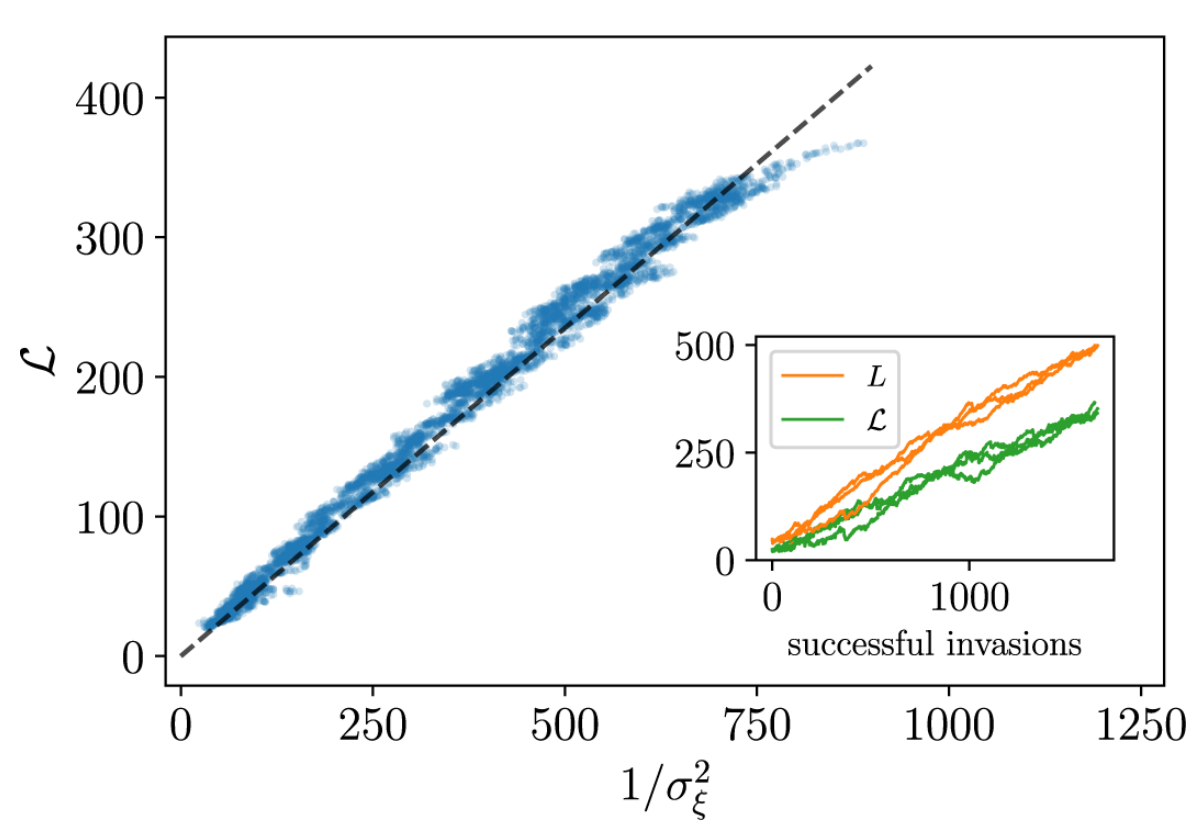


**Appendix 3—figure 4.** Scaling relationship between the effective community size, $\mathcal{L} \equiv 1/\sqrt{\sum_i \bar{\nu}_i^2}$, and the inverse variance of the bias distribution, $1/\sigma_\xi^2$, for unrelated invaders with $s = 0$. The dashed line has slope 0.47. Data are pooled over 3 runs from $L = 50$ to 500 strains (inset) so that when $L$ becomes large the initial conditions have been forgotten. Inset shows that $L$ and $\mathcal{L}$ are proportional as expected. (The bend in $\mathcal{L}$ versus $1/\sigma_\xi^2$ toward the end of the curve is likely due to stopping the simulation the first time $L$ reaches 500, which truncates the curve asymmetrically.).

The central scaling relationship from which our theoretical results follow is that the width of the bias distribution scales as $1/\sqrt{L}$, which scales the same way as $1/\sqrt{\mathcal{L}} \equiv \sqrt{\sum_i \bar{\nu}_i^2}$. In ***Appendix 3—figure 4***, we confirm these relationships by calculating the standard deviation of the bias distribution in a diversifying community where $L$ ranges over one order of magnitude. We find the the relationship holds for both $L$ and $\mathcal{L}$, which are proportional to each other with $\mathcal{L} \approx 0.74 \times L$.

## Diversifying phase with $\rho$ close to 1

One of the important results of this work is that there is a continually diversifying phase of eco-evolutionary dynamics both for unrelated invaders and for small-effect mutations, across the whole range of $\rho \in [0, 1)$. In ***Appendix 3—figure 5***, we illustrate the robustness of this phenomenon for highly correlated parents and mutants with $\rho = 0.97$, running a large number of replicate simulations and seeing that about 10% of these enter the diversifying phase, when $K = 50$. Although the community is less likely to nucleate into the diversifying phase than when $\rho = 0$, $L$ still increases linearly with attempted invasions once the community is nucleated.

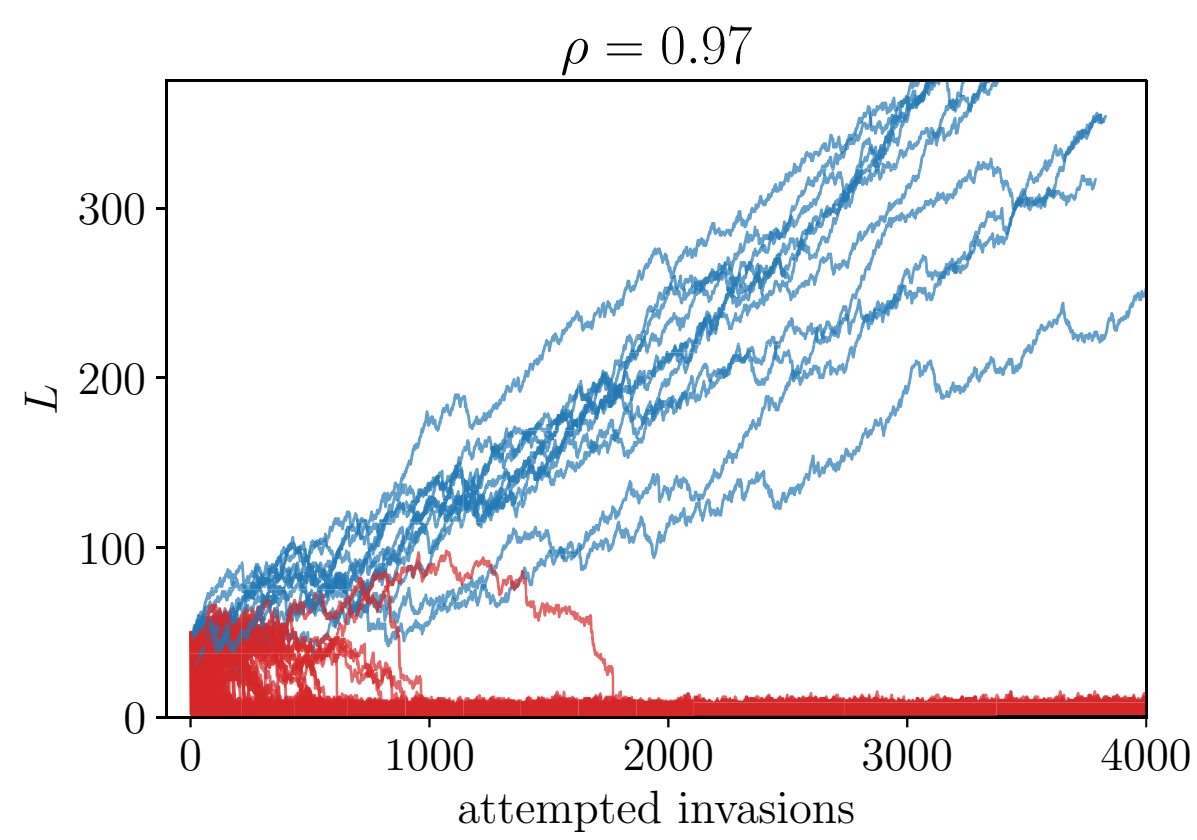


**Appendix 3—figure 5.** The diversifying phase persists for $\rho$ close to 1. Here we show 99 simulation replicates, starting from $K = 50$. Out of these, 11 nucleated into the diversifying phase, albeit with diversity increasing at a slower rate than with $\rho$ smaller (***Figure 2***). Simulations which crashed are shown in red while those that entered the diversifying phase are shown in blue.

# Appendix 4

## Statistics of $V$ conditioned on evolution

How do the statistics of the interaction matrix $V_{ij}$ change when conditioned on evolutionary history? How does this compare to conditioning on non-extinction in an assembled community? We ran a set of 10 replicates without general fitness differences over a period where the community grows from 50 initial strains to $\cong 500$ strains, for both $\rho = 0$ and $\rho = 0.95$. From these, we obtained ensembles of interaction matrices conditioned on gradual evolution with $\rho$ both 0 and close to 1.

We also examined the interaction matrices of communities that evolved with a steady state $L$, by using an exponential distribution of $s$ with $\Sigma = 0.07$, starting from $K = 50$ and reaching steady state with $L \cong 150$ while evolving over the course of $\cong 4000$ successful invasions. In this case one might be more likely to see correlations building up, since new strains might not dilute the correlations building up in $V$. For comparison, we studied assembled (but not evolved) communities starting with sufficiently large $K$ to have $L_0$ similar to the evolved condition.

**Appendix 4—table 1.** Definition of some statistics of the interaction matrix of a community. The standard deviation is calculated as $\mathrm{Stdev}_i[\{x_i\}] = \sqrt{\frac{1}{L}\sum_{i=1}^{L}(x_i - \frac{1}{L}\sum_{j=1}^{L}x_j)^2}$ (the population standard deviation) and $\mathrm{Corr}[X, Y]$ is the Pearson correlation coefficient calculated for a sample (the covariance normalized by the product of standard deviations).

| $\widehat{\gamma}$ | $\widehat{\sigma_V}$ | $\widehat{\mu}_V$ | $\widehat{\Delta}$ |
|---|---|---|---|
| $\mathrm{Corr}[V_{ij}, V_{ji}], i \neq j$ | $\mathrm{Stdev}_{ij}[V_{ij}]$ | $\frac{1}{L^2}\sum_{i,j} V_{ij}$ | $\mathrm{Stdev}_j\left[\frac{1}{L}\sum_i V_{ij}\right]$ |

In ***Appendix 4—table 1***, we list the summary statistics of the interaction matrices that we analyzed: $\widehat{\gamma}$ is the empirical cross-diagonal correlation in the $V$ matrix of extant strains, $\widehat{\sigma_V}$ is the empirical width of the interaction strength distribution after evolution, $\widehat{\mu}_V$ is the average of the interactions and $\widehat{\Delta}$ is the standard deviation, over $i$, of $\frac{1}{L}\sum_j V_{ij}$, which acts like an effective $s_i$, crudely approximating the width of the bias distribution. Aside from $\widehat{\gamma}$, all the quantities are defined including the diagonal terms in $V$.

**Appendix 4—table 2.** Means (up to corrections due to diagonal entries) and standard deviations of the chosen statistics from 500 realizations of the matrix $V$ from the original ensemble with $\gamma = -0.8$, and two values of $L$ pertaining to the different evolutionary conditions. Here we do not condition on either non-extinction or evolution.

| | $\widehat{\gamma}$ | $\widehat{\sigma_V}$ | $\widehat{\mu}_V$ | $\widehat{\Delta}$ |
|---|---|---|---|---|
| mean ($L$ = 500) | −0.8 | 1 | 0 | 0.0446 |
| standard deviation ($L$ = 500) | 0.001 | 0.002 | 0.001 | 0.0015 |
| mean ($L$ = 150) | −0.8 | 1 | 0 | 0.0816 |
| standard deviation ($L$ = 150) | 0.0036 | 0.006 | 0.0031 | 0.0046 |

**Appendix 4—table 3.** Statistics of the interaction matrix for both evolved and assembled communities. Evolved communities grew to $\cong 500$ strains from an initial $K = 50$ and assembled communities had $L_0 \cong 500$ after a single epoch of duration $12\mathcal{M}L$, both with $s_i = 0$. The last row is from a simulation with exponential $\mathcal{P}(s)$ and $\Sigma = 0.07$, which results in $L \cong 150$ at steady state. The mean of each statistic is reported across 10 simulation replicates. In Appendix 4—table 4 we show these statistics appropriately normalized with respect to the original $V$ ensemble. Bold entries correspond to those more than 3 standard deviations away from the original ensemble mean.

| | $\widehat{\gamma}$ | $\widehat{\sigma_V}$ | $\widehat{\mu}_V$ | $\widehat{\Delta}$ |
|---|---|---|---|---|
| assembled ($\Sigma = 0$) | −0.8014 | 1.0012 | 0.0026 | **0.0337** |
| diversifying ($\Sigma = 0,\ \rho = 0$) | −0.8015 | 1.0028 | **0.0061** | **0.0289** |
| diversifying ($\Sigma = 0,\ \rho = 0.95$) | **−0.8082** | **1.0427** | **0.1422** | **0.0328** |
| steady state ($\Sigma = 0.07,\ \rho = 0$) | −0.8074 | **1.0205** | **0.0165** | **0.0977** |

To check for statistical significance, we normalize each statistic of the evolved interaction matrices by the standard deviation of the estimator of the corresponding statistic from matrices drawn from the original ensemble without any conditioning. This indicates how atypical the measured statistics are for the matrices of size $L \times L$. We thus define quantities $\tilde{\gamma}$, $\tilde{\sigma}_V$, $\tilde{\mu}_V$ and $\tilde{\Delta}$ to be the empirical matrix statistics measured in units of standard deviations away from their mean in the original $V$ ensemble.

**Appendix 4—table 4.** Properties of the interaction matrix for evolved and assembled communities, displayed in terms of number of standard deviations from the mean in the original $V$ ensemble. These are the same data as in Appendix 4—table 3, but normalized according to appropriate scale of deviations calculated from original $V$ ensemble. Numbers in boldface have magnitude greater than 3, and are thus clearly statistically significant.

| | $\tilde{\gamma}$ | $\tilde{\sigma}_V$ | $\tilde{\mu}_V$ | $\tilde{\Delta}$ |
|---|---|---|---|---|
| assembled ($\Sigma = 0$) | −1.4 | 0.6 | 2.6 | **−7.2** |
| diversifying ($\Sigma = 0,\ \rho = 0$) | −1.5 | 1.4 | **6.1** | **−10.4** |
| diversifying ($\Sigma = 0,\ \rho = 0.95$) | **−8.2** | **21.4** | **142.2** | **−7.9** |
| steady state ($\Sigma = 0.07,\ \rho = 0$) | −2.1 | **3.4** | **5.3** | **3.5** |

Although a considerable number of statistics are significantly different from their original ensemble values, few incur substantial changes, with the largest change by far occurring in the $\mu_V$ for the simulations with $\rho = 0.95$; however even this is a relatively small effect.

Our results show that both evolution and assembly tend to make $\widehat{\gamma}$ slightly more negative than the $\gamma$ of the original ensemble, and that these processes also favor — as might have been expected — an increasing mean interaction strength, $\widehat{\mu}_V$. Of particular interest, given previous discussion, is $\widehat{\Delta}$, which is similar to the width of the bias distribution. For $L \cong 500$, the statistics of the matrices from the original ensemble have $\widehat{\Delta} \cong 1/\sqrt{500} \cong 0.0447$. However, once we condition on assembly or evolution, the value of $\widehat{\Delta}$ can be expected to decrease, because we eliminate those strains with negative bias, meaning that strains $i$ with an anomalously small value of $\sum_j V_{ij}$ will go extinct, making the range of $\sum_j V_{ij}$ narrower. This is indeed observed in ***Appendix 4—table 3***, at least for simulations without general fitness differences. Interestingly we see that $\widehat{\Delta}$ decreases with evolution more for $\rho = 0$ than for $\rho = 0.95$, which could be because, in the latter case, the elements in a single row of $V$ are correlated with those of a high-abundance parent, which could increase the width of the $\sum_j V_{ij}$ distribution. Nonetheless, we still expect that $\widehat{\Delta}$ scales as $1/\sqrt{L}$ for evolved communities.

For the evolution with $L$ roughly constant from exponential $\mathcal{P}(s)$, we see that $\widehat{\Delta}$ after evolution is actually larger than in an unconditioned matrix. However, here the bias has an extra contribution from the general fitness, so $\widehat{\Delta}$ is no longer as good an estimate of the bias distribution width (though it should still be similar).

We can compare these results to previous results of ***Bunin, 2016*** in which he calculated the statistics of the interaction matrix conditioned on assembly in the stable fixed-point phase where

niche interactions — large negative diagonal terms in $V$ — stabilize the diversity instead of spatial structure. Bunin observed that, conditional on assembly, correlations between the effect of strains $k$ and $l$ both on strain $i$ become negative, that is $\mathrm{Corr}[V_{ik}, V_{il}] < 0$ with small magnitude of order $1/K$. This is consistent with our observation that the distribution of the sum of these interactions for each strain, $\sum_j V_{ij}$, will have a smaller width than in the original ensemble. In addition, Bunin finds an $\mathcal{O}(1/K)$ shift in the mean of the interaction matrix toward less competition, which is consistent with $\widehat{\mu}_V > 0$. We also see slight but systematic changes in $\widehat{\gamma}$ and $\widehat{\sigma_V}$, particularly in the case of evolution of correlated mutants with $\rho = 0.95$.

Although many of the changes of the interaction statistics of assembled or evolved communities are statistically significant, they are mostly very small, with the possible exception of $\widehat{\mu}_V$ for $\rho = 0.95$. We conjecture that for $\rho = 0$ all the effects are small by some power of $L$. But for $\rho$ near one, this is less clear: whether for $L$ large compared to some inverse power of $1 - \rho$, the correlations induced by evolution will still be small, or whether the correlations will persist for arbitrarily large $L$, we leave as an open question.

# Appendix 5

## Dynamical mean field theory

In this section we provide some more background on the dynamical mean field theory used as a basis for the heuristics and scaling arguments throughout the paper. The main idea of dynamical mean field theory is to reduce an interacting many-body problem into a single-body stochastic problem, with the statistics of the stochasticity to be determined self consistently. As discussed in PAF, a dynamical mean field theory analysis of **Equation 1** results in autonomous stochastic integro-differential equations for each strain independently. Since the strains are statistically equivalent on each island, we drop the island subscript:

$$\frac{d\nu_i}{dt} = \nu_i \left( s_i + \zeta_i(t) + \gamma \int^t R(t,t')\nu_i(t')dt' - \Upsilon(t) \right) + m(\bar{\nu}_i(t) - \nu_i). \tag{A5.1}$$

Here, the correlation function $C(t,t')$ of $\zeta_i(t)$, the response function $R(t,t')$, and the island-averaged abundance $\bar{\nu}_i$, must be determined self-consistently. In order to use this description for the evolved (in addition to assembled) communities we have assumed that the statistics of the $V$ do not change substantially in evolved communities: we show in 'Appendix 4' that our simulation results are consistent with this *Ansatz*. Here, we have made the time-dependence in $\zeta_i(t)$ explicit — but elsewhere in the paper when we refer to the time-averaged or mean drive $\bar{\zeta}_i$, we have dropped the overbar for readability.

In ecological steady state, some of the strains will have gone extinct, and the correlations and responses of the persistent strains will only depend on time differences $t-t'$, and quantities will have well-behaved time averages equal to island averages, denoted by overbars. The average growth rate of strain $i$ on a single island excluding migration is

$$\bar{r}_i = s_i + \underbrace{\sum_j V_{ij}\bar{\nu}_j}_{\text{force from all strains}} - \bar{\Upsilon} = s_i + \underbrace{\zeta_i + \gamma X \bar{\nu}_i}_{\text{drive and feedback}} - \bar{\Upsilon} = \xi_i + \gamma X \bar{\nu}_i, \tag{A5.2}$$

where the static susceptibility, $X$, is the time-integrated total response function $X = \int_{-\infty}^{t} R(t,t')dt' = \sum_i \chi_i$. Here, we define the individual strain static susceptibility as $\chi_i \equiv \frac{d\bar{\nu}_i}{d\xi_i}$. In steady state with migration, $\langle \frac{d \log \nu_i}{dt} \rangle = 0$; therefore $\bar{r}_i < 0$ must be exactly compensated by the average effect of the migration term $m(\langle \bar{\nu}_i/\nu_i \rangle - 1)$, which is always positive (by the Cauchy-Schwarz inequality) and can be much larger than $m$ when $\nu_i$ is small.

The stochastic DMFT equations cannot be reduced to equations for the correlation and response function and must be treated directly. Their self-consistent solutions in the STC phase were analyzed in our previous work [PAF], by asymptotics in the large parameter $\mathcal{M}$. This analysis yields super-diffusive random walks of the log-abundances, persistence around the migration floor, and the statistics of the occasional blooms that occur for all persistent strains with negative bias. These aspects together determine how the mean abundance of a strain depends on its bias, via $\bar{\nu} \approx \mathcal{N}(\xi)$ with corrections smaller by factors of $\mathcal{O}(1/\sqrt{L})$.

When the bias of a strain is positive the average input from migration will be relatively small, and hence $\bar{r}_i \approx 0$ which implies that $\bar{\nu}_i \approx \xi_i/(-\gamma X)$. For strains with negative bias, the migration is essential and the statistics of the blooms, which are rarer the more negative the bias, make $\bar{\nu}_i$ exponentially small in $-\xi_i$ [PAF]. For strains with more negative bias, $\xi_i < \xi_c < 0$, the migration cannot sustain their local populations and they go globally extinct with their spatially averaged frequency decaying exponentially in time.

The distribution of the biases of a community plays a crucial role in determining its response to invasions. In a randomly assembled community of unrelated strains, the biases are essentially independent up to corrections smaller by $\mathcal{O}(1/\sqrt{L})$, with the mean drives, $\zeta_i$, gaussian distributed with average zero and standard deviation of order $1/\sqrt{L}$. More precisely, since $\xi_i$ is determined by the community in the absence of $i$, the $V_{ij}$ that determine it are independent of the abundances in the community. The variance of $\zeta$ is $1/\mathcal{L} \equiv \sum_j \bar{\nu}_j^2$ (with only small corrections from the neglect of the strain itself): this simple result is explicit in the self-consistency condition of the DMFT correlation function. We therefore use $\mathcal{L}$ as a measure of the *effective size* of a community which weights the contributions of strains by their mean abundances and, unlike $L$, is insensitive to whether the close to marginal strains have or have not gone extinct. The distribution of the biases of the persistent strains in the randomly assembled

community is a truncated gaussian with lower limit $\xi_c$, which is of order $1/\sqrt{\mathcal{L}}$ — the magnitude of $\xi_c$ and $\mathcal{L}$ must be determined self-consistently. Crucially, these will depend on the dynamics — especially the blooms — not just the mean quantities.

# Appendix 6

## Analysis of general fitnesses

### Number of persistent strains with exponential $\mathcal{P}(s)$ in perfectly antisymmetric model

The antisymmetric model with $\gamma = -1$ and no migration (considering only a single island) provides a situation in which we can calculate various quantities analytically and anchor the DMFT analysis. As analyzed in PAF, there is a unique uninvadable fixed point characterized by abundances $\{\nu_i^*\}$ with half the strains having gone extinct, and this fixed point is marginally stable. Indeed, there is a family of chaotic steady states, parametrized a temperature-like quantity $\Theta$ which is roughly the range of the fluctuations in $\log \nu_i$, with the average abundances $\bar{\nu}_i = \nu_i^*$.

Here we analyze the effect of introducing independent exponentially distributed $s_i$ for each strain, and find the number of strains that persist at the fixed point. Since there is no migration, the relationship between the biases and abundances is especially simple, as detailed below. The critical bias is $\xi_c = 0$ and the mean $s$ is simply $\hat{s} = \bar{\Upsilon}$. In the limit $K\Sigma^2 \gg 1$, the distribution of the drives with scale $1/\sqrt{K}$ will be much narrower than that of the $s_i$. Let us define $x_i \equiv \zeta_i + s_i$ for convenience. The scale of $p(x)$, the $x_i$ distribution, will be $\Sigma$ and it will decay exponentially for large $x$. We can get the distribution of $x$ from the convolution of an exponential and gaussian distribution, with the drives having variance $\sigma_\zeta^2$:

$$p(x) \approx \frac{e^{\sigma_\zeta^2/2\Sigma^2}}{\Sigma} e^{-x/\Sigma}, \quad \text{for } x > \sigma_\zeta \left( \frac{\sigma_\zeta}{\Sigma} + \mathcal{O}(1) \right). \tag{A6.1}$$

With the Lagrange multiplier $\bar{\Upsilon}$, the average abundances $\bar{\nu}_i$ at the mean field fixed point will be $\bar{\nu}_i = \max(0, \frac{x_i - \bar{\Upsilon}}{X}) = \max(0, \xi_i/X)$ with $X$ the total static susceptibility given by $\sum_i \chi_i$. Let us define $\phi$ as the fraction of strains that persists with positive bias. When $\bar{\Upsilon} > \sigma_\zeta[\sigma_\zeta/\Sigma + \mathcal{O}(1)]$, we can make the approximation

$$\phi = \int_{\bar{\Upsilon}}^{\infty} p(x)dx \approx e^{\sigma_\zeta^2/2\Sigma^2} e^{-\bar{\Upsilon}/\Sigma}. \tag{A6.2}$$

Self consistency requires that the average abundance over the initial $K$ strains is

$$\int_{\bar{\Upsilon}}^{\infty} \left( \frac{x - \bar{\Upsilon}}{X} \right) p(x)dx = \frac{1}{K} \implies e^{\sigma_\zeta^2/2\Sigma^2} \frac{\Sigma}{X} e^{-\bar{\Upsilon}/\Sigma} \approx \frac{1}{K} \implies \frac{\Sigma}{X}\phi \approx \frac{1}{K}. \tag{A6.3}$$

Now we combine this relation with the self consistency condition

$$X = \sum_i \frac{d\bar{\nu}_i}{d\xi_i} = \frac{K\phi}{X} \implies X = \sqrt{K\phi}, \tag{A6.4}$$

to obtain $\phi \approx \frac{1}{K\Sigma^2}$. Therefore, the number of persistent strains, when $K\Sigma^2 \gg 1$ is $\Sigma^{-2}$, with coefficient one.

To obtain the behavior for arbitrary $K$, one must solve the mean field self consistency equations, with the exact distribution

$$p(x) = \frac{1}{\Sigma} \exp\left( \frac{\sigma_\zeta^2 - 2x\Sigma}{2\Sigma^2} \right) \Phi\left( \frac{x\Sigma - \sigma_\zeta^2}{\Sigma\sigma_\zeta} \right), \tag{A6.5}$$

where $\Phi$ is the standard normal cumulative distribution function. There are three mean field equations for the three unknowns $\bar{\Upsilon}$, $\sigma_\zeta$ and $X$.

$$\frac{1}{K} = \int_{\bar{\Upsilon}}^{\infty} \left( \frac{x - \bar{\Upsilon}}{X} \right) p(x)dx, \tag{A6.6}$$

$$\frac{\sigma_\zeta^2}{K} = \int_{\bar{\Upsilon}}^{\infty} \left( \frac{x - \bar{\Upsilon}}{X} \right)^2 p(x)dx, \tag{A6.7}$$

$$X = \frac{K}{X}\int_{\bar{\Upsilon}}^{\infty} p(x)dx. \quad \text{(A6.8)}$$

The number of persistent strains is $K\int_{\bar{\Upsilon}}^{\infty} p(x)dx$, where $\bar{\Upsilon}$ solves the above equations. Since for perfectly antisymmetric interactions, $\hat{s} = \bar{\Upsilon}$, the solution of these allows us to check that $\hat{s}/\Sigma$ increases with $\Sigma$, as seen in ***Figure 3***. Though one could solve the mean field equations numerically, we can obtain asymptotic results using the approximation in ***Equation A6.1***.

For $K\Sigma^2 \gg 1$, we have $\bar{\Upsilon} \approx \Sigma(1 + \log(K\Sigma^2))$, $X \approx 1/\Sigma$ and $\sigma_\zeta \approx \sqrt{2}\Sigma$, which is consistent with the asymptotic *Ansatz* for $p(x)$ for the present strains: since $\bar{\Upsilon} - \sigma_\zeta^2/\Sigma \gg \sigma_\zeta$ all persistent strains have large enough $x$ for the approximation to be valid. Then the distribution of $\zeta$ for the *persistent* strains is

$$p(\zeta|\text{persistent}) \approx \frac{1}{2\sqrt{\pi}\Sigma} e^{-(\zeta/2\Sigma - 1)^2}, \quad \text{(A6.9)}$$

valid except for $\zeta > \bar{\Upsilon} = \Sigma(1 + \log(K\Sigma^2))$ which is a negligible fraction of the distribution. The width of both the $\zeta$ and the $s$ distributions for the persistent strains is of order $\Sigma$. As expected more generally, these are comparable once $\hat{s}$ is in the tail of $\mathcal{P}(s)$. This calculation could be done for any choice of $\mathcal{P}(s)$, and should give the behavior of $\bar{\Upsilon}$ with $K$, which is the same as the behavior of $\hat{s}$ with $T$ in the evolving phase of the STC: namely $\hat{s} \sim \Sigma(\log T)^{1/\psi}$.

## Joint distribution of $\zeta$ and $s$ of successful invaders

Here, the statistical properties of uncorrelated invaders, needed to understand the evolution of the STC state, are analyzed. Every successful invader $A$ must have $\xi_A > \xi_c$. Since the invader's bias is a sum of $s_A$, $\zeta_A$ and $-\bar{\Upsilon}$, we can calculate the distributions of $s_A$ and $\zeta_A$ conditional on invasion.

We analyze the case with an exponential distribution of the $s_i$ with scale $\Sigma$. Since the distribution of the invader $\zeta_A$ and $s_A$ are independent, we can write their joint distribution as

$$p(\zeta_A, s_A) = \frac{1}{\Sigma}\sqrt{\frac{\mathcal{L}}{2\pi}} \exp\left[-\frac{s_A}{\Sigma} - \frac{\mathcal{L}}{2}\zeta_A^2\right]\Theta(s_A). \quad \text{(A6.10)}$$

Conditioning on successful invasion multiplies by a factor of $\Theta(\zeta_A + s_A - \xi_c - \bar{\Upsilon})$, which enforces that $\xi_A > \xi_c$. We are interested in the large time limit when $\bar{\Upsilon} = \hat{s} + \mathcal{O}(\Sigma) \gg \Sigma$. In this limit, the restriction that $s_A > 0$ has negligible effect and the analysis is simple.

The invasion probability is

$$p_{\text{invade}} = \int_0^{\infty}\int_{-\infty}^{\infty} p(\zeta_A, s_A)\Theta(\zeta_A + s_A - \xi_c - \bar{\Upsilon})d\zeta_A ds_A \approx \exp\left[\frac{1}{2\Sigma^2\mathcal{L}} - \frac{\bar{\Upsilon} + \xi_c}{\Sigma}\right], \quad \text{(A6.11)}$$

which decays exponentially as $e^{-\hat{s}/\Sigma}$, as expected. The marginal distribution of $s_A$ conditioned on invasion is

$$p(s_A|\text{invasion}) \approx \frac{1}{\Sigma p_{\text{invade}}} e^{-s_A/\Sigma}\Phi\left[\sqrt{\mathcal{L}}(s_A - \bar{\Upsilon} - \xi_c)\right], \quad \text{(A6.12)}$$

where $\Phi$ is the standard normal cdf. Similarly, the distribution of $\zeta_A$ conditioned on invasion is

$$p(\zeta_A|\text{invasion}) \approx \frac{1}{p_{\text{invade}}}\sqrt{\frac{\mathcal{L}}{2\pi}} \exp\left[\frac{\zeta_A}{\Sigma} - \frac{\mathcal{L}}{2}\zeta_A^2 - \frac{\xi_c + \bar{\Upsilon}}{\Sigma}\right], \quad \text{(A6.13)}$$

and the bias distribution conditioned on invasion is

$$p(\xi_A|\text{invasion}) \approx \frac{1}{\Sigma}\exp\left[\frac{-(\xi_A - \xi_c)}{\Sigma}\right]\Theta(\xi_A - \xi_c). \quad \text{(A6.14)}$$

Of particular interest is the mean $s_A$:

$$\text{E}(s_A|\text{invasion}) \approx \bar{\Upsilon} + \xi_c + \Sigma - \frac{1}{\Sigma\mathcal{L}} \approx \hat{s} + \mathcal{O}(\Sigma), \quad \text{(A6.15)}$$

since $\mathrm{E}(\zeta_A|\mathrm{invasion}) \approx 1/(\Sigma\mathcal{L})$ and $\mathrm{E}(\xi_A|\mathrm{invasion}) \approx \xi_c + \Sigma$.

## Effects of shape of tail of general fitness distribution

Here, we analyze the evolutionary dynamics for the class of general-fitness distributions with high-$s$ tails parametrized by $\psi$, and show support from numerical simulations for the scaling *Ansatz* and results given in the main text.

For attempted invasions by unrelated strains, as the evolution progresses, $\hat{s}$ increases past $\Sigma$ and pushes up into the tail of $\mathcal{P}(s)$. The rate of successful invasions then decreases rapidly since invaders need general fitness (at least) comparable to $\hat{s}$ in order to invade: the probability of successful invasion is of order $p_{\mathrm{invade}} \sim \widehat{\Sigma}\mathcal{P}(\hat{s})$ with the prefactor from the width of the distribution of $s_A$ near $\hat{s}$. In this regime, the analysis of 'Appendix 6' can be carried over with $\Sigma$ replaced by $\widehat{\Sigma}$. The drive on the invader from the extant community, $\zeta_A \sim 1/\sqrt{L}$, will be of order $\widehat{\Sigma}$, yielding the key scaling prediction $L \sim \widehat{\Sigma}^{-2}$ which is tested in **Appendix 6—figure 1A**.

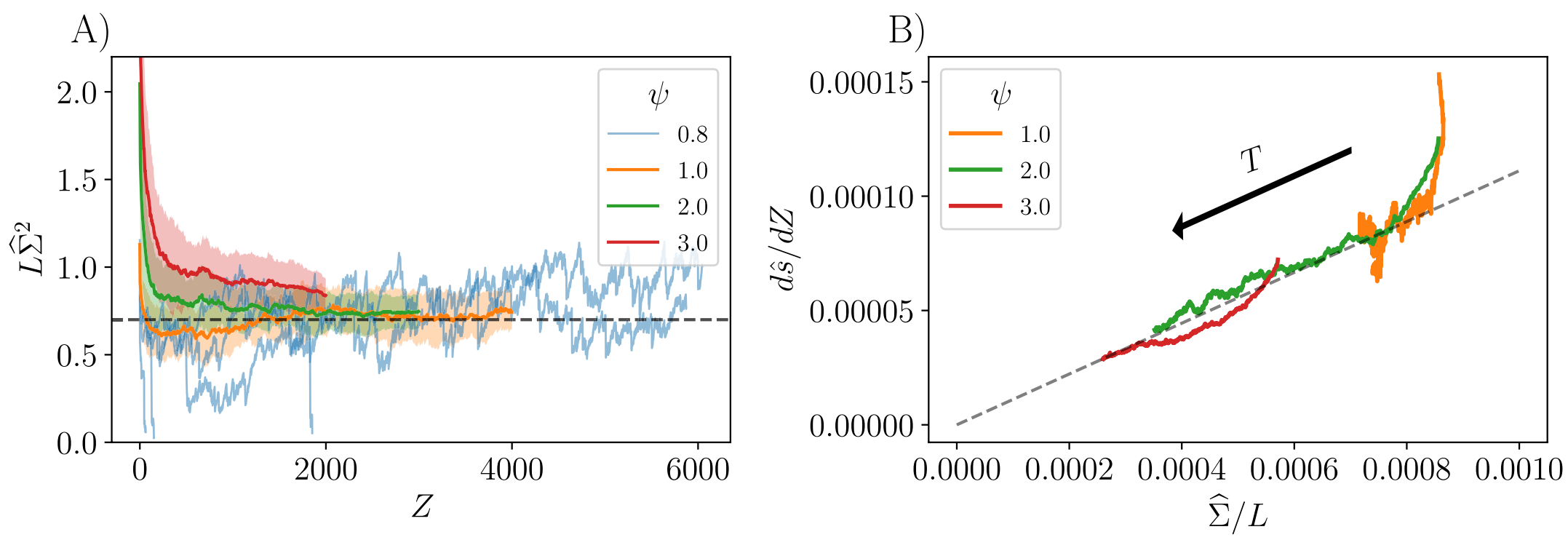


**Appendix 6—figure 1.** Consistency of predicted scaling *Ansatz* with simulations. (**A**) The combination $L\widehat{\Sigma}^2$ is predicted to approach a constant independent of $\psi$ for long evolutionary times. Solid lines indicate the mean value and shaded regions indicate standard error over 50 replicates, conditional on the diversity not crashing. The curves for $\psi = 0.8$ are shown individually as this value of $\psi$ results in crashing and the fluctuations are large. The dashed horizontal line is at 0.7, the value found for $\psi = 1$ from the data of **Figure 3A**. Note that for $\psi = 3$, transients are still substantial. (**B**) Theory predicts that $d\hat{s}/dZ$ scales as $\widehat{\Sigma}/L$. The dashed line has slope 1/9 with evolutionary time increasing along the direction of the arrow. Data for $\psi = 0.8$ are quite noisy and not shown. Here the curves are smoothed by a moving average over 1000 successful invasions for both $d\hat{s}/dZ$ and $\widehat{\Sigma}/L$, and further averaged over 50 replicates conditional on not crashing. Transients from the initial conditions are observable at the upper right.

Successful invaders will have $s_A = \hat{s} + \mathcal{O}(\widehat{\Sigma})$, since $\widehat{\Sigma}$ is the typical amount by which the invader's general fitness is likely to be larger (or smaller) than the $\hat{s}$ of the extant community. Therefore, we expect that the change in the community-mean $\hat{s}$ per successful invasion will be $d\hat{s}/dZ \sim \widehat{\Sigma}/L$ (obtainable from 'Appendix 6'): this scaling is tested in **Appendix 6—figure 1B**. Substituting for $L \sim \widehat{\Sigma}^{-2}$, we have $d\hat{s}/dZ \sim \widehat{\Sigma}^3$. Integrating this equation and using $\widehat{\Sigma} = \Sigma^{\psi}/\hat{s}^{\psi-1}$ one obtains scaling laws quoted in the main text, valid for $Z \gg 1/\Sigma^2$ and $L \sim Z^{\frac{2\psi-2}{3\psi-2}}$ with the latter exponent less than one for the relevant range of $\psi$.

For $\psi > 1$, both $L$ and $\hat{s}$ increase sub-linearly with $Z$, but $dL/dZ$ decreases steadily with $Z$ since the evolution gets closer to an average of one extinction per successful invasion, which obtains exactly in the steady state for the exponential distribution. By contrast, for $\psi < 1$, $L$ decreases with $Z$ and $\hat{s}$ increases super-linearly with $Z$, as the distribution does not decay fast enough to prevent increasingly fit mutants from emerging and outcompeting more than one strain per successful invasion.

The behaviors of the scaling combination $L\widehat{\Sigma}^2$ is shown versus $Z$ in **Appendix 6—figure 1A**. this combination is predicted to approach a $\psi$-independent constant for large $Z$. The value of this constant is consistent with the 0.7 found for $\psi = 1$ in **Figure 3A**. The other key scaling law, $d\hat{s}/dZ \sim \widehat{\Sigma}/L$, is tested in **Appendix 6—figure 1B**, with the predicted coefficient of $\cong 1/9$ from **Figure 3**. Although these show convincing evidence for the predicted asymptotic scaling forms, much of the observed evolution is not yet in the late-time asymptotic regime.

To get the growth of $\hat{s}$ — and hence the other quantities — with the number of attempted invasions, we write; $d\hat{s}/dT \sim p_{\text{invade}}\widehat{\Sigma}/L$ and integrate this using the scaling relations to get

$$\frac{\hat{s}}{\Sigma} \approx \left[\psi \log(T\Sigma^2) + (3 - 3\psi)\log\log(T\Sigma^2) + \mathcal{O}(1)\right]^{1/\psi}, \tag{A6.16}$$

with the unknown $\mathcal{O}(1)$ correction reflecting an additive uncertainty of order $\widehat{\Sigma}$ in $\hat{s}$. Up to this correction, the expression has an identical *form* to the upper limit from $p_{\text{invade}}T \geq L$ as explained in the main text, and quoted there without the $\log\log$ correction (which vanishes for $\psi = 1$). This prediction, with $T$ replaced by $T + K$ (without the $\log\log$ part or the $\mathcal{O}(1)$ correction), is plotted in ***Figure 6B***. Slow crossovers from the initial non-universal behavior towards the predicted behaviors are observable with the asymptotic predictions thus not well-testable in the time-dependences.

# Appendix 7

## Correlated general fitness mutations

We have shown that including general fitnesses with $\psi \geq 1$ causes evolution with unrelated invaders to slow down, with $Z$ increasing only as power of $\log T$. Here we analyze what happens if the general fitnesses of mutants are correlated with those of their parent. In order to generate such mutant general fitnesses that are still drawn from the overall $\mathcal{P}(s)$, we can take the mutant fitness to be sampled from a Markov chain starting at the parent, with stationary measure given by $\mathcal{P}(s)$, and a “time” between samples that depends on the desired correlation between mutant and parent fitnesses.

### Exponential distribution of $s$

The case of exponential $\mathcal{P}(s)$ is simple: one can choose the mutant fitness to be gaussian distributed with mean given by the parent's general fitness minus a constant offset which can be tuned so that the stationary distribution is exponential with the desired $\Sigma$. The variance of the gaussian determines the degree to which the parent and mutant are correlated, parametrized by $\rho_s$, which need not be the same $\rho$ defined by the correlations between parent and mutant interactions. The crucial difference from uncorrelated $s$ is that the probability of $s \gtrsim \hat{s}$ is now independent of $\hat{s}$ rather than decreasing exponentially with $\hat{s}/\Sigma$. This enables $\hat{s}$ and $Z$ to increase linearly in $T$. However, $L$ still saturates to a value of order $1/\Sigma^2$, with a coefficient that depends on the correlations. ***Appendix 7—figure 1*** shows simulations that confirm these predictions.

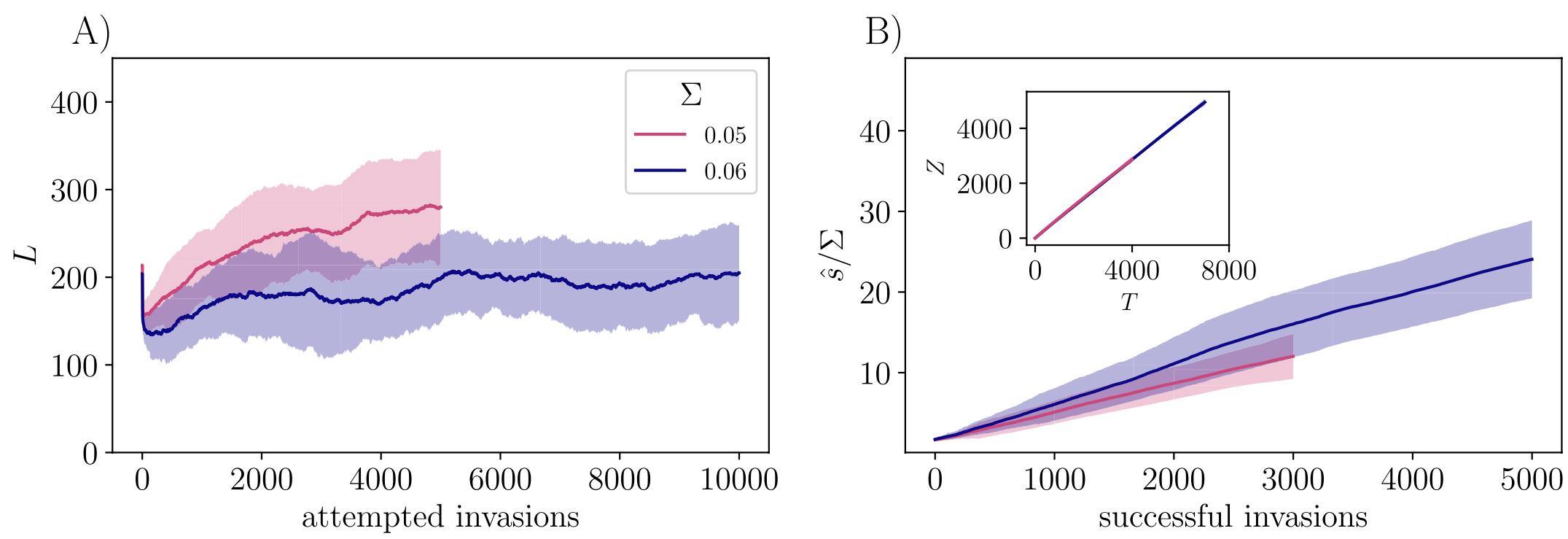


**Appendix 7—figure 1.** Simulation results for correlated mutants with an exponential $\mathcal{P}(s)$. The primary difference from the independent-invaders case (***Figure 3***) is that the number of successful invasions is proportional to the number of attempted invasions, since a mutant $s$ is correlated with that of its parent, so invasions do not slow down with increasing $\hat{s}$. Here $\rho = 0.8$ for the interactions and for the general fitnesses the correlation is $\rho_s \approx 0.9$. Note that for the same $\Sigma$ with unrelated invaders ($\rho = 0$ and $\rho_s = 0$) the steady state values of $L$ are quite similar to those shown here.

### Non-exponential $s$ distributions with correlated mutants

Incorporating correlations in the general fitnesses of parent and mutant for other distributions of $s$ gives rise to other complications. In the $\psi > 1$ case of particular interest, as $\hat{s}$ grows the mean $s - \hat{s}$ of mutants becomes more and more negative and overwhelms the random part of $s_M - s_P$. There is a then a scale of $s$ that diverges as $\rho_s \to 1$, such that for $\hat{s}$ above this scale, successful invasions become very unlikely and the diversification slows down even with strongly correlated mutants. This gives rise to complicated crossovers that we do not fully understand and are hard to disentangle in the simulations.

# Appendix 8

## Dynamics of drive distribution

### Markov approximation

In ‘Evolution without general fitness differences’, we describe how the serial invasion of strains causes the biases of extant strains to undergo a random walk with an absorbing boundary condition. In general the statistics of this random walk are complicated and depend on the conditioning on evolutionary history. In order to make analytical progress, we make a Markov approximation of the effects of the evolutionary history and assume that the distribution of the biases at epoch $T+1$ only depends on itself at epoch $T$. The average drive of probe strain 0 before the successful addition of strain $A$ is $\zeta_0 = \sum_j V_{0j}\bar{\nu}_j$, while after the invasion it changes to

$$\zeta_0' = \zeta_0 + \delta\zeta_0 = \sum_j V_{0j}(\bar{\nu}_{j\backslash A} + \delta\bar{\nu}_j) + V_{0A}\bar{\nu}_A, \tag{A8.1}$$

where $\bar{\nu}_A$ is the mean abundance at which strain $A$ establishes and $\bar{\nu}_{j\backslash A}$ is the average abundance of strain $j$ before $A$ invades. The sum on $j$ does not include 0 or $A$ and the $\delta$'s denote changes that result from the invasion. Since $\bar{\nu}_A$ is of order $1/L$, its *direct* effect on the drive is smaller than $\zeta_0$ by a factor of $1/\sqrt{L}$. However $A$ will also change all of the other biases by similar-magnitude random amounts with uncorrelated signs, causing $\delta\bar{\nu}_j \approx \chi_j\delta\zeta_j$. Thus $\mathrm{E}[\delta\zeta_0^2] = \bar{\nu}_A^2 + \sum_j V_{0j}^2\chi_j^2\delta\zeta_j^2$ with the expectation over the interactions with the probe strain. Since the properties of the extant strains depend neither on the $V_{0j}$ nor the perturbations via $V_{jA}$, the averages over each of the squared factors can be performed separately, leaving $\sum_j \chi_j^2$ times the average over $j$ of $\delta\zeta_j^2$. But now we can consider each strain separately to be the probe strain so that the average over the extant strains must be the same on both sides. This yields $\mathrm{E}_{\mathrm{strains}}[\delta\zeta_0^2] = \bar{\nu}_A^2/(1-\sum_j\chi_j^2)$. The more directly measurable quantity is the total squared abundance change of the extant strains caused by a random change in growth rate of each. With the perturbation caused by the invader, we have $\sum_j(\delta\bar{\nu}_j)^2 = \Xi\bar{\nu}_A^2$ in terms of the nonlinear response, which we call the *fragility*: $\Xi = \sum_j\chi_j^2/(1-\sum_j\chi_j^2)$. This quantity is analogous to the spin-glass susceptibility of random magnets **Fisher and Huse, 1988**. The fragility is of order unity and the form of the denominator shows that it can diverge: such divergence indicates an instability of the community and breakdown of the DMFT *Ansatz*. Note that the fragility is a general measure of the sensitivity of a community to perturbations. In the stable niche-phase of the Lotka Volterra model with large negative diagonal interactions of magnitude $Q = -\mathrm{E}[V_{ii}]$, the fragility diverges as $K$ increases to the stability boundary at $K_{\mathrm{crit}} \sim Q^2$, indicating an instability in the mean field solution **Bunin, 2017**. The fragility is infinite in the perfectly antisymmetric model (where $Q=0$), corresponding to being exactly marginal and highly sensitive to added strains.

In addition to the random change of a $\zeta_j$ induced by a successful invader, there is also a systematic change, as seen in **Figure 4**. This is because $\delta\zeta_0$ and $\zeta_0$ involve the same set of random $V_{0j}$ and are thus correlated. We can use the fact that $\mathrm{E}[X|Y=y] = \frac{\mathrm{Cov}(X,Y)}{\mathrm{Var}(Y)}y$ for zero mean gaussian random variables $X$ and $Y$ to see that the conditional expectation is $\mathrm{E}[\delta\zeta_0|\zeta_0] = \zeta_0\sum_j\bar{\nu}_j\delta\bar{\nu}_j/\sum_j\bar{\nu}_j^2$. Since each $\bar{\nu}_j$ and $\delta\bar{\nu}_j$ are correlated as a consequence of $\zeta_j$ and $\delta\zeta_j$ being correlated, we again have to self-consistently determine this correlation. Unfortunately this is more complicated as the $\delta\bar{\nu}_j$ also have a contribution from small changes in the function $\mathcal{N}(\zeta)$ caused by the invaded strain. However one can understand the *form* of the conditional expectations by noting that after the new strain has invaded,

$$\mathcal{L}(T+1)^{-1} = \mathcal{L}(T)^{-1} + \sum_j(\delta\bar{\nu}_j)^2 + 2\sum_j\bar{\nu}_j\delta\bar{\nu}_j. \tag{A8.2}$$

This implies that the last term must be negative if $L$ and $\mathcal{L}$ are increasing or staying constant. Therefore the correlation between $\zeta_j$ and $\delta\zeta_j$ caused by the invader strain is negative (this is true also if only considering the direct effect of $A$ on $j$). The additional effects on $j$ from the changes in other strains is enhanced by the fragility, $\Xi$. If the fragility is high, one can show that $2\sum_j\bar{\nu}_j\delta\bar{\nu}_j$ is proportional to $-\Xi\,\mathrm{E}[\nu_A^2]$, since the last two terms in **Equation A8.2** should be of similar magnitude to preserve $\mathcal{L}$: in this limit, as argued below, the diversity will decrease.

The above Markovian analysis is an approximation because $\delta\zeta_j$ for each extant strain is correlated with its whole past trajectory — as it involves many of the same $V_{0j}$ — and its conditional expectation will be a weighted sum over the full past $\zeta_j(T')$. Furthermore, it is possible that conditioning on all this past could suppress $\mathrm{E}[\delta\zeta_j^2]$ by a substantial factor (in the Markovian approximation the analogous correction is smaller by a $1/L$ factor). However the basic structure of the stochastic changes is correctly

captured by the Markovian approximation. This approximation is summarized in ***Equation 3***, which writes the change in bias after one invasion attempt as the sum of a systematic and stochastic part. If neither extinctions nor invasions occurred, there would be a trivial steady state with a gaussian distribution of the biases with width $\sim 1/\sqrt{L}$ as expected for an assembled community. With a source of new strains, which come in with drives that are gaussian distributed with width $1/\sqrt{\mathcal{L}}$, the effective community size is $\mathcal{L} \approx CL$ with $C < 1$. We thence obtain a Fokker Planck equation for the number density $N$ of the drives:

$$\partial_T N(\zeta, T) = B\partial_\zeta \left[\frac{\zeta}{L} N(\zeta, T)\right] + \frac{D}{L^2}\partial_\zeta^2 N(\zeta, T) + \sqrt{\frac{CL}{2\pi}}\, e^{-\zeta^2 CL/2}\Theta(\zeta - \xi_c(T) - \bar{\Upsilon}(T)), \quad \text{(A8.3)}$$

where we impose the absorbing boundary condition at the critical bias corresponding to $N(\xi_c + \bar{\Upsilon}, T) = 0$ with $\xi_c$ and $\bar{\Upsilon}$ both scaling as $1/\sqrt{L}$. The last term in ***Equation A8.3*** is the truncated-gaussian drive distribution of successfully invading strains with scale $1/\sqrt{\mathcal{L}}$. The fraction of invasions that are successful, $dZ/dT$, is just the integral of the truncated gaussian, $1 - \Phi[\sqrt{\mathcal{L}}(\xi_c + \bar{\Upsilon})]$. The number of strains, $L(T)$, is changing and at this point unknown: it is determined self-consistently from $L(T) = \int d\zeta N(\zeta, T)$. It is straightforward to show, as we do in 'Appendix 8', that this Fokker-Planck equation admits a scaling solution with the *Ansatz* that all the quantities scale as $1/\sqrt{L}$. The rate of diversification, $U = dL/dT$, is of order one and determined by an eigenvalue-like condition: it can be either positive or negative depending on $\xi_c + \bar{\Upsilon}$ and the coefficients, $B, D, C$. If the community is very fragile ($\Xi$ large), then the $B$ and $D$ terms will dominate over the input and the loss of strains per successful invasion will be large: in this regime the diversification rate is negative. On the other hand if $-(\xi_c + \bar{\Upsilon})\sqrt{\mathcal{L}}$ is large, few strains will go extinct and the community will diversify. Of course, these quantities are determined self-consistently, depending on both the distribution of biases and the function $\mathcal{N}(\xi)$, which itself will evolve, reaching a scaling form in the steadily diversifying state.

How good is the Markov approximation? The average change in the drive should really be a weighted integral over the history of the drive, $\zeta(T')$, over all $T' < T$. If one makes the *Ansatz* of a scaling solution with $L$ steadily increasing, then for large $T$, $L \sim T$ and in the variables scaled by $T$ the weighting function should be a function of of $q = \log(T/T')$, so that the integral over the history is a convolution in $q$. In general, the steady state could, of course, not be found even if this weighting function were known, and the integro-differential generalization of the Fokker-Planck equation would have to be analyzed numerically. But with the widths of the distributions of the extant and invader biases being similar, as observed, the details of the weighting function might not much affect the bias distribution.

## Fokker-Planck equation without general fitness differences

Without general fitness differences or correlations, the evolution of the average drive of a strain in the Markovian approximation obeys a Langevin equation (***Equation 3***). With a source term corresponding to incoming strains, this yields a Fokker Planck equation (***Equation A8.3***) for the number density $N(\zeta, T)$ whose integral at any time gives us the number of strains. The crucial *Ansatz* is that all the quantities ($\zeta$, $\xi_c$ and $\bar{\Upsilon}$) scale as $1/\sqrt{L}$, and we now take $L$ to be a function of $T$. Then we can hypothesize a scaling solution $N(\zeta, T) \sim L^{3/2} g(\zeta\sqrt{L})$ for some function $g$ so that $\int N(\zeta, T) d\zeta \sim L(T)$, as the integral over the distribution of drives gives us the total number of strains. For ease of notation, we define the similarity variable $u \equiv \zeta\sqrt{L}$, and derivatives with respect to $u$ by primes: then ***Equation A8.3*** becomes

$$\frac{dL}{dT}\left(\frac{3}{2}g(u) + \frac{1}{2}ug'(u)\right) = Dg''(u) + \sqrt{\frac{C}{2\pi}}e^{-Cu^2/2}\Theta(u - u_0) + Bg(u) + Bug'(u), \quad \text{(A8.4)}$$

where we define $u_0 \equiv \sqrt{L}(\xi_c + \bar{\Upsilon})$. We can make an *Ansatz* of the form

$$L(T) = UT + L_0, \quad \text{(A8.5)}$$

where the coefficient $U$ is the average rate of increase or decrease of the diversity per invasion attempt.

The equation for $g(u)$ then becomes

$$\left(B - \frac{3U}{2}\right)g + \left(B - \frac{U}{2}\right)ug' + Dg'' + \sqrt{\frac{C}{2\pi}}e^{-Cu^2/2}\Theta(u - u_0) = 0. \quad \text{(A8.6)}$$

We can solve this equation numerically with boundary conditions $g(u_0) = 0$ and $g(\infty) = 0$ (exact solutions are available for certain values of $B, D$ and $U$). The first of these boundary conditions comes from the extinction criterion that enforces $\zeta_i > \xi_c + \bar{\Upsilon}$ for every extant strain, and second comes from the need to have $L < \infty$. The solution depends on the parameters $u_0$, $B$, $D$ and $U$. By enforcing $\int_{u_0}^{\infty} g(u)du = 1$, we can find the value of $U$ in terms of $u_0, B$ and $D$. One can see, numerically, that for $B$ and $D$ both large, the consistent value of $U$ is negative, implying loss of diversity.

In ***Figure 5B***, we use the measured values of $U$ and $C$, and adjust the values of $B$ and $D$ to give a normalized function $g(u)$. The blue curve is a truncated unit-variance gaussian for the initial assembled community, with mean and lower limit fit by hand, and the orange curve is a numerical solution to ***Equation A8.6*** with $U = dL/dT \cong 0.24$ and $B = 2.11$, $D = 1.79$ adjusted to enforce normalization. Note that $C = \mathcal{L}/L \cong 0.74$. The mode of the truncated gaussian is $\cong \bar{\Upsilon}\sqrt{\mathcal{L}}$ as expected from ***Figure 4***. The bias distributions for assembled and evolved communities are very similar except near $\xi_c$, which is what one expects from the Markov approximation. For $-\zeta_c = -\xi_c - \Upsilon \gg 1/\sqrt{\mathcal{L}}$, when the ratio of extant to invading widths, $\sqrt{DC/B} \cong 0.79$, is close to unity, the bulk of the drive distribution is close to gaussian with mean zero, which is reflected in ***Figure 5B***.

## Fokker-Planck equation with independent general fitness differences

If incoming strains have independently drawn general fitnesses, $s_i$, then we can write an evolution equation for the joint distribution of the drives and the $s_i$:

$$\partial_Z N(\zeta, s, Z) = B\partial_\zeta \left(\frac{\zeta}{L} N\right) + \frac{D}{L^2}\partial_\zeta^2 N + \mathcal{P}_>(s)\sqrt{\frac{CL}{2\pi}} e^{-\zeta^2 CL/2}\Theta(\zeta + s - \xi_c - \bar{\Upsilon}), \tag{A8.7}$$

where we abbreviate $\mathcal{P}(s|s > \xi_c + \bar{\Upsilon} - \zeta)$ by $\mathcal{P}_>(s)$. However, unlike the case with general fitness differences, now $\bar{\Upsilon}$ does not scale as $1/\sqrt{L}$. Indeed, the scaling of $\bar{\Upsilon}$ with $Z$ determines how $L$ depends on $Z$. If we specify the form of $\frac{d\bar{\Upsilon}}{dZ}$, then we can look for solutions which travel up in the $s$ direction at the same speed as $\bar{\Upsilon}$.

We can define similarity variables $u \equiv \zeta\sqrt{L}$ and $v \equiv \sqrt{L}(s - \bar{\Upsilon})$. Then we look for solutions of the form $N(\zeta, s, Z) \sim L^2 g(u, v)$ so that the integral of $N$ over $\zeta$ and $s$ is proportional to $L$. The boundary condition is that $g$ vanishes along the line $u + v = u_{\text{crit}}$. Plugging this into our Fokker Planck equation, and defining $u_{\text{crit}} \equiv \sqrt{L}\xi_c$, we obtain

$$\frac{dL}{dZ}\left(2g + \frac{u}{2}g_u + \frac{v}{2}g_v\right) - \frac{d\bar{\Upsilon}}{dZ}L^{3/2}g_v = Dg_{uu} + \sqrt{\frac{C}{2\pi}}\mathcal{P}_>(v)e^{-Cu^2/2}\Theta(u + v - u_{\text{crit}}) + Bg + Bug_u, \tag{A8.8}$$

where subscripts denote derivatives of $g$. We can see that the $\mathcal{P}_>(s)$ distribution generically breaks the scaling — because it depends on $s = v/\sqrt{L}$, not only on $v$. But we can look for solutions which obey the scaling with $1/\sqrt{L}$. Take $v/\sqrt{L}$ as exponentially distributed with scale $\Sigma$. Then in order for our scaling *Ansatz* to be valid, the quantity $\frac{v}{\Sigma\sqrt{L}}$ must be independent of $L$, so we must have $\sqrt{L} \sim 1/\Sigma \implies L \sim \Sigma^{-2}$. This requires $\frac{d\bar{\Upsilon}}{dZ} \sim \Sigma^3$ in order to make the left hand side of ***Equation A8.8*** independent of $L$. Furthermore, we have $dL/dZ \to 0$ since $L \sim \Sigma^{-2}$. Therefore, we have

$$\frac{dL}{dZ} = \frac{g_v + Dg_{uu} + \sqrt{\frac{C}{2\pi}}\mathcal{P}_>(v)e^{-Cu^2/2}\Theta(u + v - u_{\text{crit}}) + Bg + Bug_u}{2g + \frac{u}{2}g_u + \frac{v}{2}g_v}. \tag{A8.9}$$

Setting this to 0 would allow us to solve for $g(u, v)$ and therefore find the scaling solution for the joint distribution of mean drives and the $s_i$.

# Appendix 9

## Coexistence of correlated strains

The simulations show that even strains which interact very similarly with the rest of the community can coexist. Here, we present some of the numerical data and show how simple approximations fail to explain this behavior.

### Lotka-Volterra approximation ignoring migration

A natural way to understand very closely related strains (e.g. a mutant and parent) is via an effective Lotka Volterra model for the dynamics of just these two strains in the presence of all the others. For a correlation $\rho$ between the parent and mutant interactions, one can obtain a pair of DMFT equations describing the correlated dynamics of parent and mutant in the form of ***Equation A5.1***. Here, we consider the crude approximation of neglecting the effects of migration and averaging the dynamics of $\log \nu$ to obtain equations for parent and mutant frequencies $\nu_P$ and $\nu_M$:

$$\dot{\nu_P} = \nu_P[\xi_P - (-\gamma X)(\nu_P + \rho\nu_M)] \tag{A9.1}$$

$$\dot{\nu_M} = \nu_M[\xi_M - (-\gamma X)(\nu_M + \rho\nu_P)]. \tag{A9.2}$$

This analysis is appropriate only for strains with positive bias — and even then approximate – but the following results provide a null model against which one can compare the dynamics of parent-mutant replacement in the evolving STC.

In the absence of the mutant, $\bar{\nu}_P \approx \xi_P/(-\gamma X)$ (which is the correct behavior for large positive $\xi_P$). The condition for invasion of a mutant into an environment containing only the parent is that its invasion eigenvalue, suppressed by the effects of the parent, is greater than the critical bias which amounts to $\xi_M - \rho\xi_P > \xi_c$ and the condition for invasion of the parent into the mutant is likewise $\xi_P - \rho\xi_M > \xi_c$. In terms of the drives, these conditions are $\zeta_M - \rho\zeta_P > (1-\rho)\bar{\Upsilon} + \xi_c$ for the mutant to invade the parent, and the same condition with $M$ and $P$ flipped for the parent to invade the mutant. With the mutant drive parameterized as $\zeta_M = \rho\zeta_P + \sqrt{1-\rho^2}Z_1$ with $s_i$ a gaussian random variable with mean 0 and scale $\zeta_P$, the probabilities of coexistence between parent and mutant, and of replacement of the parent, are

$$p_{\text{coexist}} = \Phi\left[\frac{(1-\tilde{\xi}_c)/\rho - \rho + \tilde{\Upsilon}(1-1/\rho)}{\sqrt{1-\rho^2}}\right] - \Phi\left[\frac{\tilde{\Upsilon}(1-\rho)+\tilde{\xi}_c}{\sqrt{1-\rho^2}}\right] \tag{A9.3}$$

$$p_{\text{replace}} = 1 - \Phi\left[\frac{(1-\tilde{\xi}_c)/\rho - \rho + \tilde{\Upsilon}(1-1/\rho)}{\sqrt{1-\rho^2}}\right] \tag{A9.4}$$

where $\Phi$ is the standard normal cdf, and $\tilde{\Upsilon}$ and $\tilde{\xi}_c$ are the Lagrange multiplier and critical bias, both normalized by the scale of the $\zeta$ distribution. Note that the expressions are only valid for $\tilde{\Upsilon} < 1 - \frac{\tilde{\xi}_c}{1-\rho^2}$: otherwise, in this approximation, the coexistence probability vanishes and all invasions are replacements. The probability of mutant invasion is $p_{\text{invade}} = 1 - \Phi\left[\frac{\tilde{\Upsilon}(1-\rho)+\tilde{\xi}_c}{\sqrt{1-\rho^2}}\right]$. For $1-\rho \ll 1$ and $\tilde{\xi}_c = 0$, the invasion probability is $p_{\text{invade}} \approx \frac{1}{2}(1 - \tilde{\Upsilon}\sqrt{(1-\rho)/\pi})$ and the coexistence probability predicted by this simple approximation is $p_{\text{coexist}} \approx (1-\tilde{\Upsilon})\sqrt{(1-\rho)/\pi}$. With $\rho = 0.99$ and $\tilde{\Upsilon} = 0$ (in the STC $\tilde{\Upsilon}$ should be nonnegative, and setting it to 0 maximizes the coexistence probability), this predicts that the probability of coexistence *conditional on the mutant invading* is $p_{\text{coexist}}/p_{\text{invade}} \cong 0.12$ which is much smaller than the observed $p_{\text{coexist|invade}} > 0.5$ (***Appendix 9—figure 1***).

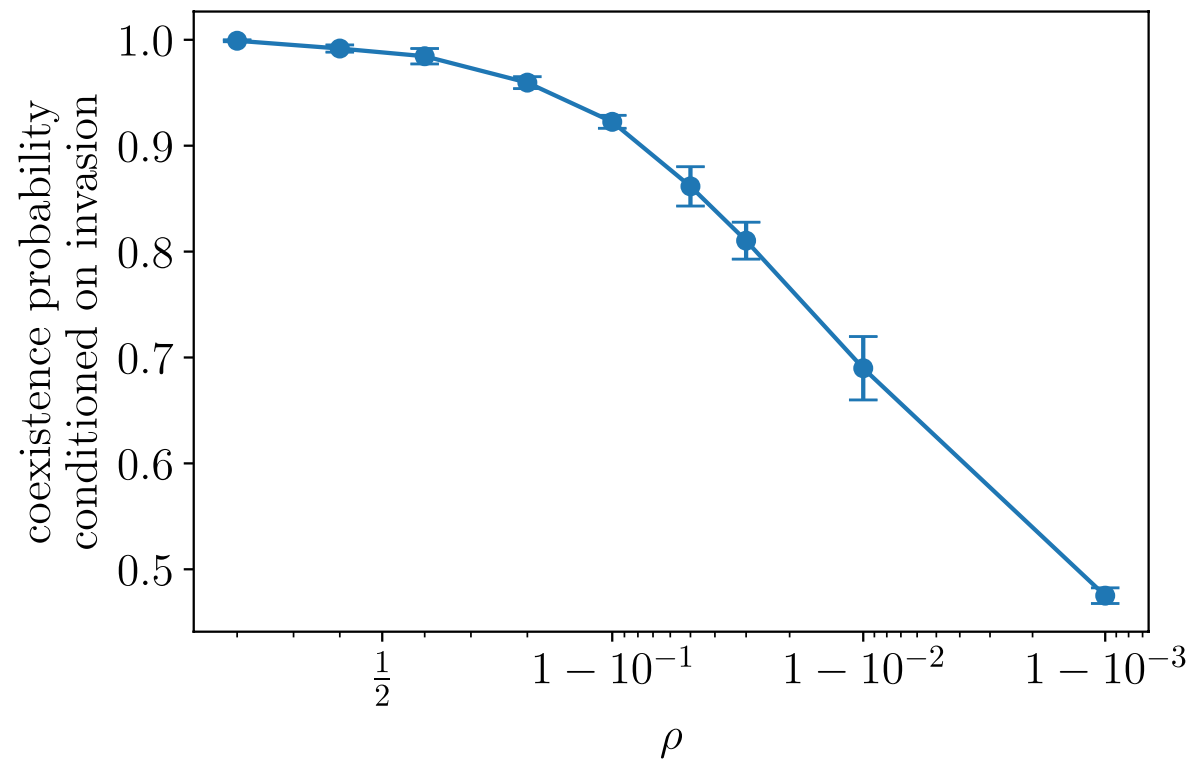


**Appendix 9—figure 1.** The coexistence probability between parent and mutant conditioned on successful invasion of the mutant (with $s_i = 0$). The probability is averaged over 4 simulations of a diversifying community for each value of $\rho$, with error bars showing the standard error. Here $\rho$ is shown on a logit scale to emphasize the difference between the points as $\rho \to 1$. Even for $\rho = 0.999$, the coexistence probability is order 1/2 over he epoch of duration $30\mathcal{M}L$, likely long enough to enable small differences between closely related strains to have an effect.

The main problem with this approximation is that it needs to be used when the biases are close to the critical $\xi_c$ which is substantially negative, while the approximation of ignoring migration certainly breaks down in this regime. Properly understanding the coexistence of replacement of related mutant and parent requires a fuller understanding of their coupled dynamics, including migration.

## Coexistence probability

Of particular interest is the probability with which mutant and parent coexist when they are correlated by amount $\rho$. The effects of the persistent chaos on this coexistence probability are subtle, and here we show that the simulation results cannot be captured by the null model of 'Appendix 9', which neglects migration and blooms.

Comparing numerical results from simulation of the dynamics to those of 'Appendix 9', we find that the probability of coexistence between parent and mutant, conditioned on mutant invasion, is substantially larger than predicted by ***Equation A9.1*** and ***Equation A9.2***. This is likely due to the fact that in the STC, a majority of persistent strains have negative bias and even for those with positive bias, the effect of migration is not small. The boom-bust dynamics with migration makes coexistence much easier, but understanding this even semi-quantitatively is challenging because of the correlations in the dynamical drive which makes parental blooms suppress the mutants and vice versa. Nevertheless, we expect that if $\rho$ is extremely close to unity, the coexistence probability of parent and mutant will go to zero as $\sqrt{1-\rho}$, as in the simple approximation from 'Appendix 9', but with a small coefficient that might be of order some inverse power of $\mathcal{M}$. In addition one should note that the natural timescale over which differences between the parent and mutant will accumulate is of order $(1-\rho^2)^{-1/2}\mathcal{M}\sqrt{L}$ with the coefficient only $\cong 20$ for $\rho = 0.999$. Therefore even for $\rho$ this close to 1, the typical differences between parent and mutant will still be felt on timescales of order $\mathcal{M}L$ over which the close-to-marginal strains rise or die out.

A dynamical analysis of a parent and mutant including migration becomes far more complicated once the community has evolved long enough that a substantial fraction of the strains have turned over (or coexist with relatives). The underlying simplification that makes DMFT valid for assembled communities, the approximate independence of the biases, will break down because of correlations among many of the strains due to their common ancestry. While some of the heuristic behavior of such communities and how they evolve may not change much even for $\rho$ quite close to 1, there will certainly be quantitative changes.

The approximate Markovian analysis of the evolution of the bias distribution ('Evolution without general fitness differences'), can be modified to roughly account for correlations between mutants and parents, $\rho > 0$. Such correlations will modify the effects of invading strains on the bias of the extant strains. This is most readily understood in the limit that $\rho$ is very close to unity. In this case, one expects that the total abundance of the parent and mutant after the invasion will be similar to

that of the parent before the invasion and, because of the high correlations, their effects on the other strains will be very similar. This suggests that both the stochastic and systematic changes of the biases should be multiplied by a factor of order $1-\rho^2$ on the right hand side of ***Equation 3***. For the first few mutants, the analysis could be carried through similarly. However once the number of successfully invaded mutants is comparable to the original $L_0$, the correlations between the biases of the extant strains can not be ignored and the approximation of independent $\zeta_i$ breaks down. We leave analysis of this for future work.