## [Editor Report]

This important study explores the question of “what gives rise to diversity in ecological settings?”. By considering the interplay between ecology and evolution, this study proposes a scenario of spatiotemporal chaos, in which interactions between strains drive large changes in the relative abundances of strains. The presented theoretical approach is compelling and goes beyond the current state of the art. This innovative theoretical work is of broad interest to the field of ecology and evolution.

---

## [Decision Letter]

**Decision letter after peer review:**

Thank you for submitting your article "Spatiotemporal Ecological Chaos Enables Gradual Evolutionary Diversification Without Niches or Tradeoffs" for consideration by *eLife*. Your article has been reviewed by 2 peer reviewers, and the evaluation has been overseen by a Reviewing Editor and Aleksandra Walczak as the Senior Editor. The following individual involved in the review of your submission has agreed to reveal their identity: Pankaj Mehta (Reviewer #1).

*Reviewer #1 (Public review):*

In there paper "Spatiotemporal Ecological Chaos Enables Gradual Evolutionary Diver- sification Without Niches or Tradeoffs", Mahadevan, Pearce, and Fisher build on previous works to explore a compelling potential answer (what they term a "scenario") to an important open and fascinating question: what gives rise to micro strain-level diversity?

Naively, the ecological principle of competitive exclusion would suggest that closely related strains should not be able to co-exist. However, Fisher and collaborators have previously proposed an interesting and potentially novel and powerful solution to this paradox. If the species have (extremely) anti-correlated species-species interactions (i.e more A helps B but more B hurts A) then there is a reasonably large set of parameters under which you can have infinite diversity due to spatial temporal chaos (STC). This is really an interesting and compelling picture.

The purpose of this paper is to explore a natural follow up question: does the STC phase still support infinite diversity even when communities are assembled using evolution, or more accurately undergo evolution starting with a sufficiently large randomly assembled community? In other words, is STC still a reasonable explanation for strain-level diversity once evolution is considered. This is an extremely interesting question since evolution and ecology are so deeply intertwined at the time scales on which strain evolve. The work is especially impressive due to the extreme dearth of analytic and computational tools to really understand eco-evolutionary dynamics.

*Reviewer #2 (Public review):*

The manuscript sets out to explain how the large micro-diversity of closely-related microbial strains might be produced and maintained. It proposes a scenario of spatiotemporal chaos, in which interactions between strains drive large changes in the relative abundances; space helps strains survive by migration between islands; and evolution produces new strains. The work presents a mathematical framework and discusses its biological relevance, and then examines its outcomes through a combination of simulations and mathematical analysis.

An important main result is that, under certain conditions, the diversity (number of extant strains) can grow continually and indefinitely. It is presented through simulations and then analyzed theoretically. Much of the work goes into understanding this increase in diversity, and the conditions required for it to happen. In particular, the effects of the distribution of mutant fitnesses, and of correlations between mutant and parent are examined.

This main result represents a significant conceptual advance on a central question in ecology, that justifies publication of this work. It is of broad interest in the field of ecology and evolution, likely to generate significant interest and lead to future work in a number of directions.

The simulations strongly support the results. The mathematical analysis provides significant insight into the phenomenology. It also develops tools that are be of interest in their own right.

In tackling this difficult and general question, the authors must make simplifying assumptions in the modeling.

The theoretical model does not assume the existence of niches (in the form of significant differences between inter- and intra-species interactions), or fine-tuned tradeoffs except as they may emerge from the evolutionary process. That such assumptions are not made is very appealing for microbial ecology.

The interactions between species are taken to be anti-symmetric or close to that, as in predator-prey interactions. The authors motivate this assumption by bacteria-phage interactions. As the authors note, in a community with many strains of both bacteria and phage, the interactions are also expected to have a block structure, with different interactions between and within each group. This additional block structure could potentially have significant effect on the phenomenology. It is not implemented in the present work, and only briefly discussed in the Discussion section, referring to unpublished work-in-progress.